# The Trade-off between Universality and Label Efficiency of Representations from Contrastive Learning

**Zhenmei Shi**[1*] **Jiefeng Chen**[1*] **Kunyang Li**[1] **Jayaram Raghuram**[1] **Xi Wu**[2] **Yingyu Liang**[1] **Somesh Jha**[1,3]
[1] University of Wisconsin-Madison    [2] Google LLC    [3] XaiPient    [*] Equal contribution
{zhmeishi,jiefeng,kli253,jayaramr,yliang,jha}@cs.wisc.edu, wuxi@google.com

## Abstract

Pre-training representations (a.k.a. foundation models) has recently become a prevalent learning paradigm, where one first pre-trains a representation using large-scale unlabeled data, and then learns simple predictors on top of the representation using small labeled data from the downstream tasks. There are two key desiderata for the representation: *label efficiency* (the ability to learn an accurate classifier on top of the representation with a small amount of labeled data) and *universality* (usefulness across a wide range of downstream tasks). In this paper, we focus on one of the most popular instantiations of this paradigm: contrastive learning with linear probing, i.e., learning a linear predictor on the representation pre-trained by contrastive learning. We show that there exists a trade-off between the two desiderata so that one may not be able to achieve both simultaneously. Specifically, we provide analysis using a theoretical data model and show that, while more diverse pre-training data result in more diverse features for different tasks (improving universality), it puts less emphasis on task-specific features, giving rise to larger sample complexity for down-stream supervised tasks, and thus worse prediction performance. Guided by this analysis, we propose a contrastive regularization method to improve the trade-off. We validate our analysis and method empirically with systematic experiments using real-world datasets and foundation models.

## 1 Introduction

Representation pre-training is a recent successful approach that utilizes large-scale unlabeled data to address the challenges of scarcity of labeled data and distribution shift. Different from the traditional supervised learning approach using a large labeled dataset, representation learning first pre-trains a representation function using large-scale diverse unlabeled datasets by self-supervised learning (e.g., contrastive learning), and then learns predictors on the representation using small labeled datasets for downstream target tasks. The pre-trained model is commonly referred to as a *foundation model* (Bommasani et al., 2021), and has achieved remarkable performance in many applications, e.g., BERT (Devlin et al., 2019), GPT-3 (Brown et al., 2020), CLIP (Radford et al., 2021), and Flamingo (Alayrac et al., 2022). To this end, we note that there are two properties that are key to their success: (1) *label efficiency*: with the pre-trained representation, only a small amount of labeled data is needed to learn accurate predictors for downstream target tasks; (2) *universality*: the pre-trained representation can be used across various downstream tasks.

In this work, we focus on *contrastive learning with linear probing* that learns a linear predictor on the representation pre-trained by contrastive learning, which is an exemplary pre-training approach (e.g., (Arora et al., 2019; Chen et al., 2020)). We highlight and study a fundamental trade-off between label efficiency and universality, though ideally, one would like to have these two key properties simultaneously. Since pre-training with large-scale diverse unlabeled data is widely used in practice, such a trade-off merits deeper investigation.

Theoretically, we provide an analysis of the features learned by contrastive learning, and how the learned features determine the downstream prediction performance and lead to the trade-off. We

propose a *hidden representation data model*, which first generates a hidden representation containing various features, and then uses it to generate the label and the input. We first show that contrastive learning is essentially generalized nonlinear PCA that can learn *hidden features invariant to the transformations* used to generate positive pairs. We also point out that additional assumptions on the data and representations are needed to obtain non-vacuous guarantees for prediction performance. We thus consider a setting where the data are generated by linear functions of the hidden representation, and formally prove that the difference in the learned features leads to the trade-off. In particular, pre-training on more diverse data learns more diverse features and is thus useful for prediction on more tasks. But it also down-weights task-specific features, implying larger sample complexity for predictors and thus worse prediction performance on a specific task. This analysis inspires us to propose a general method – *contrastive regularization* – that adds a contrastive loss to the training of predictors to improve the accuracy on downstream tasks.

Empirically, we first perform controlled experiments to reveal the trade-off. Specifically, we first pre-train on a specific dataset similar to that of the target task, and then incrementally add more datasets into pre-training. In the end, the pre-training data includes both datasets similar to the target task and those not so similar, which mimics the practical scenario that foundation models are pre-trained on diverse data to be widely applicable for various downstream tasks. Fig. 1 gives an example of this experiment: As we increase task diversity for contrastive learning, it increases the average accuracy on *all tasks* from 18.3% to 20.1%, while it harms the label efficiency of an individual task, on CIFAR-10 the accuracy drops from 88.5% to 76.4%. We also perform experiments on contrastive regularization, and demonstrate that it can *consistently improve* over the typical fine-tuning method across multiple datasets. In several cases, the improvement is significant: 1.3% test accuracy improvement for CLIP on ImageNet, 4.8% for MoCo v3 on GTSRB (see Table 1 and 2 for details). With these results, we believe that it is of importance to bring the community's attention to this trade-off and the forward path of foundation models.

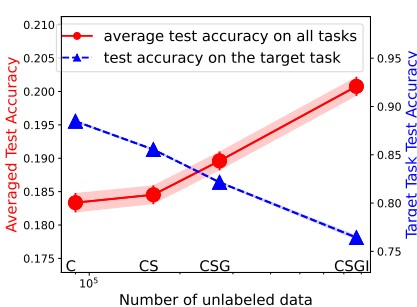

Figure 1: Illustration of the trade-off between universality and label efficiency. $x$-axis: from left to right, incrementally add CINIC-10 (C), SVHN (S), GTSRB (G), and ImageNet32 (I) for pre-training MoCo v2. For example, "CS" means CINIC-10+SVHN. The average test accuracy of prediction on all 4 datasets (red line) increases with more diverse pre-training data, while that on the target task CIFAR-10 (blue line) decreases. (The variance of the blue line is too small to be seen.) Please refer to Section 3.1 for details.

Our main contributions are summarized as follows:

- We propose a hidden representation data model and prove that contrastive learning is essentially generalized nonlinear PCA, and can encode hidden features invariant to the transformations used in positive pairs (Section 2.1).

- We formally prove the trade-off in a simplified setting with linear data (Section 2.2).

- We empirically demonstrate the trade-off across different methods and datasets for contrastive learning with linear probing (Section 3.1 and 3.2).

- We propose a contrastive regularization method for training the predictor on a target task (Section 2.2), which achieves consistent improvement in our experiments (Section 3.3).

**Related Work on Representation Pre-training.** This paradigm pre-trains a representation function on a large dataset and then uses it for prediction on various downstream tasks (Devlin et al., 2019; Kolesnikov et al., 2020; Brown et al., 2020; Newell & Deng, 2020). The representations are also called foundation models (Bommasani et al., 2021). There are mainly two kinds of approaches: (1) supervised approaches (e.g., (Kolesnikov et al., 2020)) that pre-train on large labeled datasets; (2) self-supervised approaches (e.g., (Newell & Deng, 2020)) that pre-train on large and diverse unlabeled datasets. Recent self-supervised pre-training can compete with or outperform supervised pre-training on the downstream prediction performance (Ericsson et al., 2021). Practical examples like BERT (Devlin et al., 2019), GPT-3 (Brown et al., 2020), CLIP (Radford et al., 2021),

DALL·E (Ramesh et al., 2022), PaLM (Chowdhery et al., 2022) and Flamingo (Alayrac et al., 2022) have obtained effective representations universally useful for a wide range of downstream tasks.

A popular method is contrastive learning, i.e., to distinguish matching and non-matching pairs of augmented inputs (e.g., (van den Oord et al., 2018; Chen et al., 2020; He et al., 2020; Grill et al., 2020; Chen & He, 2021; Zbontar et al., 2021; Gao et al., 2021)). Some others solve "pretext tasks" like predicting masked parts of the inputs (e.g.,(Doersch et al., 2015; Devlin et al., 2019)).

**Related Work on Analysis of Self-supervised Pre-training.** There exist abundant studies analyzing self-supervised pre-training (Arora et al., 2019; Tsai et al., 2020; Yang et al., 2020; Wang & Isola, 2020; Garg & Liang, 2020; Zimmermann et al., 2021; Tosh et al., 2021; HaoChen et al., 2021; Wen & Li, 2021; Liu et al., 2021; Kotar et al., 2021; Van Gansbeke et al., 2021; Lee et al., 2021; Saunshi et al., 2022a; Shen et al., 2022; Kalibhat et al., 2022). They typically focus on pre-training or assume the same data distribution in pre-training and prediction. Since different distributions are the critical reason for the trade-off we focus on, we provide a new analysis. Some studies have connected contrastive learning to component analysis (Balestriero & LeCun, 2022; Tian, 2022; Ko et al., 2022). Our analysis focuses on the trade-off, while also showing a connection to PCA based on our notion of invariant features and is thus fundamentally different. Recently, Cole et al. have attempted to identify successful conditions for contrastive learning and pointed out that diverse pre-training data can decrease prediction performance compared to pre-training on the specific task data. However, they do not consider universality and provide no systematic study. Similarly, Bommasani et al. call for more research on specialization vs. diversity in pre-training data but provide no study. We aim to provide a better understanding of the trade-off between universality and label efficiency.

## 2 THEORETICAL ANALYSIS

Our experiments in Section 3.1 demonstrate a trade-off between the universality and label efficiency of contrastively pre-trained representations when used for prediction on a distribution different from the pre-training data distribution. See Fig. 1 for an example. Intuitively, from the unlabeled data, pre-training can learn semantic features useful for prediction on even different data distributions. To analyze this, we need to formalize the notion of useful semantic features. So we introduce a *hidden representation data model* where a hidden representation (i.e., a set of semantic features) is sampled and then used for generating the data. Similar models have been used in some studies (HaoChen et al., 2021; Zimmermann et al., 2021), while we introduce the notion of spurious and invariant features and obtain a novel analysis for contrastive learning.

Using this theoretical model of data, Section 2.1 investigates what features are learned by contrastive learning. We show that *contrastive learning can be viewed as a generalization of Principal Components Analysis, and it encodes the invariant features not affected by the transformations but removes the others*. We also show that further assumptions on the data and the representations are needed necessary for any non-vacuous bounds for downstream prediction. So Section 2.2 considers a simplified setting with linear data. We show that when pre-trained on diverse datasets (modeled as a mixture of unlabeled data from different tasks), it encodes all invariant features from the different tasks and thus is useful for all tasks. On the other hand, it essentially emphasizes those that are shared among the tasks, but down-weights those that are specific to a single task. Compared to pre-training only on unlabeled data from the target task, this then leads to a larger sample complexity and thus worse generalization for prediction on the target task. Therefore, we show that the trade-off between universality and label efficiency occurs due to the fact that *when many useful features from diverse data are packed into the representation, those for a specific target task can be down-weighted and thus worsen the prediction performance on it*. Based on this insight, we propose a contrastive regularization method for using representations in downstream prediction tasks, which achieves consistent improvement over the typical fine-tuning method in our experiments in Section 3.3.

**Contrastive Learning.** Let $\mathcal{X} \subseteq \mathbb{R}^d$ denote the input space, $\mathcal{Y}$ the label space, and $\overline{\mathcal{Z}} \subseteq \mathbb{R}^k$ the output vector space of the learned representation function. Let $\Phi$ denote the hypothesis class of representations $\phi : \mathcal{X} \to \overline{\mathcal{Z}}$, and $\mathcal{F}_\phi$ the hypothesis class of predictors on $\phi$. A task is simply a data distribution over $\mathcal{X} \times \mathcal{Y}$. In pre-training, using transformations on unlabeled data from the tasks, we have some pre-train distribution $\mathcal{D}_{pre}$ over positive pairs $(x, x^+)$ and negative examples $x^-$, where $x, x^+$ are obtained by applying random transformations on the same input (e.g., cropping or color jitter for images), and $x^-$ is an independent example. The contrastive loss is $\ell\left(\phi(x)^\top(\phi(x^+) - \phi(x^-))\right)$

where $\ell(t)$ is a suitable loss function. Typically, the logistic loss $\ell(t) = \log(1 + \exp(-t))$ is used, while our analysis also holds for other loss functions. A representation $\phi$ is learned by:

$$\min_{\phi \in \Phi} \; \mathbb{E}_{(x,x^+,x^-) \sim \mathcal{D}_{pre}} \left[ \ell \left( \phi(x)^\top (\phi(x^+) - \phi(x^-)) \right) \right]. \tag{1}$$

(We simply consider the population loss since pre-training data are large-scale.) Then a predictor $f$ is learned on top of $\phi$ using $m$ labeled points $\{(x_i, y_i)\}_{i=1}^m$ from a specific target task $\mathcal{D}$:

$$\min_{f \in \mathcal{F}_\phi} \; \frac{1}{m} \sum_{i=1}^m \ell_c(f(\phi(x_i)), y_i) \tag{2}$$

where $\ell_c$ is a prediction loss (e.g. cross-entropy). Usually, $f$ is a linear classifier (Linear Probing) with a bounded norm: $\mathcal{F}_\phi = \{f(z) = u^\top z : u \in \mathbb{R}^k, \|u\| \leq B\}$, where $\|\cdot\|$ denotes the $\ell_2$ norm.

**Hidden Representation Data Model.** We now consider the pre-train distribution $\mathcal{D}_{pre}$ over $(x, x^+, x^-)$. To capture that pre-training can learn useful features, we assume a hidden representation for generating the data: first sample a hidden representation $z \in \mathcal{Z}$ from a distribution $\mathcal{D}_z$ over some hidden representation space $\mathcal{Z} \subseteq \mathbb{R}^d$, and then generate the input $x$ and the label $y$ from $z$. (The space $\mathcal{Z}$ models semantic features, and can be different from the learned representation space $\overline{\mathcal{Z}}$.) The dimensions of $z$ are partitioned into two disjoint subsets of $[d] := \{1, \cdots, d\}$: *spurious features* $U$ that are affected by the transformations, and *invariant features* $R$ that are not. Specifically, let $\mathcal{D}_U, \mathcal{D}_R$ denote the distributions of $z_U$ and $z_R$, respectively, and let $x = g(z)$ denote the generative function for $x$. Then the positive pairs $(x, x^+)$ are generated as follows:

$$z = [z_R; z_U] \sim \mathcal{D}_z, z_U^+ \sim \mathcal{D}_U, \; z^+ = [z_R; z_U^+], \quad x = g(z), x^+ = g(z^+). \tag{3}$$

That is, $x, x^+$ are from the same $z_R$ but two random copies of $z_U$ that model the random transformations. Finally, $x^-$ is an i.i.d. sample from the same distribution as $x$: $z^- \sim \mathcal{D}_z, x^- = g(z^-)$.

## 2.1 What Features are Learned by Contrastive Learning?

To analyze prediction performance, we first need to analyze what features are learned in pre-training.

**Contrastive Learning is Generalized Nonlinear PCA.** Recall that given data $x$ from a distribution $\mathcal{D}$, Principal Components Analysis (PCA) (Pearson, 1901; Hotelling, 1933) aims to find a linear projection function $\phi$ on some subspace such that the variance of the projected data $\phi(x)$ is maximized, i.e., it is minimizing the following PCA objective:

$$-\mathbb{E}_{x \sim \mathcal{D}}[\|\phi(x) - \mathbb{E}_{x' \sim \mathcal{D}}[\phi(x')]\|^2] = -\mathbb{E}_{x \sim \mathcal{D}}[\|\phi(x) - \phi_0\|^2] \tag{4}$$

where $\phi_0 := \mathbb{E}[\phi(x')]$ is the mean of the projected data. Nonlinear PCA replaces linear representation functions $\phi$ with nonlinear ones. We next show that contrastive learning is a generalization of nonlinear PCA on the smoothed representation after smoothing out the transformations.

**Theorem 2.1.** *If $\ell(t) = -t$, then the contrastive loss is equivalent to the PCA objective on $\phi_{z_R}$:*

$$\mathbb{E} \left[ \ell \left( \phi(x)^\top [\phi(x^+) - \phi(x^-)] \right) \right] = -\mathbb{E} \left[ \|\phi_{z_R} - \phi_0\|^2 \right] \tag{5}$$

*where $\phi_{z_R} := \mathbb{E}[\phi(x) \mid z_R] = \mathbb{E}[\phi(g(z)) \mid z_R]$. If additionally $\phi(x)$ is linear in $x$, then it is equivalent to the linear PCA objective $-\mathbb{E} \left[ \|\phi(\bar{x}) - \phi_0\|^2 \right]$ on data $\bar{x} := \mathbb{E}[x|z_R] = \mathbb{E}[g(z)|z_R]$.*

So contrastive learning is essentially nonlinear PCA when $\ell(t) = -t$, and further specializes to linear PCA when the representation is linear. As PCA finds directions with large variances, the analogue is that contrastive learning encodes important invariant features but not spurious ones.

**Contrastive Learning Encodes Invariant Features and Removes Spurious Features.** For a formal statement we need some weak assumptions on the data, the representations, and the loss:

**(A1)** $z_R$ can be recovered from $x$, i.e., the inputs $x = g(z)$ from different $z_R$'s are disjoint.

**(A2)** The representation functions are the *regular* functions with $\|\phi(x)\| = B_r$ $(\forall x)$ for some $B_r > 0$. Being regular means there are a finite $L$ and a partition of $\mathcal{Z}$ into a finite number of subsets, such that in each subset all $\phi \circ g$ have Lipschitz constants bounded by $L$.

**(A3)** The loss $\ell(t)$ is convex, decreasing, and lower-bounded.

The first condition means the invariant features $z_R$ can be extracted from $x$ (note that $g$ need not be invertible). The regular condition on the representation is to exclude some pathological cases like the Dirichlet function; essentially reasonable functions relevant for practice satisfy this condition, e.g., when $g$ is Lipschitz and $\phi$ are neural networks with the ReLU activation. Also, note that the logistic loss typically used in practice satisfies the last condition.

We say a function $f(z)$ is independent of a subset of input dimensions $z_S$, if there exists a function $f'$ such that $f(z) = f'(z_{-S})$ with probability 1, where $z_{-S}$ denotes the set of all $z_j$ with $j \notin S$. We say the representation $\phi$ encodes a feature $z_i$, if $\phi \circ g : \mathcal{Z} \to \overline{\mathcal{Z}}$ is not independent of $z_i$ as long as the generative function $g(z)$ is not independent of $z_i$.

**Theorem 2.2.** *Under Assumptions (A1)(A2)(A3), the optimal representation $\phi^*$ satisfies:*

*(1) $\phi^*$ does not encode the spurious features $z_U$: $\phi^* \circ g(z)$ is independent of $z_U$.*

*(2) For any invariant feature $i \in R$, there exists $B_i > 0$ such that as long as the representations' norm $B_r \geq B_i$, then $\phi^*$ encodes $z_i$. Furthermore, if $\mathcal{Z}$ is finite, then $B_i$ is monotonically decreasing in $\Pr[z_{R \setminus \{i\}} = z^-_{R \setminus \{i\}}, z_i \neq z^-_i]$, the probability that in $z_R$ and $z^-_R$, the $i$-th feature varies while the others remain the same.*

So contrastive learning aims to remove the spurious features and preserve the invariant features. Then the transformations should be chosen such that they will not affect the useful semantic features, but change those irrelevant to the label. Interestingly, the theorem further suggests that contrastive learning tends to favor the more "spread-out" invariant features $z_i$, as measured by $\Pr[z_{R \setminus \{i\}} = z^-_{R \setminus \{i\}}, z_i \neq z^-_i]$. As we increase the representation capacity $B_r$, $B_r$ passes the threshold $B_i$ for more features $z_i$, so $\phi^*$ first encodes the more spread-out invariant features and then the others.

This further suggests the following intuition for the trade-off. When pre-trained on diverse data modeled as a mixture from multiple tasks with different invariant features, the representation encodes all the invariant features and thus is useful for prediction on all the tasks. When pre-trained on only a specific task, features specific to this task are favored over those that only show up in other tasks, which leads to smaller sample complexity for learning the predictor and thus better prediction. However, to formalize this, some inductive bias assumptions about the data and the representation are necessary to get any non-vacuous guarantee for the prediction (see discussion in Appendix A.1). Therefore, Section 2.2 introduces additional assumptions and formalizes the trade-off.

## 2.2 ANALYZING THE TRADE-OFF: LINEAR DATA

To analyze the prediction performance, we first need to model the relation between the pre-training data and the target task. We model the diverse pre-training data as a mixture of data from $T$ different tasks $\mathcal{D}_t$'s, while the target task is one of the tasks. All tasks share a public feature set $S$ of size $s$, and each task $\mathcal{D}_t$ additionally owns a private disjoint feature set $P_t$ of size $r - s$, i.e., $P_t \cap S = \emptyset$ and $P_{t_1} \cap P_{t_2} = \emptyset$ for $t_1 \neq t_2$ (Fig. 2). The invariant features for $\mathcal{D}_t$ are then $R_t = S \cup P_t$. All invariant features are $R = \cup_{t=1}^T R_t$, and spurious features are $U = [d] \setminus R$. In task $\mathcal{D}_t$, the $(x, x^+)$ are generated as follows:

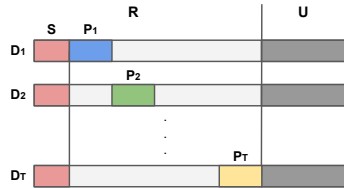

Figure 2: Illustration of the features in our data distributions.

$$z_{R_t} \sim \mathcal{N}(0, I),\ z_{R \setminus R_t} = 0,\ z_U \sim \mathcal{N}(0, I), z = [z_R; z_U],\quad x = g(z), \tag{6}$$

$$z_U^+ \sim \mathcal{N}(0, I), z^+ = [z_R; z_U^+],\quad x^+ = g(z^+), \tag{7}$$

and $x^-$ is simply an i.i.d. copy from the same distribution as $x$. In practice, multiple independent negative examples are used, and thus we consider the following contrastive loss $\min_{\phi \in \Phi}\ \mathbb{E}_{(x,x^+)}\left[\ell\left(\phi(x)^\top (\phi(x^+) - \mathbb{E}_{x^-}\phi(x^-))\right)\right]$ for a convex and decreasing $\ell(t)$ to pre-train a representation $\phi$. Then, when using $\phi$ for prediction in the target task $\mathcal{D}_t$, the predictor class should contain a predictor matching the ground-truth label:

$$\mathcal{F}_{\phi,t} = \{f(z) = u^\top z : u \in \mathbb{R}^k, \|u\| \leq B_{\phi,t}\} \tag{8}$$

where $B_{\phi,t}$ is the minimum value such that there exists $u_t \in \mathcal{F}_{\phi,t}$ with $y = u_t^\top \phi(x)$ on $\mathcal{D}_t$.

Now, given the necessity of inductive biases for non-vacuous guarantees (see Appendix A.1), and inspired by classic dictionary learning and recent analysis on such data (e.g., Olshausen & Field (1997); Wen & Li (2021); Shi et al. (2022)), we assume linear data and linear representations:

- $x$ is linear in $z$: $x = g(z) = Mz$ where $M \in \mathbb{R}^{d \times d}$ is an orthonormal dictionary. Since linear probing has strong performance on pre-trained representations, we thus assume that the label in each task $t$ is linear in its invariant features $y = (u_t^*)^\top z_{R_t}$ for some $u_t^* \in \mathbb{R}^r$.

- The representations are linear functions with weights of bounded spectral/Frobenius norms:

$$\Phi = \{\phi(x) = Wx : W \in \mathbb{R}^{k \times d}, \|W\| \leq 1, \|W\|_F \leq \sqrt{r}\}.$$

Here the norm bounds are chosen to be the minimum values to allow recovering the invariant features in the target task, i.e., there exists $\phi \in \Phi$ such that $\phi(x) = [z_{R_t}; \mathbf{0}]$.

We compare two representations: a specific one pre-trained on unlabeled data from the target task $\mathcal{D}_t$, and a universal one pre-trained on an even mixture of data from $T$ tasks. (Appendix B provides analysis for more general cases like uneven mixtures.) This captures the situation that the pre-training data contains some data similar to the target task and also other less similar data. Let $v_{t,1} = \sum_{j \in S}(u_t^*)_j^2$ and $v_{t,2} = \sum_{j \in P_t}(u_t^*)_j^2$ be the weights on the shared and task-specific invariant features, respectively. Also, assume the prediction loss $\ell_c$ is $L$-Lipschitz.

**Proposition 2.3.** *The representation $\phi^*$ obtained on an even mixture of data from all the tasks $\{\mathcal{D}_t : 1 \leq t \leq T\}$ satisfies $\phi^* \circ g(z) = Q\left(\sum_{j \in S} \sqrt{\alpha} z_j e_j + \sum_{j \in R \setminus S} \sqrt{\beta} z_j e_j\right)$ for some $\alpha \in [0, 1]$, $\beta = \min\left(1, \frac{r - \alpha s}{T(r - s)}\right)$, where $e_j$'s are the basis vectors and $Q$ is any orthonormal matrix. The Empirical Risk Minimizer $\hat{u} \in \mathcal{F}_{\phi^*, t}$ on $\phi^*$ using $m$ labeled data points from $\mathcal{D}_t$ has risk*

$$\mathbb{E}_{(x,y) \sim \mathcal{D}_t}[\ell_c(\hat{u}^\top \phi^*(x), y)]$$

$$\leq 4L\sqrt{\frac{1}{m}\left(\frac{v_{t,1}}{\alpha} + \frac{v_{t,2}}{\beta}\right)}\left(\sqrt{s\alpha + (r - s)\beta} + O\left(\sqrt{\frac{r}{s\alpha + (r - s)\beta}}\right)\right) + 8\sqrt{\frac{2\ln(4/\delta)}{m}}.$$

**Proposition 2.4.** *The representation $\phi_t^*$ obtained on data from $\mathcal{D}_t$ satisfies $\phi_t^* \circ g(z) = Q\left(\sum_{j \in R_t} z_j e_j\right)$ where $e_j$'s are the basis vectors and $Q$ is any orthonormal matrix. The Empirical Risk Minimizer $\hat{u} \in \mathcal{F}_{\phi_t^*, t}$ on $\phi_t^*$ using $m$ labeled data points from $\mathcal{D}_t$ has risk*

$$\mathbb{E}_{(x,y) \sim \mathcal{D}_t}[\ell_c(\hat{u}^\top \phi_t^*(x), y)] \leq 4L\sqrt{\frac{r}{m}}\|u_t^*\| + 8\sqrt{\frac{2\ln(4/\delta)}{m}}.$$

*While on task $\mathcal{D}_i(i \neq t)$, any linear predictor on $\phi_t^*$ has error at least $\min_u \mathbb{E}_{\mathcal{D}_i}[\ell_c(u^\top z_S, y)]$.*

**Difference in Learned Features Leads to the Trade-off.** The key of the analysis (in Appendix B) is about what features are learned in the representations. Pre-trained on all $T$ tasks, $\phi^*$ is a rotation of the weighted features, where the shared features are weighted by $\sqrt{\alpha}$ and task-specific ones are weighted by $\sqrt{\beta}$. Pre-trained on one task $\mathcal{D}_t$, $\phi_t^*$ is a rotation of the task-specific features $R_t$. So compared to $\phi_t^*$, $\phi^*$ encodes all invariant features but down-weights the task-specific features $P_t$.

The difference in the learned features then determines the prediction performance and results in a trade-off between universality and label efficiency: compared to $\phi_t^*$, $\phi^*$ is useful for more tasks but has worse performance on the specific task $\mathcal{D}_t$. For illustration, suppose $r = 2s$, and the shared and task-specific features are equally important for the labels on the target task: $v_{t,1} = v_{t,2} = \|u_t^*\|^2/2$. In Appendix B.3 we show that $\phi^*$ has $\alpha = 1, \beta = \frac{1}{T}$ and the error is $O\left(L\sqrt{\frac{Tr}{m}}\|u_t^*\|\right)$, while the error using $\phi_t^*$ is $O\left(L\sqrt{\frac{r}{m}}\|u_t^*\|\right)$. Therefore, the error when using representations pre-trained on data from $T$ tasks is $O(\sqrt{T})$ worse than that when just pre-training on data from the target task. On the other hand, the former can be used in all $T$ tasks and the prediction error diminishes with the labeled data number $m$. While the latter only encodes $R_t$ and the only useful features on the other tasks are $z_S$, then even with infinite labeled data the error can be large ($\geq \min_u \mathbb{E}[\ell_c(u^\top z_S, y)]$, the approximation error using only the common features $z_S$ for prediction).

**Improving the Trade-off via Contrastive Regularization.** The above analysis provides some guidance on improving the trade-off, in particular, improving the target prediction accuracy when

given a pre-trained representation $\phi^*$. It suggests that when $\phi^*$ is pre-trained on diverse data, one can update it by contrastive learning on some unlabeled data from the target task, which can get better features and better predictions. This is indeed the case for the illustrative example above. We can show that updating $\phi^*$ by contrastive learning on $\mathcal{D}_t$ can increase the weights $\beta$ on the task-specific features $z_{P_t}$, and thus improve the generalization error (formal analysis in Appendix B.4).

In practice, typically one will learn the classifier and also fine-tune the representation with a labeled dataset $\{(x_i, y_i)\}_{i=1}^m$ from the target task. We thus propose *contrastive regularization* for fine-tuning: for each data point $(x, y)$, generate contrastive pairs $\mathcal{R} = \{(\tilde{x}, \tilde{x}^+, \tilde{x}^-)\}$ by applying transformations, and add the contrastive loss on these pairs as a regularization term to the classification loss:

$$\ell_c(f(\phi(x)), y) \;+\; \frac{\lambda}{|\mathcal{R}|} \sum_{(\tilde{x}, \tilde{x}^+, \tilde{x}^-) \in \mathcal{R}} \ell\left(\phi(\tilde{x})^\top(\phi(\tilde{x}^+) - \phi(\tilde{x}^-))\right). \tag{9}$$

This method is simple and generally applicable to different models and algorithms. Similar ideas have been used in graph learning (Ma et al., 2021), domain generalization (Kim et al., 2021) and semi-supervised learning (Lee et al., 2022), while we use it in fine-tuning for learning predictors. Our experiments in Section 3.3 show that it can consistently improve the prediction performance compared to the typical fine-tuning approach.

## 3 EXPERIMENTS

We conduct experiments to answer the following questions. (**Q1**) Does the trade-off between universality and label efficiency exist when training on real datasets? (**Q2**) What factors lead to the trade-off? (**Q3**) How can we alleviate the trade-off, particularly in large foundation models? Our experiments provide the following answers: (**A1**) The trade-off widely exists in different models and datasets when pre-training on large-scale unlabeled data and adapting with small labeled data (see Section 3.1). This justifies our study and aligns with our analysis. (**A2**) Different datasets own many private invariant features leading to the trade-off, e.g., FaceScrub and CIFAR-10 do not share many invariant features (see Section 3.2). It supports our analysis in Section 2.2. (**A3**) Our proposed method, Finetune with Contrastive Regularization, can improve the trade-off consistently (see Section 3.3). Please refer to our released code[1] for more details.

### 3.1 VERIFYING THE EXISTENCE OF THE TRADE-OFF

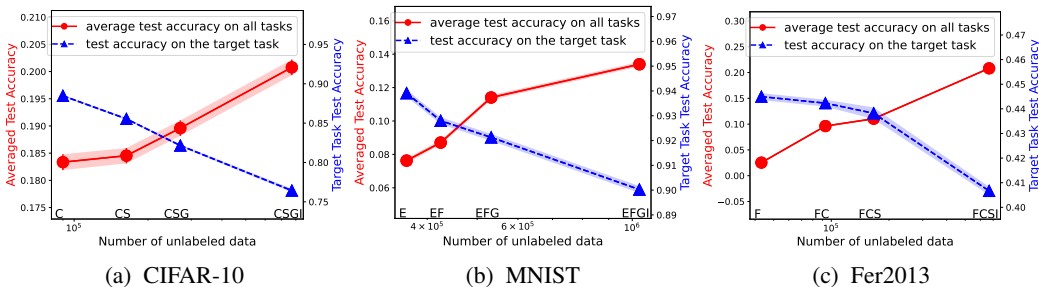

|  (a) CIFAR-10 | (b) MNIST | (c) Fer2013 |

Figure 3: Trade-off between universality and label efficiency for MoCo v2. Appendix C.5 shows similar results for more methods and datasets. $x$-axis: incrementally add datasets for pre-training MoCo v2. (**a**) Pre-training data: CINIC-10 (C), SVHN (S), GTSRB (G), and ImageNet32 (I). E.g., "CS" on the $x$-axis means CINIC-10+SVHN. Target task: CIFAR-10. Red line: average test accuracy of Linear Probing on all 4 datasets. Blue line: test accuracy on the target task. (**b**) EMNIST-Digits&Letters (E), Fashion-MNIST (F), GTSRB (G), ImageNet32 (I). Target: MNIST. (**c**) FaceScrub (F), CIFAR-10 (C), SVHN (S), ImageNet32 (I). Target: Fer2013. Note that training does *not* follow the online learning fashion, e.g., the model will pre-train from scratch (random initialization) on the CSG datasets, rather than using the model pre-trained on the CS datasets.

**Evaluation & Methods.** We first pre-train a ResNet18 backbone (He et al., 2016) with different contrastive learning methods and then do Linear Probing (LP, i.e., train a linear classifier on the feature

---

[1]https://github.com/zhmeisi/trade-off_contrastive_learning

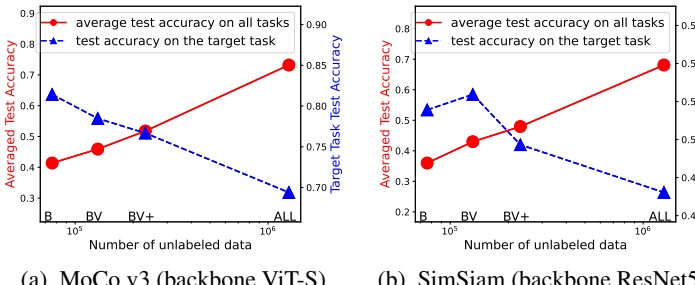

(a) MoCo v3 (backbone ViT-S)      (b) SimSiam (backbone ResNet50)

Figure 4: Trade-off between universality and label efficiency on ImageNet. $x$-axis: from left to right, incrementally add ImageNet-Bird (B), ImageNet-Vehicle (V), ImageNet-Cat/Ball/Shop/Clothing/Fruit (+), and ImageNet (ALL) for pre-training (a) MoCo v3 with backbone ViT-S (b) SimSiam with backbone ResNet50. For example, "BV" means ImageNet-Bird + ImageNet-Vehicle. Target: ImageNet-Bird.

extractor) with the labeled data from the target task. We report the test accuracy on a specific target task and the average test accuracy on all pre-training datasets (i.e., using them as the downstream tasks). Appendix C.2 presents full details and additional results, while Fig. 3 shows the results for the method MoCo v2. The size and diversity of pre-training data are increased on the $x$-axis by incrementally adding unlabeled training data from: (a) CINIC-10, SVHN, GTSRB, ImageNet32 (using only a 500k subset); (b) EMNIST-Digits&Letters, Fashion-MNIST, GTSRB, ImageNet32; (c) FaceScrub, CIFAR-10, SVHN, ImageNet32. We further perform larger-scale experiments: (1) on ImageNet (see Fig. 4); (2) on ImageNet22k and GCC-15M (see Appendix C.2.1).

**Results.** The results show that when the pre-training data becomes more diverse, the average test accuracy on all pre-training datasets increases (i.e., universality improves), while the test accuracy on the specific target task decreases (i.e., label efficiency drops). This shows a clear trade-off between universality and label efficiency. It supports our claim that diverse pre-training data allow learning diverse features for better universality, but can down-weight the features for a specific task resulting in worse prediction. Additional results in the appendix show similar trends (e.g., for methods NNCLR and SimSiam). This validates our theoretical analysis of the trade-off.

## 3.2 INSPECTING THE TRADE-OFF: FEATURE SIMILARITY

Here we compute the similarity of the features learned from different pre-training datasets for a target task. For each pre-trained model, we extract a set of features for the target task Fer2013 using the pre-trained representation function. Then we compute the similarities between the extracted features based on different pre-training dataset pairs using linear Centered Kernel Alignment (CKA) (Kornblith et al., 2019), a widely used tool for high-dimensional feature comparison. Figure 5 reports the results (rows/columns are pre-training data; numbers/colors show

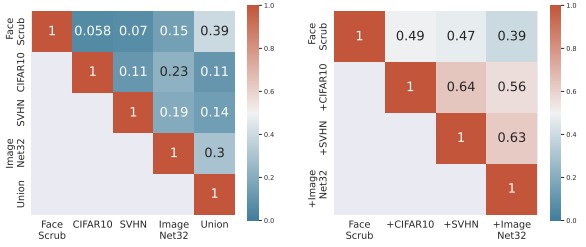

Figure 5: Linear CKA similarity among Fer2013 features from MoCo v2 pre-trained on different datasets. Left: each representation in the first four columns/rows is pre-trained on a single dataset. "Union" indicates the model pre-trained on the union of the four disjoint datasets. Right: from left column to right, from top row to bottom, we incrementally add datasets for pre-training.

the similarity). The left figure shows that the features from different pre-training datasets have low similarities. This is consistent with our setup in Section 2.2 that different tasks only share some features and each owns many private ones. The right figure shows a decreasing trend of similarity along each row. This indicates that when gradually adding more diverse pre-training data, the learned representation will encode more downstream-task-irrelevant features, and become less similar to that prior to adding more pre-training data. Additional results with similar observations, finer-grained investigation into the trade-off, and some ablation studies are provided in Appendix C.3.

### 3.3 Improving the Trade-off: Finetune with Contrastive Regularization

**Evaluation & Methods.** We pre-train ResNet18 by MoCo v2 as in Section 3.1 and report the test accuracy on CIFAR-10 when the predictor is learned by: Linear Probing (LP), Finetune (FT), and Finetune with Contrastive Regularization (Ours). LP follows the training protocol in Section 3.1. FT and Ours learn a linear predictor and update the representation, and use the same data augmentation for a fair comparison. FT follows MAE (He et al., 2022), while Ours uses MoCo v2 contrastive loss and regularization coefficient $\lambda = 0.1$. More details and results are given in Appendix C.4.

| | Pre-training dataset | | | |
| Method | CINIC-10 | +SVHN | +GTSRB | +ImageNet32 |
|---|---|---|---|---|
| LP | 88.41±0.01 | 85.18±0.01 | 82.07±0.01 | 75.64±0.03 |
| FT | 93.58±0.14 | 93.35±0.10 | 93.42±0.13 | 92.92±0.06 |
| Ours | **94.51**±0.02 | **94.26**±0.01 | **94.32**±0.13 | **93.66**±0.12 |

Table 1: Test accuracy on CIFAR-10 with different evaluation methods on MoCo v2 by using all CIFAR-10 training data. From left to right: incrementally add datasets for pre-training.

**Results.** Table 1 shows that our method can consistently outperform the other baselines. In particular, it outperforms the typical fine-tuning method by about $0.7\% - 1\%$, even when the latter also uses the same amount of data augmentation. This confirms the benefit of contrastive regularization. To further support our claim, Fig. 13 in Appendix C.4 visualizes the features of different methods by t-SNE, showing that contrastive regularization can highlight the task-specific features and provide cleaner clustering, and thus improve the generalization, as discussed in our theoretical analysis.

| | CLIP | | | MoCo v3 | | | SimCSE | |
| Method | ImageNet | SVHN | GTSRB | CIFAR-10 | SVHN | GTSRB | IMDB | AGNews |
|---|---|---|---|---|---|---|---|---|
| LP | 77.84±0.02 | 63.44±0.01 | 86.56±0.01 | 95.82±0.01 | 61.92±0.01 | 75.37±0.01 | 86.49±0.16 | 87.76±0.66 |
| FT | 83.65±0.01 | 78.22±0.18 | 90.74±0.06 | 96.17±0.12 | 65.36±0.33 | 76.45±0.29 | 92.31±0.26 | 93.57±0.23 |
| Ours | **84.94**±0.09 | **78.72**±0.37 | **92.01**±0.28 | **96.71**±0.10 | **66.29**±0.20 | **81.28**±0.10 | **92.85**±0.03 | **93.94**±0.02 |

Table 2: Test accuracy for different evaluation methods on different datasets using all training data and using foundation models from CLIP, MoCo v3, and SimCSE. Data augmentation is not used for LP (Linear Probing). For FT (Finetune) and Ours (our method), 10 augmentations to each training images are used for CLIP, MoCo v3, and unique augmentation in each training step is used for SimCSE. More results are in Appendix C.4.1.

**Larger Foundation Models.** We further evaluate our method on several popular real-world large representation models (foundation models). On some of these models, the user may be able to fine-tune the representation when learning predictors. On very large foundation models, the user typically extracts feature embeddings of their data from the models and then trains a small predictor, called adapter (Hu et al., 2021; Sung et al., 2022), on these embeddings. We evaluate CLIP (ViT-L (Dosovitskiy et al., 2020) as the representation backbone), MoCo v3 (ViT-B backbone), and SimCSE (Gao et al., 2021) (BERT backbone). They are trained on (image, text), (image, image), and (text, text) pairs, respectively, so cover a good spectrum of methods. For CLIP and MoCo v3, the backbone is fixed. LP uses a linear classifier, while FT and Ours insert a two-layer ReLU network as an adapter between the backbone and the linear classification layer. Ours uses the SimCLR contrastive loss on the output of the adapter. For SimCSE, all methods use linear classifiers. LP fixes the backbone, while FT and Ours train the classifier and fine-tune the backbone simultaneously. Ours uses the SimCSE contrastive loss on the backbone feature. We set the regularization coefficient $\lambda = 1.0$.

Table 2 again shows that our method can consistently improve the downstream prediction performance for all three models by about $0.4\% - 4.8\%$, and quite significantly in some cases (e.g., 1.3% for CLIP on ImageNet, 4.8% for MoCo v3 on GTSRB). This shows that our method is also useful for large foundation models, even when the foundation models cannot be fine-tuned and only the extracted embeddings can be adapted. Full details and more results are provided in Appendix C.4.1.

## 4 Conclusion and Future Work

In this work, we have shown and analyzed the trade-off between universality and label efficiency of representations in contrastive learning. There are many interesting open questions for future work. (1) What features does the model learn from specific pre-training and diverse pre-training datasets beyond linear data? (2) Do the other self-supervised learning methods have a similar trade-off? (3) Can we address the trade-off better to gain both properties at the same time?

## ACKNOWLEDGMENTS

The work is partially supported by Air Force Grant FA9550-18-1-0166, the National Science Foundation (NSF) Grants CCF-FMitF-1836978, IIS-2008559, SaTC-Frontiers-1804648, CCF-2046710 and CCF-1652140, and ARO grant number W911NF-17-1-0405. Jiefeng Chen and Somesh Jha are partially supported by the DARPA-GARD problem under agreement number 885000. Jayaram Raghuram was partially supported through the National Science Foundation's grants CNS-2112562 and CNS-2003129.

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

# Appendix

## A  PROOFS FOR SECTION 2.1

**Theorem A.1** (Restatement of Theorem 2.1). *If $\ell(t) = -t$, then the contrastive loss is equivalent to the PCA objective on $\phi_{z_R}$:*

$$\mathbb{E}\left[\ell\left(\phi(x)^\top[\phi(x^+) - \phi(x^-)]\right)\right] = -\mathbb{E}\left[\|\phi_{z_R} - \phi_0\|^2\right]. \tag{10}$$

*If additionally $\phi(x)$ is linear in $x$, then the contrastive loss is equivalent to the linear PCA objective on data from the distribution $p_{\bar{x}}$ of $\bar{x} = \mathbb{E}_{z_U}[x]$:*

$$\mathbb{E}\left[\ell\left(\phi(x)^\top[\phi(x^+) - \phi(x^-)]\right)\right] = -\mathbb{E}\left[\|\phi(\bar{x}) - \phi_0\|^2\right]. \tag{11}$$

*Proof.* We first present some preliminaries for the proof. Recall that in our hidden representation data model $x = g(z)$. The learned representation is $\phi(x) = \phi(g(z)) = \phi \circ g(z)$. For brevity, let us define $\phi(x) = \phi \circ g(z) := h(z)$. Also, the hidden representations corresponding to $(x, x^+, x^-)$ are given by $(z, z^+, z^-)$, where

$$z = [z_R\,;\,z_U], \quad z^+ = [z_R\,;\,z_U^+], \quad z^- = [z_R^-\,;\,z_U^-],$$

where $z_R$ and $z_R^-$ are sampled independently from the distribution $\mathcal{D}_R$; and $z_U, z_U^+$, and $z_U^-$ are sampled independently from the distribution $\mathcal{D}_U$. The expectation of an arbitrary function $f(z, z^+, z^-)$ can be simplified as follows:

$$
\begin{aligned}
\mathbb{E}_{(z,z^+,z^-)}\left[f(z, z^+, z^-)\right] &= \mathbb{E}_{(z_R,z_R^-,z_U,z_U^+,z_U^-)}\left[f(z, z^+, z^-)\right] \\
&= \mathbb{E}_{(z_R,z_R^-)}\left[\mathbb{E}_{(z_U,z_U^+,z_U^-)}\left[f(z, z^+, z^-)\mid z_R, z_R^-\right]\right].
\end{aligned}
$$

The second step follows the law of iterated expectations.

The negative expected contrastive loss is

$$-\mathbb{E}_{(x,x^+,x^-)}\left[\ell\left(\phi(x)^\top[\phi(x^+) - \phi(x^-)]\right)\right] \tag{12}$$

$$= -\mathbb{E}_{(z,z^+,z^-)}\left[\ell\left(\phi(g(z))^\top[\phi(g(z^+)) - \phi(g(z^-))]\right)\right] \tag{13}$$

$$= \mathbb{E}_{(z,z^+,z^-)}\left[h(z)^\top[h(z^+) - h(z^-)]\right] \tag{14}$$

$$= \mathbb{E}_{(z_R,z_R^-)}\left[\mathbb{E}\left[h(z)^\top[h(z^+) - h(z^-)]\mid z_R, z_R^-\right]\right] \tag{15}$$

$$= \mathbb{E}_{(z_R,z_R^-)}\left[\mathbb{E}\left[h(z)\mid z_R\right]^\top\left(\mathbb{E}\left[h(z^+)\mid z_R\right] - \mathbb{E}\left[h(z^-)\mid z_R^-\right]\right)\right] \tag{16}$$

$$= \mathbb{E}_{(z_R,z_R^-)}\left[\mathbb{E}\left[\phi(x)\mid z_R\right]^\top\left(\mathbb{E}\left[\phi(x^+)\mid z_R\right] - \mathbb{E}\left[\phi(x^-)\mid z_R^-\right]\right)\right] \tag{17}$$

$$= \mathbb{E}_{(z_R,z_R^-)}\left[\phi_{z_R}^\top\left(\phi_{z_R} - \phi_{z_R^-}\right)\right]. \tag{18}$$

The second equality follows from the choice of loss $\ell(t) = -t$, and the fourth equality follows from the fact that $z_U, z_U^+$, and $z_U^-$ are sampled independently from the distribution $\mathcal{D}_U$. Also, we have defined $\phi_{z_R} := \mathbb{E}[\phi(x)\mid z_R]$.

Denote the centered representation as $\bar{\phi}_{z_R} = \phi_{z_R} - \phi_0$. Then we have

$$-\mathbb{E}_{(x,x^+,x^-)}\left[\ell\left(\phi(x)^\top[\phi(x^+) - \phi(x^-)]\right)\right] \tag{19}$$

$$= \mathbb{E}_{(z_R,z_R^-)}\left[\phi_{z_R}^\top\left(\phi_{z_R} - \phi_{z_R^-}\right)\right] \tag{20}$$

$$= \mathbb{E}_{(z_R,z_R^-)}\left[(\bar{\phi}_{z_R} + \phi_0)^\top\left(\bar{\phi}_{z_R} + \phi_0 - \bar{\phi}_{z_R^-} - \phi_0\right)\right] \tag{21}$$

$$= \mathbb{E}_{(z_R,z_R^-)}\left[(\bar{\phi}_{z_R} + \phi_0)^\top\left(\bar{\phi}_{z_R} - \bar{\phi}_{z_R^-}\right)\right] \tag{22}$$

$$= \mathbb{E}_{(z_R,z_R^-)}\left[\bar{\phi}_{z_R}^\top\bar{\phi}_{z_R} - \bar{\phi}_{z_R}^\top\bar{\phi}_{z_R^-}\right] + \mathbb{E}_{(z_R,z_R^-)}\left[\phi_0^\top\left(\bar{\phi}_{z_R} - \bar{\phi}_{z_R^-}\right)\right]. \tag{23}$$

Since $\bar{\phi}_{z_R}$ and $\bar{\phi}_{z_R^-}$ are independent with mean 0, we have $\mathbb{E}_{(z_R, z_R^-)}[\bar{\phi}_{z_R}^\top \bar{\phi}_{z_R^-}] = 0$, $\mathbb{E}_{(z_R, z_R^-)}[\phi_0^\top \bar{\phi}_{z_R}] = 0$, and $\mathbb{E}_{(z_R, z_R^-)}[\phi_0^\top \bar{\phi}_{z_R^-}] = 0$. Therefore,

$$- \mathbb{E}_{(x, x^+, x^-)} \left[ \ell \left( \phi(x)^\top [\phi(x^+) - \phi(x^-)] \right) \right] \tag{24}$$

$$= \mathbb{E}_{z_R} \left[ \bar{\phi}_{z_R}^\top \bar{\phi}_{z_R} \right] \tag{25}$$

$$= \mathbb{E}_{z_R} \left[ \|\bar{\phi}_{z_R}\|^2 \right] \tag{26}$$

$$= \mathbb{E}_{z_R} \left[ \|\phi_{z_R} - \phi_0\|^2 \right], \tag{27}$$

which is the PCA objective on the mean representation $\phi_{z_R}$.

If additionally $\phi(x)$ is linear in $x$, then

$$- \mathbb{E}_{(x, x^+, x^-)} \left[ \ell \left( \phi(x)^\top [\phi(x^+) - \phi(x^-)] \right) \right] \tag{28}$$

$$= \mathbb{E}_{z_R} \left[ \|\phi_{z_R} - \phi_0\|^2 \right] \tag{29}$$

$$= \mathbb{E}_{\bar{x}} \left[ \|\phi(\bar{x}) - \phi(x_0)\|^2 \right] \tag{30}$$

which is the linear PCA objective on the data from the distribution of $\bar{x} = \mathbb{E}[x|z_R]$. $\qquad\square$

**Theorem A.2** (Restatement of Theorem 2.2). *Under Assumptions (A1)(A2)(A3):*

*(1) The optimal representation $\phi^*$ does not encode $z_U$: $\phi^* \circ g(z)$ is independent of $z_U$.*

*(2) For any invariant feature $i \in R$, there exists $B_i > 0$ such that as long as the representations' norm $B_r \geq B_i$, the optimal representation encodes $z_i$. Furthermore, if $z_R$ is discrete, then $B_i$ is monotonically decreasing in $\Pr[z_{R \setminus \{i\}} = z_{R \setminus \{i\}}^-, z_i \neq z_i^-]$, the probability that in $z_R$ and $z_R^-$, the $i$-th feature varies while the others remain the same.*

*Proof.* (1) Recall that

$$\phi_{z_R} = \mathbb{E}[\phi \circ g(z) \mid z_R], \quad \phi_0 = \mathbb{E}_z[\phi \circ g(z)] = \mathbb{E}_{z_R}[\phi_{z_R}]. \tag{31}$$

Then the contrastive loss at pre-training is:

$$\mathbb{E}_{(x, x^+, x^-)} \left[ \ell \left( \phi(x)^\top [\phi(x^+) - \phi(x^-)] \right) \right] \tag{32}$$

$$= \mathbb{E}_{(z, z^+, z^-)} \left[ \ell \left( (\phi \circ g(z))^\top (\phi \circ g(z^+) - \phi \circ g(z^-)) \right) \right] \tag{33}$$

$$= \mathbb{E}_{(z_R, z_R^-)} \left[ \mathbb{E} \left[ \ell \left( (\phi \circ g(z))^\top (\phi \circ g(z^+) - \phi \circ g(z^-)) \right) \mid z_R, z_R^- \right] \right] \tag{34}$$

$$\geq \mathbb{E}_{(z_R, z_R^-)} \left[ \ell \left( \mathbb{E} \left[ (\phi \circ g(z))^\top (\phi \circ g(z^+) - \phi \circ g(z^-)) \mid z_R, z_R^- \right] \right) \right] \tag{35}$$

$$= \mathbb{E}_{(z_R, z_R^-)} \left[ \ell \left( \mathbb{E}[\phi \circ g(z) \mid z_R]^\top \left( \mathbb{E}[\phi \circ g(z^+) \mid z_R] - \mathbb{E}[\phi \circ g(z^-) \mid z_R^-] \right) \right) \right] \tag{36}$$

$$= \mathbb{E}_{(z_R, z_R^-)} \left[ \ell \left( \phi_{z_R}^\top \phi_{z_R} - \phi_{z_R}^\top \phi_{z_R^-} \right) \right], \tag{37}$$

where the inequality comes from the convexity of $\ell(z)$ and Jensen's inequality applied to the inner expectation. The inequality becomes equality when the representation function $\phi$ is invariant to the spurious features $z_U$, i.e., with probability 1 over the distribution, $\phi \circ g(z) = \phi_{z_R}$. Therefore, the spurious features $z_U$ are not encoded in the optimal representation, proving the first part.

(2) First consider the case when $z$ has discrete values from a finite set. When the generative function $g(z)$ is not independent of $z_i$, we assume for contradiction that the optimal representation $\phi$ is independent of $z_i$. From (1), we know that it is independent of $z_U$. So there exists an $f$ such that $\phi \circ g(z) = f(z_{R \setminus \{i\}})$. Without loss of generality, suppose $U = \emptyset$, then $\phi \circ g(z) = f(z_{-i})$.

Since the generative function $g(z)$ is not independent of $z_i$, there exist $z$ and $z^-$, such that $z_{-i} = z_{-i}^-$, $z_i \neq z_i^-$, $g(z) \neq g(z^-)$, and $z, z^-$ have non-zero probabilities. So $\Pr[z_{-i} = z_{-i}^-, z_i \neq z_i^-] > 0$.

Now construct a new representation function $\bar{\phi} \in \mathbb{R}^{k+n}, n = |\mathcal{Z}|$ such that $\bar{\phi} \circ g(z) = h(z)$ as follows :

$$h(z) = \left[ \sqrt{1 - \alpha^2} f(z_{-i}), \quad \alpha \|f(z_{-i})\| \mathbf{I}_z \right] \tag{38}$$

where $\mathbf{I}_z$ is the one-hot encoding of the value $z$. Note that $\bar{\phi}$ still satisfies that norm bound since $\|\bar{\phi}(x)\| = \|h(z)\| = \|f(z_{-i})\|$. We next show that the contrastive loss of $\bar{\phi}$ can be smaller than that of $\phi$, leading to a contradiction and finishing the proof.

The contrastive loss of $\bar{\phi}$ (using the fact that $z^+ = z$ when $U = \emptyset$) is

$$\mathbb{E}_{(z,z^-)} \left[ \ell \left( h(z)^\top h(z) - h(z)^\top h(z^-) \right) \right] \tag{39}$$

$$= \mathbb{E}_{(z,z^-)} \left[ \ell \left( h(z)^\top h(z) - h(z)^\top h(z^-) \right) \mid z \neq z^- \right] \Pr[z \neq z^-] \; + \; \mathbb{E}_{z,z^-} \left[ \ell(0) \right] \Pr[z = z^-]. \tag{40}$$

We only need to consider the first term.

$$\mathbb{E}_{(z,z^-)} \left[ \ell \left( h(z)^\top h(z) - h(z)^\top h(z^-) \right) \mid z \neq z^- \right] \Pr[z \neq z^-] \tag{41}$$

$$= \mathbb{E}_{(z,z^-)} \big[ \underbrace{\ell \left( \|f(z_{-i})\|^2 - (1-\alpha^2) f(z_{-i})^\top f(z_{-i}^-) \right)}_{T_1} \mid z_{-i} \neq z_{-i}^- \big] \Pr[z_{-i} \neq z_{-i}^-] \tag{42}$$

$$+ \; \mathbb{E}_{(z,z^-)} \big[ \underbrace{\ell \left( \alpha^2 \|f(z_{-i})\|^2 \right)}_{T_2} \mid z_{-i} = z_{-i}^-, \, z_i \neq z_i^- \big] \Pr[z_{-i} = z_{-i}^-, \, z_i \neq z_i^-]. \tag{43}$$

When $\alpha = 0$, the above reduces to the corresponding terms for $\phi$, so we would like to show that there exists non-zero $\alpha$ that leads to smaller loss values.

Recall that $\ell(\cdot)$ is decreasing by property (A3). Let $\alpha = \sqrt{1/2}/B_r$, where $B_r = \|f(z_{-i})\|$. Then when switching from $\phi$ to $\bar{\phi}$, $T_2$ goes from $\ell(0)$ to $\ell(1/2)$, a constant reduction. For $T_1$, if $f(z_{-i})^\top f(z_{-i}^-)$ is positive, then $T_1$ decreases; if $f(z_{-i})^\top f(z_{-i}^-)$ is negative, then $T_1$ increases from $\ell(B_r^2 - f(z_{-i})^\top f(z_{-i}^-))$ to $\ell(B_r^2 - f(z_{-i})^\top f(z_{-i}^-) + \alpha^2 f(z_{-i})^\top f(z_{-i}^-))$. Note that $|\alpha^2 f(z_{-i})^\top f(z_{-i}^-)| \leq 1$ (by the Cauchy-Schwarz inequality); so the increase in $T_1$ diminishes when $B_r$ grows, by the property (A3) of $\ell$. Then when $B_r$ is large enough, the increase in $T_1$ is smaller than the decrease in $T_2$. So from $\phi$ to $\bar{\phi}$, the contrastive loss decreases, contradicting that $\phi$ is optimal. Finally, since the reduction in (43) is smaller when $\Pr[z_{-i} = z_{-i}^-, z_i \neq z_i^-]$ is smaller, then $B_i$ needs to be larger. So $B_i$ is monotonically decreasing in $\Pr[z_{-i} = z_{-i}^-, z_i \neq z_i^-]$.

Now consider the general case when $z$ may not be from a finite set. For any $\epsilon_0 > 0$, there exists a $\ell_2$ ball $\mathcal{B}$ of bounded radius such that the probability of $z$ outside the ball is at most $\epsilon_0$. Since $\phi \circ g$'s are regular by assumption, there exists a partition $\mathcal{Z} \cap \mathcal{B}$ into finitely many subsets such that in each subset and for each $\phi \circ g$, the function value varies by at most $\epsilon_0$. Construct a new distribution $\mathcal{D}'_z$ for $z$: select a representative point in each subset, and put a probability mass to it equal to that of the original distribution $\mathcal{D}_z$ in this subset, and normalize the probabilities over the subsets. The new distribution is over a finite set so the above argument holds. Furthermore, the difference in the $T_1$ term for $\mathcal{D}'_z$ and $\mathcal{D}_z$ can be made arbitrarily small by choosing sufficiently small $\epsilon_0$; similarly for $T_2$. Then the argument also holds for $\mathcal{D}_z$, which completes the proof for the general case. □

## A.1 Inductive Biases are Needed for Analyzing Prediction Success

We have analyzed what features are encoded in the representation. However, encoding the information does not equate to good prediction performance, in particular, with linear predictors. Recently, Saunshi et al. (2022b) demonstrated that existing analyses that ignore the inductive biases of the model and algorithm cannot adequately explain the prediction success, and provided examples where such analysis can lead to vacuous bounds. One may wonder if our hidden representation data model can provide inductive biases that avoid such vacuous bounds. Unfortunately, similar issues as in Saunshi et al. (2022b) remain.

To illustrate that inductive biases are still needed in our data model, consider the following simple example. Suppose $z_R \in \{-1, 1\}^2$ and can be recovered from $x$; the label $y$ is simply the first coordinate in $z_R$. Suppose the representation satisfies $\phi(x) \in \mathbb{R}^2, \|\phi(x)\| = 1$, and contrastive learning uses the logistic loss $\ell(z)$. Let $\phi(x)$ be such that $\phi \circ g(z) = h(z_R)$, and $h((-1,-1)) = (-1,0), h((-1,1)) = (1,0), h((1,-1)) = (0,-1), h((1,1)) = (0,1)$. It can be verified that this $\phi$ is optimal for the contrastive loss. However, on the representation $\phi$, the classification is an XOR-problem (Fig. 6), for which there is no non-trivial error bound for linear predictors. This contradicts the success of linear probing in practice.

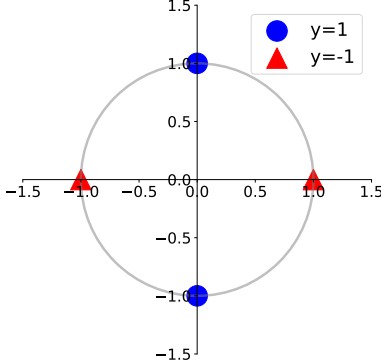

Figure 6: A two-dim example of XOR structure in the space of $\phi$.

Furthermore, some restrictions on the data distributions are also needed. Suppose *all* optimal representations are linearly separable with certain inductive biases on the representation function class. Suppose the label $y$ depends on $z_R$. Without restrictions on the labeling function, one can consider a random $y \in \{-1, +1\}$ over any $z_R$. Then for any linear predictor on any optimal representation, in expectation the error is $1/2$, so there is always a labeling function for which no non-trivial error can be achieved. Our analysis thus requires restrictions on the dependence of the label on $z_R$ (in particular, we will assume linear dependence).

## B  Proofs and More Analysis for Section 2.2

### B.1  Lemmas for a more general setting

We will prove the results in a more general setting, where the mixture can be uneven and the variances of different types of features can be different. The results in Section 2.2 then follow from these lemmas.

In the more general setting, the diverse pre-training data is a mixture of data from $T$ different tasks $\mathcal{D}_t$'s, while the target task is one of the tasks. In the mixture, the task $\mathcal{D}_t$ has weight $w_t > 0$ and $\sum_{t=1}^{T} w_t = 1$. All tasks share a public feature set $S$ of size $s$, and each task $\mathcal{D}_t$ additionally owns a private disjoint feature set $P_t$ of size $r - s$, i.e., $P_t \cap S = \emptyset$ for $t \in [T]$ and $P_{t_1} \cap P_{t_2} = \emptyset$ for $t_1 \neq t_2$. The invariant features for $\mathcal{D}_t$ are then $R_t = S \cup P_t$. All invariant features are $\cup_{t=1}^{T} R_t \subseteq R$, $k := |R|$, and spurious features are $U = [d] \setminus R$. In task $\mathcal{D}_t$, the positive pairs $(x, x^+)$ are generated as follows:

$$z_S \sim \mathcal{N}(0, \sigma_{S,t}^2 I), z_{P_t} \sim \mathcal{N}(0, \sigma_{R,t}^2 I), z_{R \setminus R_t} = 0, \tag{44}$$

$$z_U \sim \mathcal{N}(0, \sigma_{U,t}^2 I), z = [z_R; z_U], \quad x = g(z), \tag{45}$$

$$z_U^+ \sim \mathcal{N}(0, \sigma_{U,t}^2 I), z^+ = [z_R; z_U^+], \quad x^+ = g(z^+), \tag{46}$$

and $x^-$ is simply an i.i.d. copy from the same distribution as $x$. In practice, multiple independent negative examples are used, and thus we consider the following contrastive loss

$$\min_{\phi \in \Phi} \mathbb{E}_{(x, x^+)} \left[ \ell \left( \phi(x)^\top (\phi(x^+) - \mathbb{E}_{x^-} \phi(x^-)) \right) \right] \tag{47}$$

to pre-train a representation $\phi$. Then, when using $\phi$ for prediction in the target task $\mathcal{D}_t$, the predictor class should contain a predictor matching the ground-truth label, so consider the class:

$$\mathcal{F}_{\phi,t} = \{ f(z) = u_t^\top z : u_t \in \mathbb{R}^k, \|u_t\| \leq B_{\phi,t} \} \tag{48}$$

where $B_{\phi,t}$ is the minimum value such that there exists $u_t \in \mathcal{F}_{\phi,t}$ with $y = u_t^\top \phi(x)$ on $\mathcal{D}_t$.

Recall that we assume a linear data model and linear representation functions $\phi$:

- $x$ is linear in $z$: $x = g(z) = Mz$ where $M \in \mathbb{R}^{d \times d}$ is an orthonormal dictionary. The label in task $\mathcal{D}_t$ is linear in its invariant features $y = (u_t^*)^\top z_{R_t}$ for some $u_t^* \in \mathbb{R}^r$.

- The representations are linear functions with weights of bounded spectral/Frobenius norms:

$$\Phi = \{\phi(x) = Wx : W \in \mathbb{R}^{k \times d}, \|W\| \leq 1, \|W\|_F \leq \sqrt{r}\}.$$

Here the norm bounds are chosen to be the minimum values to allow recovering the invariant features in the target task, i.e., there exists $\phi \in \Phi$ such that $\phi(x) = [z_{R_t}; \mathbf{0}]$.

**Lemma B.1.** *Consider the above setting. Let $\alpha, \alpha_t (t \in [T])$ be the optimizer for*

$$\min_{\tilde{\alpha}, \tilde{\alpha}_1, \ldots, \tilde{\alpha}_T} \sum_{t=1}^T w_t \mathbb{E}\left[\ell\left(\tilde{\alpha}\sigma_{S,t}^2 Z + \tilde{\alpha}_t \sigma_{R,t}^2 Z_t\right)\right], \tag{49}$$

$$subject\ to \quad \tilde{\alpha}s + \sum_{t=1}^T \tilde{\alpha}_t(r - s) \leq r, \tag{50}$$

$$\tilde{\alpha}, \tilde{\alpha}_t \in [0, 1], \tag{51}$$

*where $Z \sim \chi_s^2$ and $Z_t \sim \chi_{r-s}^2$.*

*Then the optimal representation $\phi^*(x)$ the loss (47) in contrastive learning satisfies $\phi^*(x) = W^*x$ with any $W^*$ of the form:*

$$W^* = [QA^*, \mathbf{0}]M^{-1} \tag{52}$$

*where $Q \in \mathbb{R}^{k \times k}$ is any orthonormal matrices, $A^*$ is a $k \times k$ diagonal matrix with*

$$A_{jj}^* = \begin{cases} \sqrt{\alpha} & \text{if } j \in S, \\ \sqrt{\alpha_t} & \text{if } j \in P_t, \\ 0 & \text{otherwise}, \end{cases} \tag{53}$$

*and the matrix of zeros has size $k \times (d - k)$.*

*Proof.* For each $\mathcal{D}_t$,

$$\mathbb{E}_{(x,x^+)}\left[\ell\left(\phi(x)^\top[\phi(x^+) - \mathbb{E}_{x^-}\phi(x^-)]\right)\right] \tag{54}$$

$$= \mathbb{E}_{(z,z^+)}\left[\ell\left((WMz)^\top(WMz^+ - \mathbb{E}_{z^-}[WMz^-])\right)\right] \tag{55}$$

$$= \mathbb{E}_{(z,z^+)}\left[\ell\left(z^\top(M^\top W^\top WM)(z^+ - \mathbb{E}_{z^-}[z^-])\right)\right] \tag{56}$$

$$\geq \mathbb{E}_{z_R}\left[\ell\left((\mathbb{E}_{z_U}[z])^\top M^\top W^\top WM(\mathbb{E}_{z_U^+}[z^+] - \mathbb{E}_{z^-}[z^-])\right)\right] \tag{57}$$

$$= \mathbb{E}_{z_R}\left[\ell\left([z_R; \mathbf{0}]^\top M^\top W^\top WM([z_R; \mathbf{0}] - 0)\right)\right] \tag{58}$$

$$= \mathbb{E}_{z_R}\left[\ell\left(\|WM[z_R; \mathbf{0}]\|^2\right)\right] \tag{59}$$

where the inequality comes from the convexity of $\ell(t)$ and Jensen's inequality. Similar to Theorem 2.2, the equality holds if and only if $WMz$ does not depend on $z_U$ and $WMz^+$ does not depend on $z_U^+$, so the optimal solution should satisfy this condition.

Let $WM = [A_R, A_U]$ where $A_R \in \mathbb{R}^{k \times k}$, $A_U \in \mathbb{R}^{k \times (d-k)}$. By rotational invariance of $z_S$, and $z_{P_t}$, without loss of generality, we can assume $A_R = QA$ where $A$ is a diagonal matrix with diagonal entries $a_{jj}$'s and $Q$ is any orthonormal matrix. Furthermore, $A_U = 0$ in the optimal solution since it does not affect the loss but only decreases the norm bound on $A_R$. So on data from the task $\mathcal{D}_t$,

$$\mathbb{E}_{\mathcal{D}_t}\left[\ell\left(\|WM[z_R; \mathbf{0}]\|^2\right)\right] = \mathbb{E}_{z_{R_t}}\left[\ell\left(\sum_{j \in R_t} a_{jj}^2 z_j^2\right)\right]. \tag{60}$$

Then on the mixture,

$$\mathbb{E}_{(x,x^+)} \left[ \ell \left( \phi(x)^\top [\phi(x^+) - \mathbb{E}_{x^-} \phi(x^-)] \right) \right] \tag{61}$$

$$\geq \sum_{t=1}^{T} w_t \mathbb{E}_{\{z_j\}} \left[ \ell \left( \sum_{j \in R_t} a_{jj}^2 z_j^2 \right) \right] \tag{62}$$

$$= \sum_{t=1}^{T} w_t \mathbb{E}_{\{\tilde{z}_j \sim \mathcal{N}(0,1)\}} \left[ \ell \left( \sum_{j \in S} a_{jj}^2 \sigma_{S,t}^2 \tilde{z}_j^2 + \sum_{j \in P_t} a_{jj}^2 \sigma_{R,t}^2 \tilde{z}_j^2 \right) \right] \tag{63}$$

$$:= g(\{a_{jj}\}), \tag{64}$$

where each $\tilde{z}_j$ is a random variable drawn from standard Gaussian.

Now consider the minimum of the function $g(\{a_{jj}\})$ on the right hand side, under the constraints that $|a_{jj}| \leq 1$ and $\sum_j a_{jj}^2 \leq r$. Before finishing the proof of Lemma B.1, we have the following claim for this optimization.

**Claim 1.** *There exist* $\alpha, \alpha_t$ *satisfying* $0 \leq \alpha, \alpha_t \leq 1$ *and* $\alpha s + \sum_{t=1}^{T} \alpha_t (r - s) = \sum_j a_{jj}^2 \leq r$, *such that the minimum of the above optimization (64) is achieved when* $a_{jj}^2 = \alpha$ *for any* $j \in S$, *and* $a_{jj}^2 = \alpha_t$ *for any* $j \in P_t$ *and* $t \in [T]$.

*Proof.* We need to prove that to achieve the minimum,

(1) $a_{\ell\ell}^2 = a_{\ell'\ell'}^2$ for any $\ell \neq \ell' \in S$;

(2) $a_{\ell\ell}^2 = a_{\ell'\ell'}^2$ for any $\ell \neq \ell' \in P_t$ and any $t \in [T]$;

For (1): By symmetry of $z_j$'s and the convexity of $\ell(\cdot)$, for any $t \in [T]$,

$$\mathbb{E} \left[ \ell \left( \sum_{j \in R_t} a_{jj}^2 z_j^2 \right) \right] \tag{65}$$

$$= \frac{1}{2} \mathbb{E} \left[ \ell \left( \sum_{j \in S, j \neq \ell, j \neq \ell'} a_{jj}^2 z_j^2 + a_{\ell\ell}^2 z_\ell^2 + a_{\ell'\ell'}^2 z_{\ell'}^2 + \sum_{j \in P_t} a_{jj}^2 z_j^2 \right) \right] \tag{66}$$

$$+ \frac{1}{2} \mathbb{E} \left[ \ell \left( \sum_{j \in S, j \neq \ell, j \neq \ell'} a_{jj}^2 z_j^2 + a_{\ell\ell}^2 z_{\ell'}^2 + a_{\ell'\ell'}^2 z_\ell^2 + \sum_{j \in P_t} a_{jj}^2 z_j^2 \right) \right] \tag{67}$$

$$\geq \mathbb{E} \left[ \ell \left( \sum_{j \in S, j \neq \ell, j \neq \ell'} a_{jj}^2 z_j^2 + \frac{a_{\ell\ell}^2 + a_{\ell'\ell'}^2}{2} z_{\ell'}^2 + \frac{a_{\ell\ell}^2 + a_{\ell'\ell'}^2}{2} z_\ell^2 + \sum_{j \in P_t} a_{jj}^2 z_j^2 \right) \right]. \tag{68}$$

Then

$$g(\{a_{jj}\}) \geq \sum_{t=1}^{T} w_t \mathbb{E} \left[ \ell \left( \sum_{j \in S, j \neq \ell, j \neq \ell'} a_{jj}^2 z_j^2 + \frac{a_{\ell\ell}^2 + a_{\ell'\ell'}^2}{2} z_{\ell'}^2 + \frac{a_{\ell\ell}^2 + a_{\ell'\ell'}^2}{2} z_\ell^2 + \sum_{j \in P_t} a_{jj}^2 z_j^2 \right) \right]. \tag{69}$$

Therefore, the minimum is achieved when $a_{\ell\ell}^2 = a_{\ell'\ell'}^2$.

A similar argument as above proves statement (2). $\square$

These statements mean that, for any $t \in [T]$, the minimum is achieved when $a_{jj}^2 = \alpha$ for $j \in S$, and $a_{jj}^2 = \alpha_t$ for $j \in P_t$, for some values $\alpha, \alpha_t \geq 0$. Let $Z = \sum_{j \in S} \tilde{z}_j^2, Z_t = \sum_{j \in P_t} \tilde{z}_j^2$. Then $Z \sim \chi_s^2$

and $Z_t \sim \chi^2_{r-s}$, and we have:

$$g(\{a_{jj}\}) = \sum_{t=1}^T w_t \mathbb{E}\left[\ell\left(\sum_{j \in S} \alpha \sigma^2_{S,t} \tilde{z}^2_j + \sum_{j \in P_t} \alpha_t \sigma^2_{R,t} \tilde{z}^2_j\right)\right] \tag{70}$$

$$= \sum_{t=1}^T w_t \mathbb{E}\left[\ell\left(\alpha \sigma^2_{S,t} Z + \alpha_t \sigma^2_{R,t} Z_t\right)\right]. \tag{71}$$

Given the constraint $\alpha s + \sum_{t=1}^T \alpha_t(r - s) = \sum_j a^2_{jj} \leq r, 0 \leq \alpha, \alpha_t \leq 1$, we complete the proof of Lemma B.1. $\qquad\square$

Given this result we can now analyze the generalization error when predicting on the target task $\mathcal{D}_t$.

**Lemma B.2.** *Consider any $t \in [T]$. Let $v_{t,1} = \sum_{j \in S}(u^*_t)^2_j$ and $v_{t,2} = \sum_{j \in P_t}(u^*_t)^2_j$. Suppose in $\phi^*$ (calculated in Lemma B.1), $\alpha, \alpha_t > 0$. Suppose the prediction loss $\ell_c$ is $L$-Lipschitz.*

*Then the Empirical Risk Minimizer $\hat{u}_t \in \mathcal{F}_{\phi^*,t}$ on $\phi^*$ using $m$ labeled data points from $\mathcal{D}_t$ has risk*

$$\mathbb{E}_{(x,y)\sim\mathcal{D}_t}[\ell_c(\hat{u}_t^\top \phi^*(x), y)] \leq 8\sqrt{\frac{2\ln(4/\delta)}{m}}$$

$$+ 4L\sqrt{\frac{1}{m}\left(\frac{v_{t,1}}{\alpha} + \frac{v_{t,2}}{\alpha_t}\right)}\left(\sqrt{s\alpha\sigma^2_{S,t} + (r-s)\alpha_t\sigma^2_{R,t}} + O\left(\sqrt{\frac{\max\{\alpha\sigma^2_{S,t}, \alpha_t\sigma^2_{R,t}\}^2 r}{s\alpha\sigma^2_{S,t} + (r-s)\alpha_t\sigma^2_{R,t}}}\right)\right).$$

*Proof.* For any $t \in [T]$, we only need to bound the Rademacher complexity $\mathcal{R}_m(\mathcal{F}_{\phi^*,t})$ of $\mathcal{F}_{\phi^*,t}$; the statement then follows from standard generalization bounds,

$$\mathbb{E}_{(x,y)\sim\mathcal{D}_t}[\ell_c(\hat{u}_t^\top \phi^*(x), y)] \leq 4L\mathcal{R}_m(\mathcal{F}_{\phi^*,t}) + 8\sqrt{\frac{2\ln(4/\delta)}{m}}.$$

Given the representation $\phi^*$ in Lemma B.1, to ensure there exists a predictor in $\mathcal{F}_{\phi^*,t}$ matching the ground-truth label, $f(\phi^*(x)) = u_t^\top \phi^*(x) = y = (u_t^*)^\top z_{R_t}$, predictor $u_t$ should satisfy

$$\mathbb{E}_{\mathcal{D}_t}[(\hat{y} - y)^2] = 0 \Leftrightarrow \forall z_{R_t}, \quad u_t^\top[QA^*, \mathbf{0}]M^{-1}M[z_{R_t}; \mathbf{0}; z_U] = u_t^{*\top} z_{R_t} \tag{72}$$

$$\Leftrightarrow \forall z_{R_t}, \quad u_t^\top QA^*[z_{R_t}; \mathbf{0}] = u_t^{*\top} z_{R_t} \tag{73}$$

$$\overset{(*)}{\Leftrightarrow} A^*_{1:r,1:r}(Q^\top)_{1:r,1:k} u_t = u_t^* \tag{74}$$

$$\Leftrightarrow \forall v \in \mathbb{R}^r, \quad u_t = Q_{1:k,1:r}(A^*_{1:r,1:r})^{-1} u_t^* + Q_{1:k,r+1:k} v. \tag{75}$$

The $(*)$ is from non-zero variance for $z_{R_t}$. $u_t = Q_{1:k,1:r}(A^*_{1:r,1:r})^{-1} u_t^*$ is the least-norm optimal solution, so we have $B_{\phi^*,t} = \|Q_{1:k,1:r}(A^*_{1:r,1:r})^{-1} u_t^*\| = \sqrt{\frac{v_{t,1}}{\alpha} + \frac{v_{t,2}}{\alpha_t}}$. So the predictor class should be

$$\mathcal{F}_{\phi^*,t} = \left\{f(\phi^*) = u_t^\top \phi^* : u_t \in \mathbb{R}^k, \|u_t\| \leq B_{\phi^*,t} = \sqrt{\frac{v_{t,1}}{\alpha} + \frac{v_{t,2}}{\alpha_t}}\right\}. \tag{76}$$

The empirical Rademacher complexity and Rademacher complexity of $\mathcal{F}_{\phi^*,t}$ with $m$ samples are

$$\hat{\mathcal{R}}_m(\mathcal{F}_{\phi^*,t}) = \frac{1}{m}\mathbb{E}_\sigma\left[\sup_{f_{u,\phi} \in \mathcal{F}_{\phi^*,t}} \sum_{i=1}^m \sigma_i f_{u,\phi}(x^{(i)})\right] \tag{77}$$

$$= \frac{1}{m}\mathbb{E}_\sigma\left[\sup_{\|u\| \leq B_{\phi^*,t}} \sum_{i=1}^m \sigma_i u_t^\top QA^*[z_{R_t}^{(i)}; \mathbf{0}]\right] \tag{78}$$

$$= \frac{1}{m}\mathbb{E}_\sigma\left[\sup_{\|u\| \leq B_{\phi^*,t}} u_t^\top \sum_{i=1}^m \sigma_i Q_{1:k,1:r} A^*_{1:r,1:r} z_{R_t}^{(i)}\right] \tag{79}$$

$$= \frac{B_{\phi^*,t}}{m}\mathbb{E}_\sigma\left[\left\|\sum_{i=1}^m \sigma_i Q_{1:k,1:r} A^*_{1:r,1:r} z_{R_t}^{(i)}\right\|\right], \tag{80}$$

$$\mathcal{R}_m(\mathcal{F}_{\phi^*,t}) = \mathbb{E}_{z_R, z_U} \left[ \hat{\mathcal{R}}_m(\mathcal{F}_{\phi^*,t}) \right] \tag{81}$$

$$= \frac{B_{\phi^*,t}}{m} \mathbb{E}_{z_{R_t}^{(i)}} \left[ \mathbb{E}_\sigma \left[ \left\| \sum_{i=1}^m \sigma_i Q_{1:k,1:r} A^*_{1:r,1:r} z_{R_t}^{(i)} \right\| \right] \right] \tag{82}$$

$$= \frac{B_{\phi^*,t}}{m} \mathbb{E}_{z_{R_t}^{(i)}} \left[ \left\| A^*_{1:r,1:r} \sum_{i=1}^m z_{R_t}^{(i)} \right\| \right]. \tag{83}$$

For any $t \in [T]$, define $X_t := A^*_{1:r,1:r} \sum_{i=1}^m z_{R_t}^{(i)}$. Note that for $j \in S$, $X_{t,j} = \alpha \sum_{i=1}^m z_j^{(i)}$ is a Gaussian of mean zero and variance $\mathbb{E}[X_{t,j}^2] = \alpha \mathbb{E}\left[ \left( \sum_{i=1}^m z_j^{(i)} \right)^2 \right] = \alpha \mathbb{E}\left[ \sum_{i=1}^m \left( z_j^{(i)} \right)^2 \right] = m\alpha\sigma_{S,t}^2$. Similarly, for $j \in P_t$, $X_{t,j} = \alpha_t \sum_{i=1}^m z_j^{(i)}$ is a Gaussian of mean zero and variance $\mathbb{E}[X_{t,j}^2] = m\alpha_t \sigma_{R,t}^2$. Since $X_{t,j}$ is sub-gaussian, $X_{t,j}^2 - m\alpha\sigma_{S,t}^2$ for $j \in S$ and $X_{t,j}^2 - m\alpha_t \sigma_{R,t}^2$ for $j \in P_t$ are sub-exponential and more precisely

$$\|X_{t,j}^2 - m\alpha\sigma_{S,t}^2\|_{\psi_1} \leq C_1 \|X_{t,j}^2\|_{\psi_1} = C_1 \|X_{t,j}\|_{\psi_2}^2 \leq C_2 m\alpha\sigma_{S,t}^2, \quad j \in S, \tag{84}$$

$$\|X_{t,j}^2 - m\alpha_t\sigma_{R,t}^2\|_{\psi_1} \leq C_1 \|X_{t,j}^2\|_{\psi_1} = C_1 \|X_{t,j}\|_{\psi_2}^2 \leq C_2 m\alpha_t\sigma_{R,t}^2, \quad j \in P_t, \tag{85}$$

where $C_1, C_2$ are absolute constants and $C_2 > 1$. Let $K = \max(C_2 m\alpha\sigma_{S,t}^2, C_2 m\alpha_t\sigma_{R,t}^2) = C_2 m \max\{\alpha\sigma_{S,t}^2, \alpha_t\sigma_{R,t}^2\}$ and $\mu := m(s\alpha\sigma_{S,t}^2 + (r-s)\alpha_t\sigma_{R,t}^2)$. By Bernstein's inequality, we have for every $\gamma \geq 0$ that

$$\mathbb{P}\left\{ \left| \frac{1}{r}(\|X_t\|^2 - \mu) \right| \geq \gamma \right\} \leq 2\exp\left[ -c\min\left( \frac{\gamma^2}{K^2}, \frac{\gamma}{K} \right) r \right] \tag{86}$$

$$\Rightarrow \mathbb{P}\left\{ \left| \frac{\|X_t\|^2}{\mu} - 1 \right| \geq \frac{r\gamma}{\mu} \right\} \tag{87}$$

$$\leq 2\exp\left[ -\frac{c}{C_2^2} \min\left( \frac{\gamma^2}{m^2 \max\{\alpha\sigma_{S,t}^2, \alpha_t\sigma_{R,t}^2\}^2}, \frac{\gamma}{m\max\{\alpha\sigma_{S,t}^2, \alpha_t\sigma_{R,t}^2\}} \right) r \right], \tag{88}$$

where $c$ is an absolute constant. For all numbers $z \geq 0$, we have $|z - 1| \geq \delta \Rightarrow |z^2 - 1| \geq \max(\delta, \delta^2)$. Thus, for any $\delta \geq 0$, we have

$$\mathbb{P}\left\{ \left| \frac{\|X_t\|}{\sqrt{\mu}} - 1 \right| \geq \delta \right\} \tag{89}$$

$$\leq \mathbb{P}\left\{ \left| \frac{\|X_t\|_2^2}{\mu} - 1 \right| \geq \max(\delta, \delta^2) \right\} \tag{90}$$

$$\leq 2\exp\left[ -\frac{c}{C_2^2} \min\left( \left( \frac{\mu\max(\delta, \delta^2)}{m\max\{\alpha\sigma_{S,t}^2, \alpha_t\sigma_{R,t}^2\}r} \right)^2, \frac{\mu\max(\delta, \delta^2)}{m\max\{\alpha\sigma_{S,t}^2, \alpha_t\sigma_{R,t}^2\}r} \right) r \right] \tag{91}$$

$$\leq 2\exp\left[ -\frac{c}{C_2^2} \left( \frac{\mu}{m\max\{\alpha\sigma_{S,t}^2, \alpha_t\sigma_{R,t}^2\}r} \right)^2 \min\left( \left(\max(\delta, \delta^2)\right)^2, \max(\delta, \delta^2) \right) r \right] \tag{92}$$

$$= 2\exp\left[ -\frac{c}{C_2^2} \frac{\mu^2}{m^2 \max\{\alpha\sigma_{S,t}^2, \alpha_t\sigma_{R,t}^2\}^2 r} \delta^2 \right], \tag{93}$$

where the last inequality is from $\mu = m(s\alpha\sigma_{S,t}^2 + (r-s)\alpha_t\sigma_{R,t}^2) \leq m\max\{\alpha\sigma_{S,t}^2, \alpha_t\sigma_{R,t}^2\}r$. Changing variables to $\theta = \delta\sqrt{\mu}$, we obtain the desired sub-gaussian tail

$$\mathbb{P}\left\{ |\|X_t\| - \sqrt{\mu}| \geq \theta \right\} \leq 2\exp\left[ -\frac{c}{C_2^2} \frac{\mu}{m^2 \max\{\alpha\sigma_{S,t}^2, \alpha_t\sigma_{R,t}^2\}^2 r} \theta^2 \right]. \tag{94}$$

By generalization of integral identity, we have

$$|\mathbb{E}\left[\|X_t\|\right] - \sqrt{\mu}| = \left|\int_0^\infty \mathbb{P}\{\|X_t\| - \sqrt{\mu} > \theta\}d\theta - \int_{-\infty}^0 \mathbb{P}\{\|X_t\| - \sqrt{\mu} < \theta\}d\theta\right| \qquad (95)$$

$$\leq 2\int_0^\infty \mathbb{P}\{|\|X_t\| - \sqrt{\mu}| > \theta\}d\theta \qquad (96)$$

$$\leq 4\int_0^\infty \exp\left[-\frac{c}{C_2^2}\frac{\mu}{m^2\max\{\alpha\sigma_{S,t}^2, \alpha_t\sigma_{R,t}^2\}^2 r}\theta^2\right]d\theta \qquad (97)$$

$$\leq C_3\frac{m\max\{\alpha\sigma_{S,t}^2, \alpha_t\sigma_{R,t}^2\}\sqrt{r}}{\sqrt{\mu}}, \qquad (98)$$

where $C_3$ is an absolute constant. Thus, we have

$$\left|\mathcal{R}_m(\mathcal{F}_{\phi^*,t}) - \sqrt{\frac{1}{m}\left(\frac{v_{t,1}}{\alpha} + \frac{v_{t,2}}{\alpha_t}\right)(s\alpha\sigma_{S,t}^2 + (r-s)\alpha_t\sigma_{R,t}^2)}\right| \qquad (99)$$

$$= \frac{B_{\phi^*,t}}{m}|\mathbb{E}\left[\|X_t\|\right] - \sqrt{\mu}| \qquad (100)$$

$$\leq O\left(\sqrt{\frac{1}{m}\left(\frac{v_{t,1}}{\alpha} + \frac{v_{t,2}}{\alpha_t}\right)\frac{\max\{\alpha\sigma_{S,t}^2, \alpha_t\sigma_{R,t}^2\}^2 r}{s\alpha\sigma_{S,t}^2 + (r-s)\alpha_t\sigma_{R,t}^2}}\right). \qquad (101)$$

$$\square$$

## B.2 PROOFS OF PROPOSITION 2.3 AND PROPOSITION 2.4

Given the lemmas for the general case, we are now ready to prove the results in Proposition 2.3 and Proposition 2.4.

**Proposition B.3** (Restatement of Proposition 2.3). *Suppose* $\sigma_{S,t} = \sigma_{R,t} = \sigma_{U,t} = 1$ *for any* $t \in [T]$. *The representation* $\phi^*$ *obtained on an even mixture of data from all the tasks* $\{\mathcal{D}_t : 1 \leq t \leq T\}$ *satisfies* $\phi^* \circ g(z) = Q\left(\sum_{j\in S}\sqrt{\alpha}z_je_j + \sum_{j\in R\backslash S}\sqrt{\beta}z_je_j\right)$ *for some* $\alpha \in [0, 1]$, $\beta = \min\left(1, \frac{r-\alpha s}{T(r-s)}\right)$, *where* $e_j$'s *are the basis vectors and* $Q$ *is any orthonormal matrix.*
*The Empirical Risk Minimizer* $\hat{u} \in \mathcal{F}_{\phi^*,t}$ *on* $\phi^*$ *using* $m$ *labeled data points from* $\mathcal{D}_t$ *has risk*

$$\mathbb{E}_{(x,y)\sim\mathcal{D}_t}[\ell_c(\hat{u}^\top\phi^*(x), y)]$$
$$\leq 4L\sqrt{\frac{1}{m}\left(\frac{v_{t,1}}{\alpha} + \frac{v_{t,2}}{\beta}\right)}\left(\sqrt{s\alpha + (r-s)\beta} + O\left(\sqrt{\frac{r}{s\alpha + (r-s)\beta}}\right)\right) + 8\sqrt{\frac{2\ln(4/\delta)}{m}}.$$

*Proof.* This follows from Lemma B.1, and considering the optimal $\alpha, \alpha_t$ for the following:

$$g(\{\alpha, \alpha_t\}) = \sum_{t=1}^T w_t\mathbb{E}\left[\ell\left(\alpha\sigma_{S,t}^2 Z + \alpha_t\sigma_{R,t}^2 Z_t\right)\right] \qquad (102)$$

$$= \frac{1}{T}\sum_{t=1}^T \mathbb{E}\left[\ell\left(\alpha Z + \alpha_t Z_1\right)\right] \qquad (103)$$

$$\geq \mathbb{E}\left[\ell\left(\alpha Z + Z_1\sum_{t=1}^T\frac{1}{T}\alpha_t\right)\right]. \qquad (104)$$

The second equation is from that $Z_t$'s follow the same distribution by the symmetry of $\tilde{z}_j$'s. The inequality comes from the convexity of $\ell(t)$ and Jensen's inequality. So the minimum is achieved when $\alpha_t := \beta$ for any $t \in [T]$, leading to

$$g(\{\alpha, \alpha_t\}) = \mathbb{E}\left[\ell\left(\alpha Z + \beta Z_1\right)\right] \qquad (105)$$

subject to the constraints $\alpha s + T\beta(r-s) \le r$, $0 \le \alpha, \beta \le 1$. Then we get $\phi^* \circ g(z) = W^* M z = Q\left(\sum_{j \in S} \sqrt{\alpha} z_j e_j + \sum_{j \in R \setminus S} \sqrt{\beta} z_j e_j\right)$ for some $\alpha \in [0,1]$, $\beta = \min\left(1, \frac{r-\alpha s}{T(r-s)}\right)$, where $e_j$'s are the basis vectors and $Q$ is any orthonormal matrix.

Finally, the generalization bound follows from Lemma B.2, and that

$$O\left(\sqrt{\frac{\max\{\alpha,\beta\}^2 r}{s\alpha + (r-s)\beta}}\right) = O\left(\sqrt{\frac{r}{s\alpha + (r-s)\beta}}\right). \tag{106}$$

This completes the proof. □

**Proposition B.4** (Restatement of Proposition 2.4). *Suppose $\sigma_{S,t} = \sigma_{R,t} = \sigma_{U,t} = 1$. The representation $\phi_t^*$ obtained on data from $\mathcal{D}_t$ satisfies $\phi_t^* \circ g(z) = Q\left(\sum_{j \in R_t} z_j e_j\right)$ where $e_j$'s are the basis vectors and $Q$ is any orthonormal matrix.*
*The Empirical Risk Minimizer $\hat{u} \in \mathcal{F}_{\phi_t^*, t}$ on $\phi_t^*$ using $m$ labeled data points from $\mathcal{D}_t$ has risk*

$$\mathbb{E}_{(x,y) \sim \mathcal{D}_t}[\ell_c(\hat{u}^\top \phi_t^*(x), y)] \le 4L\sqrt{\frac{r}{m}}\|u_t^*\| + 8\sqrt{\frac{2\ln(4/\delta)}{m}}.$$

*While on task $\mathcal{D}_i (i \ne t)$, any linear predictor on $\phi_t^*$ has error at least $\min_u \mathbb{E}_{\mathcal{D}_i}[\ell_c(u^\top z_S, y)]$.*

*Proof.* Following Lemma B.1 (with $r = s$), we get $\phi_t^* \circ g(z) = Q\left(\sum_{j \in R_t} z_j e_j\right)$, where $e_j$'s are the basis vectors and $Q$ is any orthonormal matrix. Following the same argument as in the proof of Lemma B.2, we get

$$\mathcal{R}_m(\mathcal{F}_{\phi^*, t}) = \frac{\|u_t^*\|}{m} \mathbb{E}_{z_{R_t}^{(i)}}\left[\left\|\sum_{i=1}^m z_{R_t}^{(i)}\right\|\right] \tag{107}$$

$$\le \sqrt{\frac{r}{m}}\|u_t^*\|, \tag{108}$$

where the last inequality comes from the property of chi-squared distribution expectation. □

### B.3 IMPLICATION FOR THE TRADE-OFF

The propositions then imply the trade-off between universality and label efficiency. Below we formalize the example discussed in Section 2.2.

**Proposition B.5** (A specific version of Proposition 2.3). *Suppose $\sigma_{S,t} = \sigma_{R,t} = \sigma_{U,t} = 1$ for any $t \in [T]$ and $r = 2s$. The representation $\phi^*$ obtained on an even mixture of data from all the tasks $\{\mathcal{D}_t : 1 \le t \le T\}$ satisfies $\phi^* \circ g(z) = Q\left(\sum_{j \in S} z_j e_j + \sum_{j \in R \setminus S} \sqrt{\frac{1}{T}} z_j e_j\right)$, where $e_j$'s are the basis vectors and $Q$ is any orthonormal matrix.*
*The Empirical Risk Minimizer $\hat{u} \in \mathcal{F}_{\phi^*, t}$ on $\phi^*$ using $m$ labeled data points from $\mathcal{D}_t$ has risk*

$$\mathbb{E}_{(x,y) \sim \mathcal{D}_t}[\ell_c(\hat{u}^\top \phi^*(x), y)] \le 4L\sqrt{\frac{1}{m}(v_{t,1} + Tv_{t,2})}\left(\sqrt{r\left(\frac{T+1}{2T}\right)} + O(1)\right) + 8\sqrt{\frac{2\ln(4/\delta)}{m}}.$$

*Proof.* This follows from Proposition 2.3, and noting that when $r = 2s$, $\alpha = 1$ and $\beta = 1/T$ are the optimal for:

$$g(\{\alpha, \beta\}) = \mathbb{E}\left[\ell(\alpha Z + \beta Z_1)\right] \tag{109}$$

$$= \mathbb{E}\left[\ell\left(\alpha Z + \frac{r - \alpha s}{T(r-s)} Z_1\right)\right] \tag{110}$$

$$= \mathbb{E}\left[\ell\left(\alpha Z + \frac{2 - \alpha}{T} Z_1\right)\right] \tag{111}$$

subject to the constraints $\alpha s + T\beta(r-s) \le r$, $0 \le \alpha, \beta \le 1$. To see this, note that $Z \sim \chi_s^2$ and $Z_1 \sim \chi_{r-s}^2 = \chi_s^2$ follow the same distribution, so $\alpha Z + \frac{2-\alpha}{T} Z_1$ for $\alpha = 1$ will stochastically dominate its value for other $\alpha \in [0,1)$. The optimal is then achieved when $\alpha = 1$ and $\beta = \frac{2-\alpha}{T} = \frac{1}{T}$. □

## B.4 Improving the Trade-off by Contrastive Regularization

The above analysis shows that contrastive learning a representation on unlabeled data from the target task can help in prediction on this target task. This suggests that given a representation $\phi^*$ pre-trained on diverse data, one can fine-tune it by contrastive learning on some unlabeled data from the target task to get a representation that can lead to better prediction on the target task. In the following, we will formally show that this is indeed the case for the illustrative example in Section 2.2.

Recall that in this example, $\sigma_{S,t} = \sigma_{R,t} = \sigma_{U,t} = 1$, $r = 2s$, and $v_{t,1} = v_{t,2}$. The representation $\phi^*$ obtained on an even mixture of data from all the tasks $\{\mathcal{D}_t : 1 \le t \le T\}$ satisfies $\phi^* \circ g(z) = Q\left(\sum_{j \in S} \sqrt{\alpha} z_j e_j + \sum_{j \in R \setminus S} \sqrt{\beta} z_j e_j\right)$, where $e_j$'s are the basis vectors and $Q$ is any orthonormal matrix, and $\alpha = 1, \beta = \frac{1}{T}$.

Now, suppose we are given unlabeled data from $\mathcal{D}_t$, and we use them to fine-tune $\phi^*(x) = W^* x$ by contrastive learning on these unlabeled data. That is, we find $W$ near $W^*$ to minimize the contrastive loss on the unlabeled data from $\mathcal{D}_t$:

$$\min_{\phi(x) = Wx} \mathbb{E}\left[\ell\left(\phi(x)^\top (\phi(x^+) - \mathbb{E}_{x^-} \phi(x^-))\right)\right] \tag{112}$$

$$\text{subject to} \quad \|W - W^*\|_F \le \gamma, \|W\| \le 1. \tag{113}$$

for some small $\gamma > 0$.

**Proposition B.6.** *For (112), $\phi^*_{CR,t}$ satisfying the following on $x$ from $\mathcal{D}_t$ is an optimal representation:*

$$\phi^*_{CR,t} \circ g(z) = Q\left(\sum_{j \in S} \sqrt{\alpha} z_j e_j + \sum_{j \in P_t} \sqrt{\beta} z_j e_j\right)$$

*where $\sqrt{\alpha} = 1, \sqrt{\beta} = \min\left(1, \sqrt{\frac{1}{T}} + \frac{\gamma}{\sqrt{s}}\right)$.*

*Proof.* Following the argument in Lemma B.1, we still have that $\phi^*_{CR,t}(x) = Wx$ where $W = Q_2[A_2; \mathbf{0}]M^{-1}$ for any orthonormal matrix $Q_2$ and some diagonal matrix $A_2 = \text{diagonal}(a_{jj})$, with $a_{jj} = \sqrt{\alpha}$ for $j \in S$ and $a_{jj} = \sqrt{\beta}$ for $j \in P_t$ for some $\alpha, \beta \in [0,1]$. And the contrastive loss is:

$$\mathbb{E}\left[\ell\left(\phi(x)^\top (\phi(x^+) - \mathbb{E}_{x^-} \phi(x^-))\right)\right] = \mathbb{E}\left[\ell\left(\sum_{j \in R_t} a_{jj}^2 z_j^2\right)\right] \tag{114}$$

$$= \mathbb{E}\left[\ell\left(\alpha \sum_{j \in S} z_j^2 + \beta \sum_{j \in P_t} z_j^2\right)\right]. \tag{115}$$

Recall that $\phi^*(x) = W^* x$ with $W = Q[A; \mathbf{0}]M^{-1}$ for any orthonormal matrix $Q$ and some diagonal matrix $A$, with $A_{jj} = 1$ for $j \in S$ and $A_{jj} = \sqrt{1/T}$ for $j \in R_i \setminus S$ for any $i \in [T]$. Then

$$\|W - W^*\|_F = \|Q_2[A_2; \mathbf{0}]M^{-1} - Q[A; \mathbf{0}]M^{-1}\|_F \tag{116}$$

$$= \|Q_2 A_2 - QA\|_F \tag{117}$$

$$= \|A_2 - Q_2^{-1}QA\|_F. \tag{118}$$

Since $Q_2^{-1}Q$ is a rotation and $A, A_2$ are diagonal, we can always set $Q_2 = Q$ without increasing $\|W - W^*\|_F$. Then

$$\|W - W^*\|_F^2 = \|A_2 - A\|_F^2 \tag{119}$$

$$= s(\sqrt{\alpha} - 1)^2 + s(\sqrt{\beta} - \sqrt{1/T})^2 + \sum_{j \in P_i, i \ne t} ((A_2)_{jj} - \sqrt{1/T})^2. \tag{120}$$

To minimize the contrastive loss, we need $\alpha, \beta$ to be as large as possible, subject to $\|W - W^*\|_F^2 \le \gamma^2$, and $\alpha, \beta, (A_2)_{jj}^2 \in [0,1]$. The optimal is then achieved when $\alpha = 1, \sqrt{\beta} = \min\left(1, \sqrt{\frac{1}{T}} + \frac{\gamma}{\sqrt{s}}\right)$, and $(A_2)_{jj} = \sqrt{1/T}$ for $j \in P_i, i \ne t$. $\qquad\square$

Now, recall that by Proposition 2.3, the ERM has risk:

$$\mathbb{E}_{(x,y)\sim\mathcal{D}_t}[\ell_c(\hat{u}^\top\phi^*(x),y)]$$

$$\leq 4L\sqrt{\frac{1}{m}\left(\frac{v_{t,1}}{\alpha}+\frac{v_{t,2}}{\beta}\right)}\left(\sqrt{s\alpha+(r-s)\beta}+O\left(\sqrt{\frac{r}{s\alpha+(r-s)\beta}}\right)\right)+8\sqrt{\frac{2\ln(4/\delta)}{m}}$$

$$= 4L\sqrt{\frac{1}{m}\left(\frac{\|u_t^2\|^2}{2\alpha}+\frac{\|u_t^2\|^2}{2\beta}\right)}\left(\sqrt{s\alpha+s\beta}+O\left(1\right)\right)+8\sqrt{\frac{2\ln(4/\delta)}{m}}$$

$$= O\left(L\sqrt{\frac{r\|u_t^*\|^2}{m}\left(2+\frac{\alpha}{\beta}+\frac{\beta}{\alpha}\right)}\right)$$

With the fine-tuning using contrastive learning, in the representation learned, $\alpha$ remains to be $1$, while $\beta$ increases from $1/T$ to $(\sqrt{1/T}+\gamma/\sqrt{s})^2$. Then the error bound decreases. This shows that fine-tuning with contrastive learning on unlabeled data from the target task can emphasize the task-specific features $z_{P_t}$, which then leads to better prediction performance.

# C   MORE EXPERIMENTAL DETAILS AND RESULTS

## C.1   DATASETS

**CIFAR-10.** CIFAR-10 (Krizhevsky et al., 2009) dataset consists of 60,000 $32 \times 32$ color images in 10 classes: airplane, automobile, bird, cat, deer, dog, frog, horse, ship, truck. Each class has 6,000 images. There are 50,000 training images and 10,000 test images.

**CINIC-10.** CINIC-10 (Darlow et al., 2018) consists of $32 \times 32$ color images from both CIFAR and ImageNet and has 90,000 training images with ten classes identical to CIFAR-10.

**ImageNet.** ImageNet (Deng et al., 2009) is a huge visual dataset which is composed of 1,281,167 training data and 50,000 test data with 1,000 classes. We crop each color image to $224 \times 224$ as the standard setting.

**ImageNet32.** ImageNet32 (Deng et al., 2009) is a huge dataset made up of small color images called the down-sampled version of ImageNet. ImageNet32 comprises 1,281,167 training data and 50,000 test data with 1,000 classes. Each color image is down-sampled to $32 \times 32$.

**ImageNet22k.** ImageNet22k (Deng et al., 2009; Ridnik et al., 2021) is a superset of ImageNet which contains 14.2M training data and 522,500 test data with 21,841 classes. We crop each color image to $224 \times 224$ as the standard setting.

**ImageNet-Bird.** The ImageNet-Bird is a subset of ImageNet and contains all bird-related images from ImageNet, which have 59 classes and 76k training images.

**ImageNet-Vehicle.** The ImageNet-Vehicle is a subset of ImageNet and contains all vehicle-related images from ImageNet, which have 43 classes and 55k training images.

**ImageNet-Cat/Ball/Shop/Clothing/Fruit.** The ImageNet-Cat/Ball/Shop/Clothing/Fruit is a subset of ImageNet and contains all cat/ball/shop/clothing/fruit-related images from ImageNet, which have 76 classes and 100k training images.

**GCC-15M.** GCC-15M denotes the merged version of GCC-3M (Sharma et al., 2018) and GCC-12M (Changpinyo et al., 2021). It is a diverse dataset of visual concepts with image-text pairs meant to be used for vision-and-language pre-training. GCC-15M contains 15M training data and more than 600k concepts.

**SVHN.** The Street View House Numbers (Netzer et al., 2011) contains 10 digits color images of size $32 \times 32$ in the natural scene. It has 73,257 digits for training and 26,032 digits for testing.

**MNIST.** The Modified National Institute of Standards and Technology (LeCun et al., 1998) is a database of handwritten gray-scale digits of size $28 \times 28$. It contains 60,000 training images and 10,000 testing images.

**EMNIST.** Extended MNIST (Cohen et al., 2017) includes gray images of handwritten letters and digits. The images in EMNIST were converted into the same size $28 \times 28$ by the same process as MNIST. EMNIST-Letters has 145,600 lowercase characters with 26 balanced classes, and EMNIST-Digits has 280,000 characters with ten balanced classes.

**Fashion-MNIST.** Fashion-MNIST (Xiao et al., 2017) is a dataset of $28 \times 28$ gray-scale images with ten classes: T-shirt/top, trouser, pullover, dress, coat, sandal, shirt, sneaker, bag, ankle boot. The training set size is 60,000, and the test set size is 10,000.

**Fer2013.** Fer2013 is a dataset (Goodfellow et al., 2013) of $48 \times 48$ gray-scale images with 7 face expression classes: angry, disgust, fear, happy, sad, surprise, neutral. The training set size is 28,709, and the test set size is 3,589.

**FaceScrub.** FaceScrub (Ng & Winkler, 2014) is a dataset with 141,130 color face images of 695 public figures.

**GTSRB.** The German Traffic Sign Recognition Benchmark (Stallkamp et al., 2012) is a dataset of color images depicting 43 different traffic signs. The images are not of fixed dimensions and have a rich background and varying light conditions as expected of photographed images of traffic signs. The original training set contains 34,799 images, and the original test set contains 12,630 images. We resize each image to $32 \times 32$. The dataset has a significant imbalance in the number of sample occurrences across classes. We use data augmentation techniques to enlarge the training data and balance the number of samples in each class. We construct a class-preserving data augmentation pipeline consisting of rotation, translation, and projection transforms and apply this pipeline to the training images until each class contains 2,500 examples. So we construct a new training set containing 107,500 images in total. We also construct a new test set by randomly selecting 10,000 images from the original test set for evaluation.

**IMDB.** IMDB (Maas et al., 2011) is a large movie review text dataset. The dataset is for binary sentiment classification, positive review or negative review. The dataset contains 25,000 movie reviews for training and 25,000 for testing.

**AGNews.** AGNews (Zhang et al., 2015) is a sub-dataset of AG's corpus of news articles for text topic classification. It covers the 4 largest classes: world, sports, business, sci/tech. The AG News contains 30,000 training and 1,900 test samples per class.

### C.2 VERIFYING THE EXISTENCE OF THE TRADE-OFF

**Model.** We evaluate three popular contrastive learning frameworks, MoCo v2 (He et al., 2020), NNCLR (Dwibedi et al., 2021) and SimSiam (Chen & He, 2021) . MoCo v2 can be regarded as SimCLR (Chen et al., 2020) equipped with a memory bank, while NNCLR uses nearest-neighbor as the positive pairs. SimSiam can be regarded as a modification from BYOL (Grill et al., 2020) similar to Barlow Twins (Zbontar et al., 2021), which does not need negative pairs. We follow the same data augmentation methods as SimSiam (Chen & He, 2021) for all datasets.

**Datasets.** We consider three sets of data. In the first set, our downstream task is CIFAR-10, and the pre-training datasets may include CINIC-10, SVHN, GTSRB, and ImageNet32. CINIC-10 has classes identical to CIFAR-10 and is the most target-relevant, while the others are less similar to CIFAR-10. This then provides more and more diverse pre-training data w.r.t. the target task. In the second set, our downstream task is MNIST, and the pre-training datasets may include EMNIST-Digits&Letters, Fashion-MNIST, GTSRB, and ImageNet32. Here, the handwritten dataset EMNIST-Digits&Letters is the most target-relevant. In the last set, our downstream task is Fer2013, a face expression classification dataset. The pre-training datasets may include FaceScrub, CIFAR-10, SVHN, and ImageNet32, where the face dataset Facescrub is the most target-relevant.

**Evaluation & Methods.** We pre-train a ResNet18 network (He et al., 2016) as a feature extractor under different contrastive learning methods using SGD for 800 epochs with a cosine learning-rate scheduler, the base learning rate of 0.06, weight decay 5e-4, momentum 0.9 and batch size 512. Then we fix the pre-trained feature extractor and train a linear classifier (Linear Probing, LP) on $1\%, 5\%, 10\%, 20\%, 100\%$ of the labeled data from the downstream task. For LP we use SGD for 200 epochs with a cosine learning-rate scheduler, the base learning rate of 5.0, no weight decay, momentum 0.9, and batch size 256. We finally report the test accuracy on a specific target task and

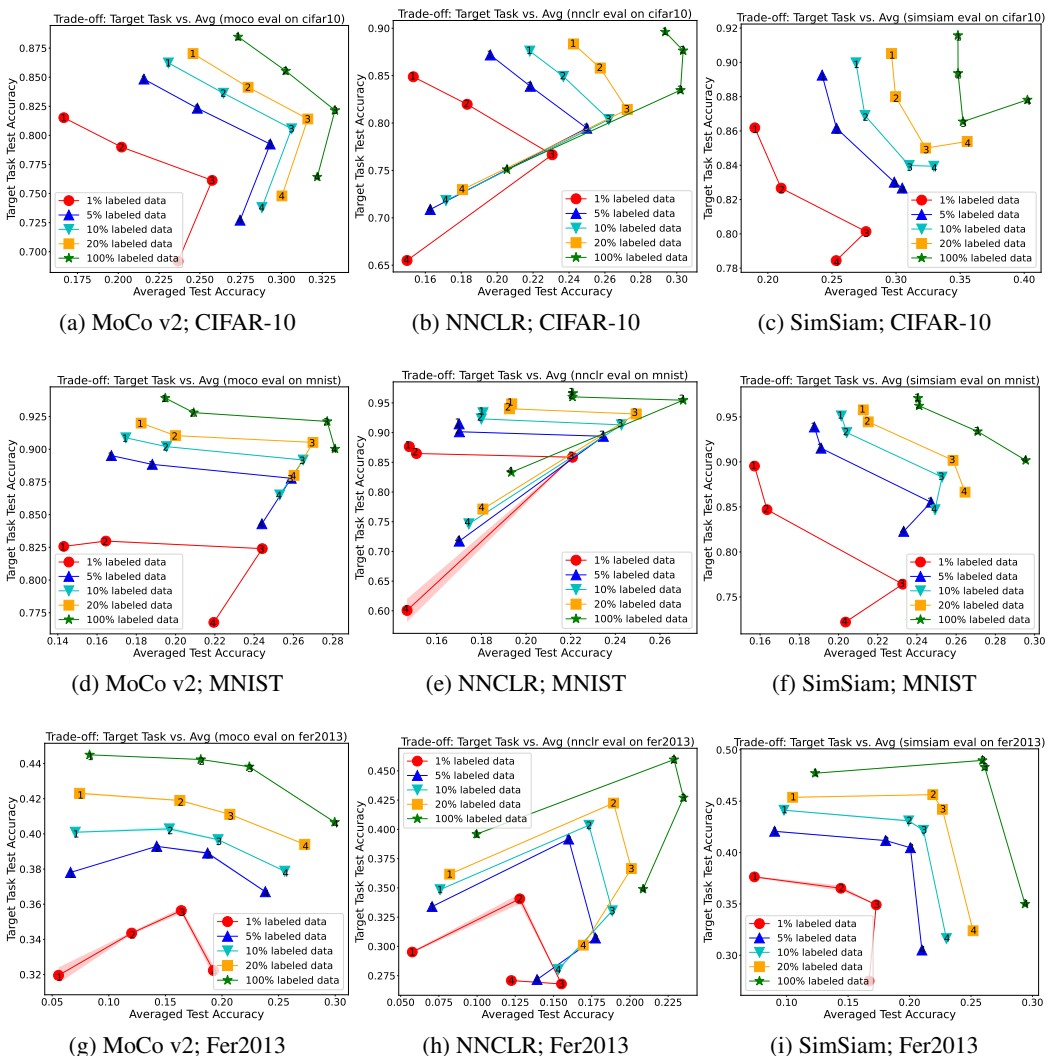

Figure 7: Trade-off of universality and label efficiency for MoCo v2, NNCLR, SimSiam on downstream tasks CIFAR-10, MNIST, Fer2013. "1, 2, 3, 4" means incrementally adding datasets for pre-training. The $x$-axis is the average test accuracy of Linear Probing on all 4 datasets. The $y$-axis is test accuracy on the target task. Pre-training data: (a)(b)(c) CINIC-10, SVHN, GTSRB, and ImageNet32. Target task: CIFAR-10. (d)(e)(f) EMNIST-Digits&Letters, Fashon-MNIST, GTSRB, ImageNet32. Target: MNIST. (g)(h)(i) FaceScrub, CIFAR-10, SVHN, ImageNet32. Target: Fer2013.

the weighted average test accuracy on all pre-training datasets (i.e., using them as the downstream tasks). We use the class number of each pre-training dataset as the weight, which is consistent with random guessing as a baseline. We have three types of models pre-trained on three sets of datasets. Thus, we have nine tasks in total. For each task, we have two pre-trained models initialized by different random seeds. We evaluated each model three times and we report the average test accuracy with standard deviation based on multiple runs (six evaluations).

In Figs. 7(a)(b)(c), we report results for MoCo v2, NNCLR, SimSiam (respectively) on CIFAR-10 as the downstream task. The size and diversity of unlabeled data for pre-training are increased on the $x$-axis by incrementally adding datasets in the following order: CINIC-10 (C), SVHN (S), GTSRB (G), and ImageNet32 (only use a 500k subset)(I). Then, we do LP on CIFAR-10 using different proportions of labeled samples. Similarly, in Figs. 7(d)(e)(f), we report results for three models on

MNIST. We incrementally add pretraining datasets in the following order: EMNIST-Digits&Letters (E), Fashion-MNIST (F), GTSRB (G), ImageNet32 (I). In Figs. 7(g)(h)(i), the downstream task is Fer2013 and we incrementally add datasets in the following order: FaceScrub (I), CIFAR-10 (C), SVHN (S), ImageNet32 (I).

**Results.** In Figs. 7, when the pre-training data becomes more diverse, the average test accuracy on all pre-training datasets increases (i.e., universality improves), while the test accuracy on the specific target task decreases (i.e., label efficiency drops). This shows a clear trade-off between universality and label efficiency. Moreover, with fewer labeled data (from the green line to the red line), the trade-off phenomenon will be more significant. It supports our claim that diverse pre-training data allow learning diverse features for better universality, but can down-weight the features for a specific task resulting in worse prediction. As more diverse unlabeled data are included, more labeled data from the target task is needed to achieve comparably-good prediction accuracy. This validates our analysis of the trade-off in Section 2.2.

In Figs. 7(a)(b)(d)(e)(f)(h), the average test accuracy ($x$-axis) decreases in the end because when we add one pre-training dataset, it may hurt the test accuracy of all other pre-training datasets. In Figs. 7(g)(h), the downstream task test accuracy ($y$-axis) increases in the beginning because when the pre-training unlabeled data from relevant tasks is not sufficiently large, introducing other pre-training datasets will help the model to learn features relevant for the downstream task. However, the downstream task test accuracy will drop later as in other figures.

Please refer to Appendix C.5 for more figures.

### C.2.1 LARGER SCALE EXPERIMENTS

The datasets involved are ImageNet (1.2M data points, 1k classes), ImageNet22k (14M, 22k classes), and GCC-15M (15M). We compare two UniCL representations (Yang et al., 2022): the more specific representation pre-trained on ImageNet, and the one pre-trained on the more diverse dataset ImageNet+GCC-15M. We compare their performance in two tasks: classification on ImageNet (using 2k labeled data) and classification on ImageNet22k (using 44k labeled data).

The results are reported in Table 3. From the specific representation to the diverse one, we observe that the test accuracy on ImageNet decreases (i.e., efficiency drops), while the test accuracy over ImageNet22k increases (i.e., universality improves). This again confirms the trade-off.

| LP Accuracy | **Pre-training** dataset | |
|---|---|---|
| **Target** dataset | ImageNet | ImageNet+GCC-15M |
| ImageNet (2k label) | 79.05 | 77.66 |
| ImageNet22k (44k label) | 9.02 | 9.86 |

Table 3: LP test accuracy on ImageNet and ImageNet22k with UniCL (Swin-T) pre-trained 500 epochs on ImageNet and ImageNet+GCC-15M.

### C.3 INSPECTING THE TRADE-OFF

### C.3.1 FEATURE SIMILARITY

For each set of pre-training data, we extract a set of features for the target task, CIFAR-10, MNIST, and Fer2013 respectively, using the pre-trained representation function. In Fig. 8 (rows/columns are pre-training data; numbers/colors show the similarity) first row, the features from different pre-training datasets have low similarities. This is consistent with our setup in Section 2.2 that different tasks only share some features and each owns many private ones. In the second row, we can see a decreasing trend of similarity in each row of each sub-figure. This indicates when gradually adding more diverse pre-training data, the obtained representation will encode more downstream task-irrelevant information and become less similar to that before adding more pre-training data. It will hurt the downstream task performance. This result is consistent with our Proposition 2.3 and 2.4.

Finally, we would like to verify in Theorem 2.2 via CKA similarities. The theorem says that when we increase the norm bound, the representation can encode more and more features. To verify this, our

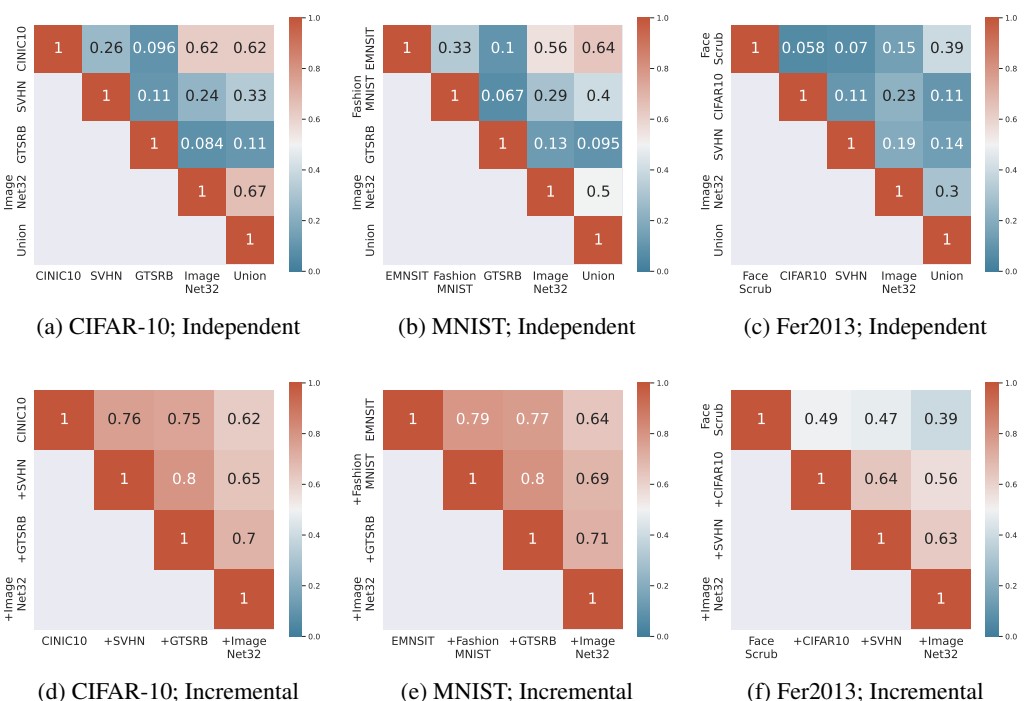

(a) CIFAR-10; Independent     (b) MNIST; Independent     (c) Fer2013; Independent

(d) CIFAR-10; Incremental     (e) MNIST; Incremental     (f) Fer2013; Incremental

Figure 8: Linear CKA similarity score among downstream task features from MoCo v2 pretrained on three sets of datasets. For "Independent", each representation model in the first four columns/rows is pre-trained on a single dataset. "Union" indicates the model pre-trained on the union among four disjoint datasets. "Incremental" means from left column to right, from top row to bottom, we incrementally add datasets for pre-training.

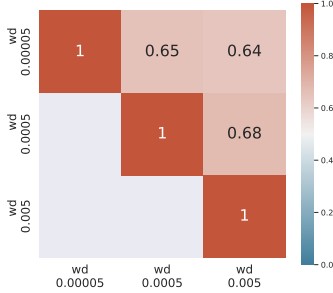

Figure 9: Linear CKA similarity among CIFAR10 features from MoCo v2 pre-trained on CINIC10. Each representation in the first three columns/rows is pre-trained with a different weight decay value.

experiments will vary the weight decay regularization coefficient (larger weight decay corresponds to a smaller norm bound and fewer features learned in the representation). Fig. 9 shows that the linear CKA similarity decreases with the increase of the weight decay, then it provides some support for the theorem.

### C.3.2   THE EFFECT OF PRE-TRAINING AND LABELED DATA SIZES

We have also conducted finer-grained investigations into the trade-off by varying the pre-training dataset size and the downstream labeled data on the specific task. Table 4 shows the results for different pre-training datasets (ImageNet-Bird, ImageNet, and 10% of ImageNet data) and different labeled datasets (500 to 8k samples from ImageNet-Bird, and the whole ImageNet). Table 5 shows

the results for a similar setting with ImageNet-Vehicle. Table 6 shows the results of UniCL with Swin-T backbone using different pre-training datasets (ImageNet, and ImageNet+GCC-15M) and different labeled datasets (2k samples from ImageNet, 44k samples from ImageNet22k, the whole ImageNet, and the whole ImageNet22k).

First, we find that the trade-off is hidden when a small amount of data from the specific task is used for pre-training. The results show that when the specific pre-training data is small, the representation learned is noisy and the downstream prediction performance is poor. This is not surprising: in the extreme case with only 1 pre-training image from the bird task or vehicle task, there is essentially no information for pre-training. In this case, the results are well inside the Pareto front of the trade-off curve and thus cannot demonstrate the trade-off.

Second, we find that the trade-off is hidden when a large amount of labeled data are available for learning predictors on the representation in the specific task. If a large amount of labeled data is available for training the predictor, the prediction performance is similar when using the specific or universal representations. This is consistent with the insights from our analysis: when pre-training on diverse data, the features specific to the target task are down-weighted but can still be in the representation, then with a large amount of labeled data from the specific task, the sample complexity issue is not significant, and thus the trade-off is hidden.

The above experimental studies show that the trade-off is revealed when we have *large-scale pre-training data* and *a limited amount of labeled data from the target task*, which is indeed the typical interesting case for using pre-trained representations (especially the large foundation models). The trade-off is significant in this case and thus crucial for further development of pre-training representations.

| LP Accuracy | Pre-training dataset | | |
|---|---|---|---|
| **Target** dataset | ImageNet-Bird | ImageNet | 10% ImageNet |
| ImageNet-Bird (500 label) | 76.00 | 58.78 | 30.06 |
| ImageNet-Bird (1k label) | 81.42 | 69.39 | 35.96 |
| ImageNet-Bird (2k label) | 82.88 | 79.66 | 39.49 |
| ImageNet-Bird (4k label) | 83.83 | 83.59 | 44.13 |
| ImageNet-Bird (8k label) | 84.71 | 85.93 | 48.50 |
| ImageNet (all label) | 41.38 | 73.20 | 54.08 |

Table 4: LP test accuracy on ImageNet-Bird and ImageNet with MoCo v3 (ViT-S) pre-trained on ImageNet-Bird and ImageNet.

| LP Accuracy | Pre-training dataset | | |
|---|---|---|---|
| **Target** dataset | ImageNet-Vehicle | ImageNet | 10% ImageNet |
| ImageNet-Vehicle (500 label) | 61.81 | 57.86 | 31.48 |
| ImageNet-Vehicle (1k label) | 70.93 | 67.90 | 37.76 |
| ImageNet-Vehicle (2k label) | 72.09 | 69.81 | 39.07 |
| ImageNet-Vehicle (4k label) | 74.14 | 72.93 | 39.62 |
| ImageNet-Vehicle (8k label) | 75.53 | 74.55 | 43.53 |
| ImageNet (all label) | 39.84 | 73.20 | 54.08 |

Table 5: LP test accuracy on ImageNet-Vehicle and ImageNet with MoCo v3 (ViT-S) pre-trained on ImageNet-Vehicle and ImageNet.

| LP Accuracy | Pre-training dataset | |
|---|---|---|
| **Target** dataset | ImageNet | ImageNet+GCC-15M |
| ImageNet (2k label) | 79.05 | 77.66 |
| ImageNet22k (44k label) | 9.02 | 9.86 |
| ImageNet (all label) | 79.61 | 81.37 |
| ImageNet22k (all label) | 30.90 | 36.69 |

Table 6: LP test accuracy on ImageNet and ImageNet22k with UniCL (Swin-T) pre-trained 500 epochs on ImageNet and ImageNet+GCC-15M.

### C.3.3  MORE ABLATION STUDIES

We report results from three ablation studies: (1) varying the class number of ImageNet32, (2) varying the percentage of target-relevant pre-training data, and (3) replacing CINIC-10 with CIFAR-10 in the pre-training dataset. The results consistently show the trade-off.

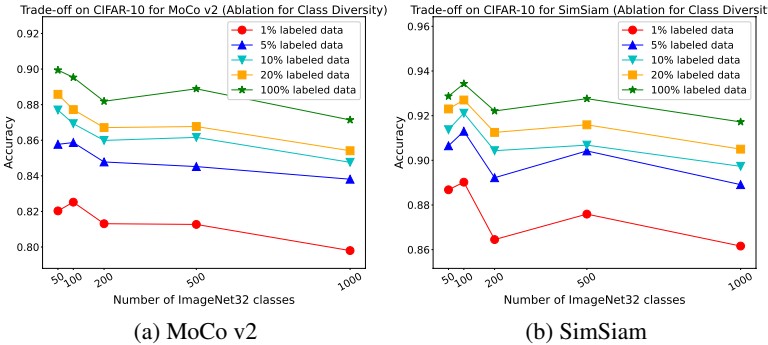

(a) MoCo v2      (b) SimSiam

Figure 10: Pre-train MoCo v2 and SimSiam on CIFAR-10 + ImageNet32(200k) with varying number of classes of ImageNet32 from 50 to 1000 ($x$-axis) under a fixed size of pre-training data. The $y$-axis is LP test accuracy on CIFAR-10.

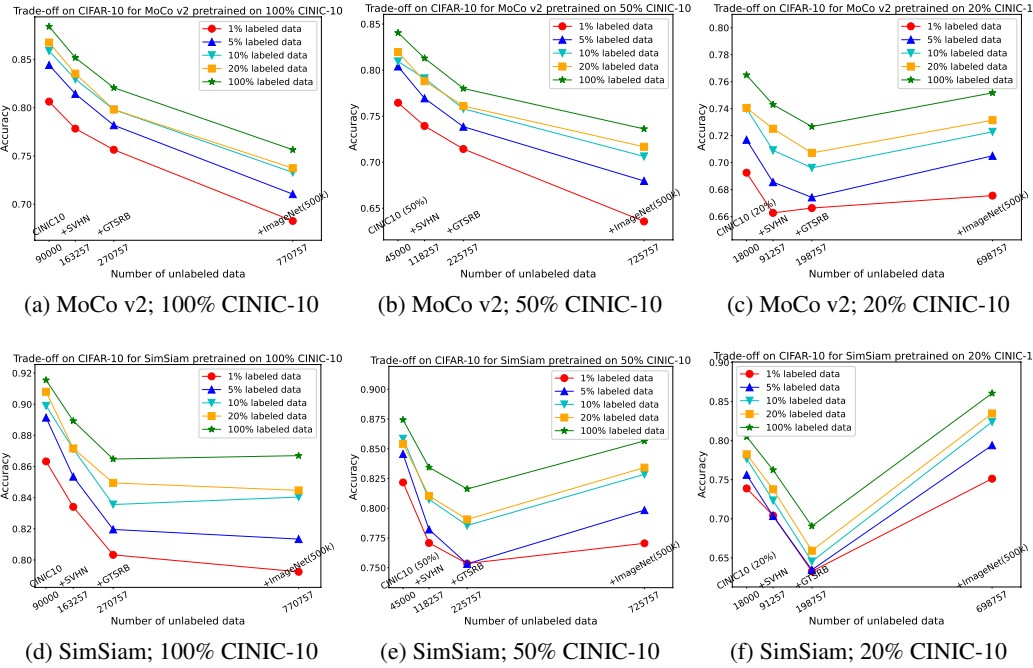

(a) MoCo v2; 100% CINIC-10    (b) MoCo v2; 50% CINIC-10    (c) MoCo v2; 20% CINIC-10

(d) SimSiam; 100% CINIC-10    (e) SimSiam; 50% CINIC-10    (f) SimSiam; 20% CINIC-10

Figure 11: Trade-off on CIFAR-10 LP test accuracy ($y$-axis) for MoCo v2 and SimSiam with varying target relevant (CINIC-10) pre-training data percentage (100%, 50%, 20%).

**Varying the Class Number of ImageNet32.** To further support **A1**, we show that the trade-off between universality and label efficiency also exists under a fixed dataset size. In Fig. 10, we pre-train MoCo v2 and SimSiam on CIFAR-10 + ImageNet32(200k) and keep the same setting as Fig. 7 except that we vary the class number of ImageNet32(200k). In previous experiments, we randomly pick 500,000 images from ImageNet32 without considering labels. Here, we fix the number of classes to 50, 100, 200, 500, 1000. Then we randomly sample 200,000 images from the subset of classes. The downstream task is CIFAR-10. In Fig. 10, we observe that with a fixed pre-training

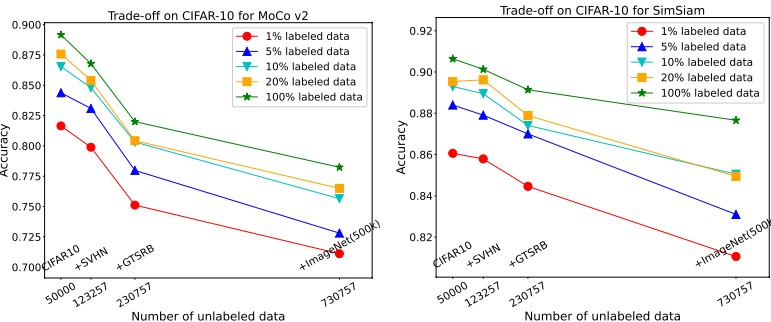

Figure 12: Trade-off on CIFAR-10 LP test accuracy ($y$-axis) for MoCo v2 and SimSiam pre-trianed on datasets including CIFAR-10.

datasets size, e.g., 250,000, when the data is more diverse, the pre-training will learn more irrelevant features, and the performance will drop on the downstream task. This supports our analysis as well.

**Varying Target-Relevant pre-training Data Percentage.** In Fig. 11, we use (a)(d) 100% (b)(e) 50% (c)(f) 20% CINIC-10 to train MoCo v2 and SimSiam, and keep the same setting as Fig. 7. For Fig. 11 (b) with 50% CINIC-10, test accuracy drops, e.g., the test accuracy of 1% CIFAR-10 in Fig. 11 (a) 80.63% vs. (b) 76.45%. We can also see the decreasing curve in Fig. 11 (b). On the other hand, we also have test accuracy drops in Fig. 11 (c)(e)(f). However, we can see a U-curve rather than a strictly decreasing curve in Fig. 11 (c)(e)(f). ImageNet32 is more relevant with CIFAR-10 than SVHN and GTSRB, consistent with human intuition. When we have a small partition of CINIC-10 that does not cover all target-relevant features, the feature extractor can learn these missing features from ImageNet32. Although there are many irrelevant features in ImageNet32, the positive effect is larger than the negative effect, and so it plots a U-curve. It is consistent with our statement that we need a large and target-relevant pre-training dataset rather than a diverse irrelevant one.

**Replacing CINIC-10 With CIFAR-10.** In Fig. 12, we keep the same setting as Fig. 7 except we replace CINIC-10 with CIFAR-10. Note that our downstream task is still CIFAR-10. In Fig. 12, we can see the same phenomena and similar performance as Fig. 7. Thus, if we have a good choice of a task-relevant pre-training dataset, we can get a similar performance as pre-training on the down-stream task domain directly.

The pre-training unlabeled data from diverse tasks may have a positive effect when the pre-training unlabeled data from similar (relevant) tasks is not sufficiently large. Moreover, if we choose a good task-relevant pre-training dataset, we can directly get a similar performance as pre-training on the downstream task. However, when the task-relevant dataset is sufficient, the performance will drop if we introduce task-irrelevant data in the pre-training dataset (Fig. 11 (a)(d)).

## C.4 IMPROVING THE TRADE-OFF: FINETUNE WITH CONTRASTIVE REGULARIZATION

| **Pretrain** data | Method | 1% label data | 5% label data | 10% label data | 20% label data | 100% label data |
|---|---|---|---|---|---|---|
| CINIC10 | LP | 80.63±0.01 | **84.42**±**0.01** | 85.88±0.05 | 86.75±0.01 | 88.41±0.01 |
| | FT | 68.74±0.46 | 81.46±0.34 | 85.20±0.14 | 88.69±0.29 | 93.58±0.14 |
| | Ours | **83.66**±**0.43** | 83.04±0.08 | **86.18**±**0.24** | **89.61**±**0.12** | **94.51**±**0.02** |
| +SVHN | LP | 77.83±0.01 | 81.43±0.04 | 82.95±0.01 | 83.53±0.01 | 85.18±0.01 |
| | FT | 66.01±0.52 | 80.20±0.07 | 83.96±0.46 | 88.01±0.07 | 93.35±0.10 |
| | Ours | **79.95**±**0.36** | **81.77**±**0.11** | **84.98**±**0.02** | **88.97**±**0.14** | **94.26**±**0.01** |
| +GTSRB | LP | 75.64±0.01 | 78.18±0.01 | 79.80±0.02 | 79.81±0.01 | 82.07±0.01 |
| | FT | 62.79±0.57 | 78.90±0.41 | 83.89±0.03 | 87.85±0.03 | 93.42±0.13 |
| | Ours | **76.20**±**0.36** | **80.26**±**0.05** | **84.11**±**0.06** | **88.74**±**0.01** | **94.32**±**0.13** |
| +ImageNet32 | LP | 68.26±0.01 | 71.04±0.01 | 73.29±0.02 | 73.73±0.02 | 75.64±0.03 |
| | FT | 65.40±0.16 | **78.99**±**0.21** | 82.96±0.17 | 86.75±0.10 | 92.92±0.06 |
| | Ours | **71.70**±**1.20** | 78.94±0.06 | **83.48**±**0.02** | **87.77**±**0.13** | **93.66**±**0.12** |

Table 7: Test accuracy on CIFAR-10 with different evaluation methods on MoCo v2 under different percentages of labeled data. From top to bottom: incrementally add datasets for pre-training.

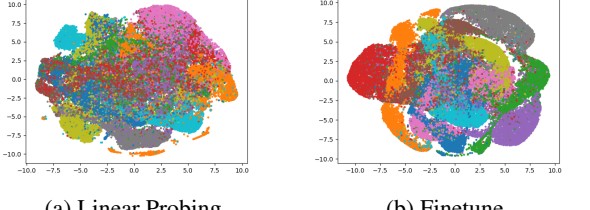 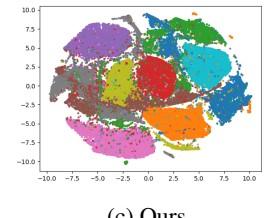

|          (a) Linear Probing          |          (b) Finetune          |          (c) Ours          |

Figure 13: The t-SNE visualization (Van der Maaten & Hinton, 2008) for CIFAR-10 training data normalized features from different evaluation methods, where the model is pre-trained on (CSGI) defined in Fig. 3. FT and Ours are trained on the 20% CIFAR-10 training dataset. Different colors correspond to different classes.

**Evaluation & Methods.** We use the feature extractor (ResNet18) pre-trained as in Section 3.1 by MoCo v2. Then, we report three evaluation methods, Linear Probing (LP), Finetune (FT), and Finetune with Contrastive Regularization (Ours) on CIFAR-10 under different percentages of labeled data as in Appendix C.2. LP follows the training protocol in Section 3.1. FT and Ours learn a linear predictor and update the representation, and use the same data augmentation for a fair comparison. FT follows the training in MAE (He et al., 2022) mostly, using SGD for 200 epochs with a cosine learning-rate scheduler, the base learning rate of $0.06$, weight decay 5e-4, momentum $0.9$ and batch size 256. Moreover, Ours uses the contrastive loss from MoCo v2 and regularization coefficient $\lambda = 0.1$.

**Results.** In Table 7, the trade-off phenomenon also exists for FT evaluation, where the FT test accuracy drops when the pre-training dataset contains more diverse data points. Table 7 shows that Ours can achieve better performance than the other baselines. In particular, it outperforms the typical fine-tuning method consistently, even when the latter also uses the same amount of data augmentation. This confirms the benefit of contrastive regularization. Fig. 13 visualizes the features of different methods by t-SNE. It shows that contrastive regularization can down-weight the downstream task-irrelevant invariant feature, so it can improve the model generalization ability, which is consistent with the discussion in Section 2.2.

| **Target** data | Method | Aug | 1% label data | 5% label data | 10% label data | 20% label data | 100% label data |
|---|---|---|---|---|---|---|---|
| ImageNet | LP | 1 | $66.04\pm0.26$ | $76.28\pm0.05$ | $78.06\pm0.02$ | $78.73\pm0.03$ | $77.84\pm0.02$ |
| | FT | 1 | $71.09\pm0.05$ | $75.32\pm0.11$ | $76.38\pm0.01$ | $76.92\pm0.10$ | $82.97\pm0.02$ |
| | FT | 10 | $73.28\pm0.01$ | $76.43\pm0.14$ | $78.69\pm0.04$ | $81.56\pm0.10$ | $83.65\pm0.01$ |
| | FT | 100 | $71.83\pm0.05$ | $77.55\pm0.07$ | $80.16\pm0.09$ | $82.23\pm0.01$ | $83.58\pm0.05$ |
| | Ours | 10 | $\mathbf{74.36}\pm\mathbf{0.09}$ | $\mathbf{78.60}\pm\mathbf{0.03}$ | $80.02\pm0.08$ | $81.28\pm0.07$ | $84.94\pm0.09$ |
| | Ours | 100 | $73.03\pm0.04$ | $78.41\pm0.04$ | $\mathbf{80.48}\pm\mathbf{0.05}$ | $\mathbf{82.42}\pm\mathbf{0.03}$ | $\mathbf{85.34}\pm\mathbf{0.07}$ |
| SVHN | LP | 1 | $59.24\pm0.02$ | $57.25\pm0.08$ | $56.32\pm0.17$ | $58.35\pm0.08$ | $63.44\pm0.01$ |
| | FT | 1 | $55.20\pm0.16$ | $61.73\pm0.32$ | $64.91\pm0.12$ | $64.70\pm0.64$ | $65.76\pm0.10$ |
| | FT | 10 | $61.24\pm0.08$ | $65.89\pm0.54$ | $68.61\pm0.52$ | $70.89\pm0.07$ | $78.22\pm0.18$ |
| | FT | 100 | $65.23\pm0.03$ | $72.34\pm0.07$ | $75.13\pm0.11$ | $77.20\pm0.03$ | $80.13\pm0.14$ |
| | Ours | 10 | $62.87\pm0.42$ | $67.56\pm0.23$ | $70.16\pm0.02$ | $72.23\pm0.09$ | $78.72\pm0.37$ |
| | Ours | 100 | $\mathbf{65.30}\pm\mathbf{0.43}$ | $\mathbf{72.61}\pm\mathbf{0.07}$ | $\mathbf{75.34}\pm\mathbf{0.12}$ | $\mathbf{77.64}\pm\mathbf{0.05}$ | $\mathbf{82.50}\pm\mathbf{0.01}$ |
| GTSRB | LP | 1 | $71.20\pm0.61$ | $83.97\pm0.08$ | $85.31\pm0.02$ | $86.34\pm0.06$ | $86.56\pm0.01$ |
| | FT | 1 | $77.18\pm0.40$ | $87.65\pm0.18$ | $88.56\pm0.75$ | $88.76\pm0.01$ | $89.83\pm0.39$ |
| | FT | 10 | $84.55\pm0.30$ | $88.74\pm0.32$ | $89.11\pm0.11$ | $89.52\pm0.12$ | $90.74\pm0.06$ |
| | FT | 100 | $86.28\pm0.26$ | $90.50\pm0.29$ | $90.83\pm0.08$ | $91.15\pm0.01$ | $91.89\pm0.12$ |
| | Ours | 10 | $84.84\pm0.33$ | $89.53\pm0.13$ | $89.44\pm0.01$ | $91.35\pm0.12$ | $92.01\pm0.28$ |
| | Ours | 100 | $\mathbf{87.24}\pm\mathbf{0.51}$ | $\mathbf{90.86}\pm\mathbf{0.05}$ | $\mathbf{91.29}\pm\mathbf{0.26}$ | $\mathbf{92.11}\pm\mathbf{0.15}$ | $\mathbf{92.65}\pm\mathbf{0.10}$ |

Table 8: Test accuracy for different evaluation methods on different datasets using foundation model CLIP (backbone ViT-L). We do not use data augmentation for LP. We evaluate FT without data augmentation, with 10 augmentation and with 100 augmentation to each training images. For Ours, we use 10, 100 augmentation.

| **Target** data | Method | Aug | 1% label data | 5% label data | 10% label data | 20% label data | 100% label data |
|---|---|---|---|---|---|---|---|
| CIFAR-10 | LP | 1 | **91.44**±**0.01** | 93.51±0.01 | 94.11±0.01 | 94.22±0.01 | 95.82±0.01 |
| | FT | 1 | 89.93±0.38 | 94.21±0.35 | 95.15±0.06 | 95.80±0.06 | 96.12±0.11 |
| | FT | 10 | 90.82±0.19 | 93.24±0.08 | 94.59±0.08 | 95.48±0.01 | 96.17±0.12 |
| | FT | 100 | 90.84±0.03 | 93.05±0.02 | 94.23±0.08 | 95.37±0.01 | **97.01**±**0.07** |
| | Ours | 10 | 90.73±0.18 | **94.28**±**0.03** | **95.43**±**0.15** | **95.83**±**0.14** | 96.71±0.10 |
| | Ours | 100 | 91.04±0.06 | 94.00±0.06 | 95.29±0.01 | 95.72±0.05 | 96.86±0.06 |
| SVHN | LP | 1 | 39.40±0.02 | 50.50±0.02 | 54.61±0.02 | 57.66±0.01 | 61.92±0.01 |
| | FT | 1 | 38.82±0.10 | 48.15±0.37 | 51.03±0.14 | 52.07±0.09 | 56.19±0.33 |
| | FT | 10 | 45.00±0.34 | 52.61±0.24 | 54.45±0.12 | 57.05±0.11 | 65.36±0.33 |
| | FT | 100 | 45.70±0.13 | 54.51±0.23 | 57.76±0.01 | 61.13±0.39 | 69.10±0.39 |
| | Ours | 10 | **46.53**±**0.23** | 55.59±0.03 | 57.32±0.05 | 59.20±0.09 | 66.29±0.20 |
| | Ours | 100 | 46.27±0.09 | **55.70**±**0.07** | **59.45**±**0.04** | **61.97**±**0.13** | **70.22**±**0.01** |
| GTSRB | LP | 1 | 48.52±0.03 | 66.68±0.02 | 71.69±0.01 | 72.27±0.02 | 75.37±0.01 |
| | FT | 1 | 52.54±0.67 | 72.19±0.33 | 73.75±0.71 | 73.30±0.46 | 75.16±0.01 |
| | FT | 10 | 58.68±0.01 | 75.21±0.23 | 75.87±0.06 | 76.75±0.28 | 76.45±0.29 |
| | FT | 100 | 61.65±0.16 | 75.22±0.77 | 76.49±0.31 | 77.67±0.63 | 78.05±0.30 |
| | Ours | 10 | 59.28±0.04 | 75.35±0.31 | 77.38±0.09 | **80.52**±**0.12** | **81.28**±**0.10** |
| | Ours | 100 | **63.61**±**0.92** | **76.70**±**0.17** | **77.85**±**0.23** | 80.13±0.01 | 80.92±0.19 |

Table 9: Test accuracy for different evaluation methods on different datasets using foundation model MoCo v3 (backbone ViT-B). We do not use data augmentation for LP. We evaluate FT without data augmentation, with 10 augmentations and with 100 augmentations to each training image. For Ours, we use 10, 100 augmentation.

| **Target** data | Method | 1% label data | 5% label data | 10% label data | 20% label data | 100% label data |
|---|---|---|---|---|---|---|
| IMDB | LP | 79.72±0.06 | 83.08±0.20 | 81.48±0.89 | 84.87±0.03 | 86.49±0.16 |
| | FT | 82.54±2.88 | 87.78±0.42 | 87.96±1.27 | 89.49±1.02 | 92.31±0.26 |
| | Ours | **84.48**±**1.62** | **90.12**±**0.41** | **90.41**±**0.49** | **90.79**±**1.58** | **92.85**±**0.03** |
| AGNews | LP | 85.52±0.31 | 86.78±0.62 | 86.75±0.66 | 87.62±0.24 | 87.76±0.66 |
| | FT | 88.74±0.34 | 90.76±0.70 | 91.22±0.07 | 92.36±0.14 | 93.57±0.23 |
| | Ours | **89.20**±**0.72** | **91.22**±**0.06** | **91.33**±**1.33** | **92.45**±**0.01** | **93.94**±**0.02** |

Table 10: Test accuracy for different evaluation methods on different datasets using foundation model SimCSE (backbone BERT). We do not use data augmentation for LP. We evaluate FT and Ours with the same data augmentation as SimCSE.

### C.4.1 LARGER FOUNDATION MODELS

We verify our method's effectiveness in real-world scenarios. When users use foundation models, they cannot access the model and can only call the official API, (e.g. GPT-3, DALL·E (Ramesh et al., 2022)). The pricing for GPT-3 to get feature embedding is $0.20 / 1k tokens. If users would like to use a foundation model on their downstream task, the most efficient way is to get feature embedding of their data from the API and train a small model, called adapter (Hu et al., 2021; Sung et al., 2022), on these embedding rather than on raw input data.

We evaluate CLIP (with ViT-L as the representation backbone), MoCo v3 (ViT-B backbone), and SimCSE (Gao et al., 2021) (BERT backbone). They are trained on (image, text), (image, image), and (text, text) contrastive pairs, respectively, so cover a good spectrum of methods.

For CLIP and MoCo v3, we fix the backbone for all evaluation methods. For LP, we add a linear FC layer on top of the backbone. For FT and Ours, we insert a two-layer ReLU neural network (1024 dimensions for the hidden layer) as an adapter between the backbone and the linear classification layer. For Ours, we apply SimCLR contrastive loss on the adapted feature (output of adapter) and set $\lambda = 1.0$. We use the same training strategy as Section 3.1 for LP and as Table 7 for FT and Ours. In line with the actual situation, we control the number of augmentation number used in FT and Ours (more data augmentation leads to higher prices in practice).

We conduct experiments on NLP tasks as well. SimCSE proposes a contrastive learning method for sentence embeddings, which uses the dropout feature from BERT as data augmentation. The max

sequence embedding length is 512. For SimCSE, all three evaluation methods use a linear classifier. For all evaluation methods, we use AdamW (Loshchilov & Hutter, 2018) with weight decay $= 0.01$ and train 3 epochs with batch size 16. For LP, we fix the backbone and set the learning rate as 5e-3. For FT and Ours, we train the backbone and linear classification layer simultaneously using unique data augmentation in each training step and set the learning rate as 5e-5. For Ours, we apply SimCSE contrastive loss on the feature (output of the backbone) and set $\lambda = 1.0$.

As in Appendix C.2, we report LP, FT, and Ours on $1\%, 5\%, 10\%, 20\%, 100\%$ of the labeled data from the downstream task in Table 8, Table 9 and Table 10 for CLIP, MoCo v3, and SimCSE respectively. For CLIP and MoCo v3 in Table 8 and Table 9, we consider different data augmentation numbers, e.g. 10, 100. For SimCSE, we use the standard data augmentation, i.e. generating unique augmentation for each gradient step.

These tables again show our method can consistently improve the downstream prediction performance in all three real-world scenarios, and quite significantly in some cases (e.g., MoCo v3 on GTSRB). This shows that the method is also useful for large foundation models, including the case when the foundation models cannot be fine-tuned and only the extracted embeddings can be adapted.

### C.4.2 THE EFFECT OF CONTRASTIVE REGULARIZATION ON LINEAR PROBING

We show that contrastive regularization can also improve over linear probing. Note that the contrastive regularization loss term is only relevant to the backbone; see the definition in Equation (9). While linear probing fixes the backbone. Thus, we cannot do contrastive regularization and linear probing at the same time. To show its effect, we first apply contrastive regularization to update the backbone, and after that use linear probing.

The results in Table 11 show that contrastive regularization leads to better prediction accuracy. Furthermore, the improvement is more significant on diverse pre-training data, consistent with our analysis.

| Method | Pre-training dataset | | | |
|---|---|---|---|---|
| | CINIC-10 | +SVHN | +GTSRB | +ImageNet32 |
| LP | 88.41 | 85.18 | 82.07 | 75.64 |
| Contrastive regularization then Linear probing | 88.38 | 86.91 | 85.95 | 82.43 |

Table 11: Test accuracy on CIFAR-10 with different evaluation methods on MoCo v2 with ResNet18 backbone. From left to right: incrementally add datasets for pre-training.

### C.4.3 THE EFFECT OF CONTRASTIVE REGULARIZATION ON CLOSING THE GAP

We show that contrastive regularization can reduce the target task performance gap between pre-training on the specific dataset (the same or similar as the target task) and that on diverse datasets.

We pre-train with MoCo v3 (backbone ViT-S) on ImageNet-Bird or the whole ImageNet and then perform linear probing (LP) on the target task of ImageNet-Bird with 1k labeled samples. The results in Table 12 show that pre-training on the diverse data leads to worse performance. Then we pre-train on ImageNet followed by contrastive regularization on ImageNet-Bird and then perform linear probing (LP) on the target task. This leads to improved performance than without contrastive regularization, closing the gap between pre-training on diverse and specific data. Similar results are observed in Table 13 when ImageNet-Vehicles are used. This confirms the benefits of contrastive regularization for highlighting task-specific features.

| LP Accuracy | Pre-training dataset | | |
|---|---|---|---|
| **Target** dataset | ImageNet-Bird | ImageNet | ImageNet then Contrastive Reguarlization on ImageNet-Bird |
| ImageNet-Bird (1k label) | 81.42 | 69.39 | 73.25 |

Table 12: LP test accuracy on ImageNet-Bird and ImageNet with MoCo v3 (ViT-S) pre-trained on ImageNet-Bird and ImageNet.

| LP Accuracy | **Pre-training** dataset | | |
|---|---|---|---|
| **Target** dataset | ImageNet-Vehicle | ImageNet | ImageNet then Contrastive Reguarlization on ImageNet-Vehicle |
| ImageNet-Vehicle (1k label) | 70.93 | 67.90 | 71.34 |

Table 13: LP test accuracy on ImageNet-Vehicle and ImageNet with MoCo v3 (ViT-S) pre-trained on ImageNet-Vehicle and ImageNet.

## C.5 ADDITIONAL RESULTS VERIFYING EXISTENCE OF THE TRADE-OFF

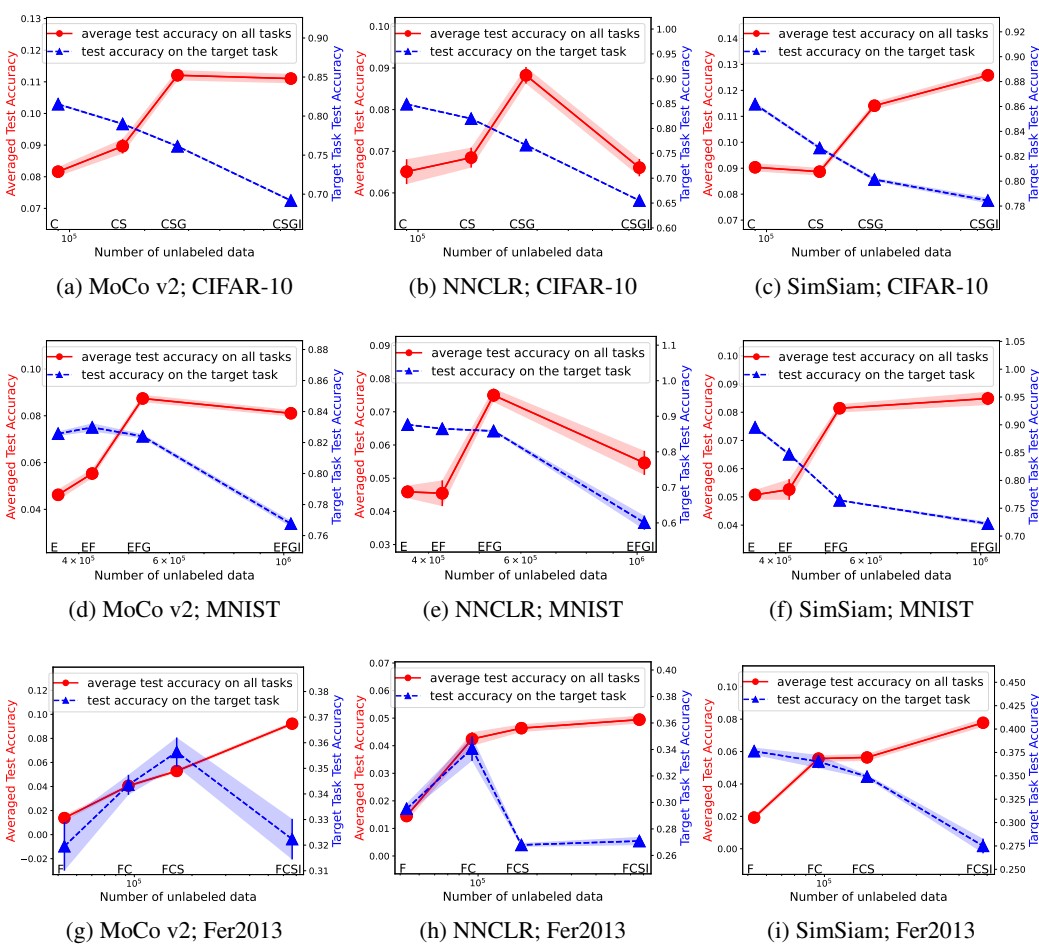

Figure 14: Trade-off of universality and label efficiency for MoCo v2, NNCLR, SimSiam on down-stream tasks CIFAR-10, MNIST, Fer2013. $x$-axis: incrementally add datasets for pre-training. Pre-training data: (a)(b)(c) CINIC-10 (C), SVHN (S), GTSRB (G), and ImageNet32 (I). For example, "CS" on the $x$-axis means CINIC-10+SVHN. Target task: CIFAR-10. Red line: average test accuracy of Linear Probing on all 4 datasets. Blue line: test accuracy on the target task. (d)(e)(f) EMNIST-Digits&Letters (E), Fashon-MNIST (F), GTSRB (G), ImageNet32 (I). Target: MNIST. (g)(h)(i) FaceScrub (F), CIFAR-10 (C), SVHN (S), ImageNet32 (I). Target: Fer2013. All evaluations are trained with 1% labeled data.

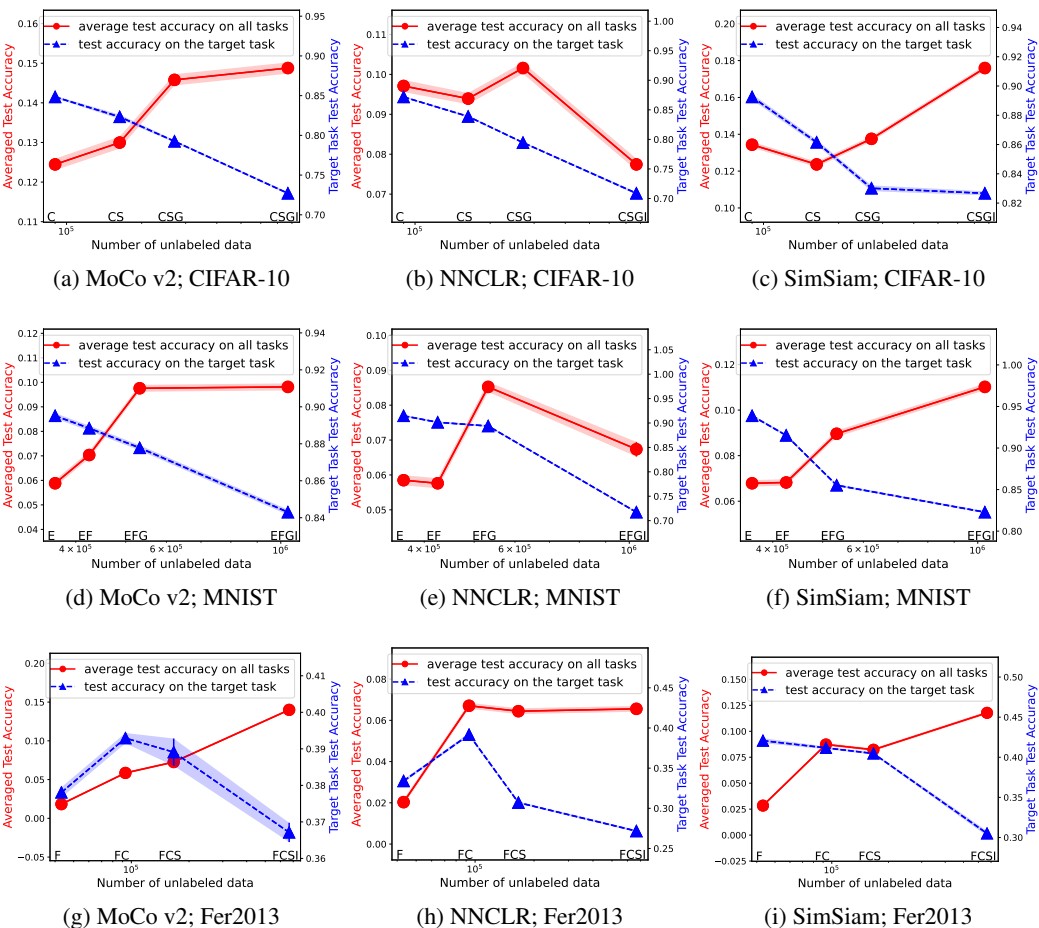

Figure 15: Trade-off of universality and label efficiency for MoCo v2, NNCLR, SimSiam on downstream tasks CIFAR-10, MNIST, Fer2013. $x$-axis: incrementally add datasets for pre-training. Pre-training data: (a)(b)(c) CINIC-10 (C), SVHN (S), GTSRB (G), and ImageNet32 (I). For example, "CS" on the $x$-axis means CINIC-10+SVHN. Target task: CIFAR-10. Red line: average test accuracy of Linear Probing on all 4 datasets. Blue line: test accuracy on the target task. (d)(e)(f) EMNIST-Digits&Letters (E), Fashon-MNIST (F), GTSRB (G), ImageNet32 (I). Target: MNIST. (g)(h)(i) FaceScrub (F), CIFAR-10 (C), SVHN (S), ImageNet32 (I). Target: Fer2013. All evaluations are trained with 5% labeled data.

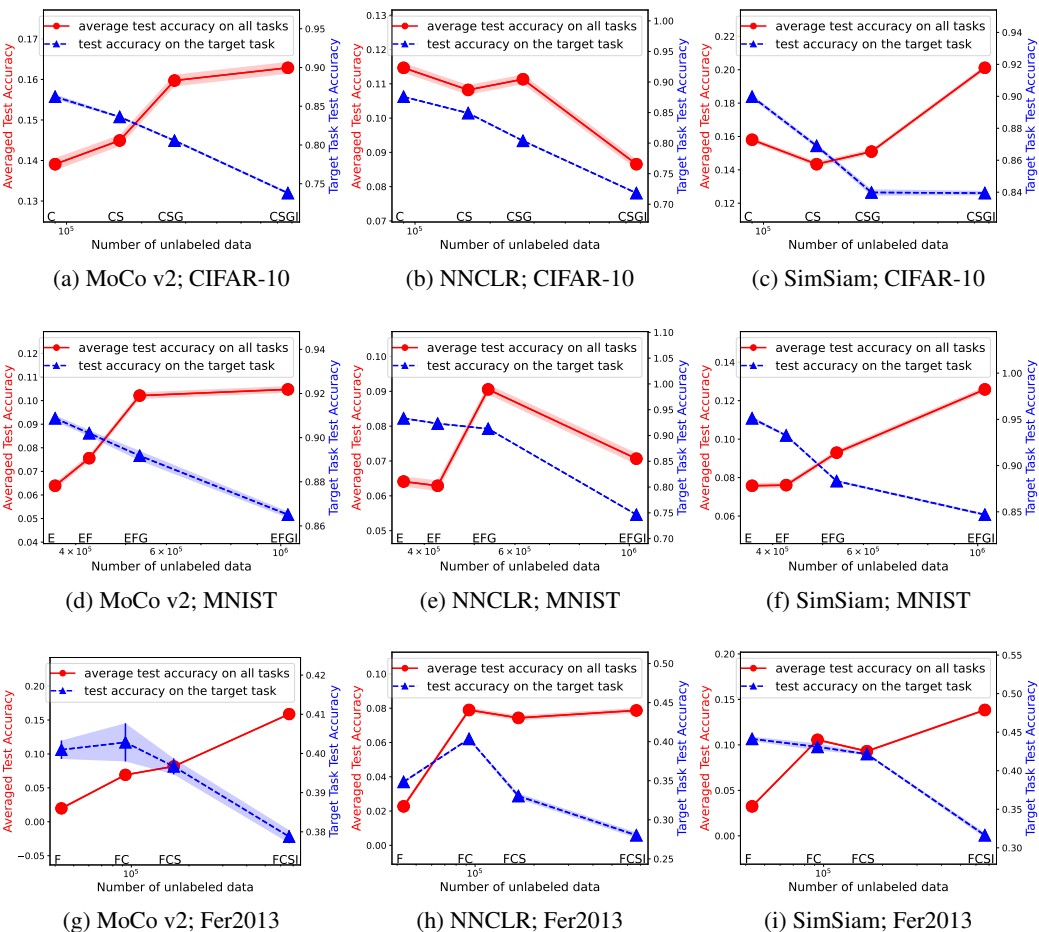

Figure 16: Trade-off of universality and label efficiency for MoCo v2, NNCLR, SimSiam on downstream tasks CIFAR-10, MNIST, Fer2013. $x$-axis: incrementally add datasets for pre-training. Pretraining data: (a)(b)(c) CINIC-10 (C), SVHN (S), GTSRB (G), and ImageNet32 (I). For example, "CS" on the $x$-axis means CINIC-10+SVHN. Target task: CIFAR-10. Red line: average test accuracy of Linear Probing on all 4 datasets. Blue line: test accuracy on the target task. (d)(e)(f) EMNIST-Digits&Letters (E), Fashon-MNIST (F), GTSRB (G), ImageNet32 (I). Target: MNIST. (g)(h)(i) FaceScrub (F), CIFAR-10 (C), SVHN (S), ImageNet32 (I). Target: Fer2013. All evaluations are trained with 10% labeled data.

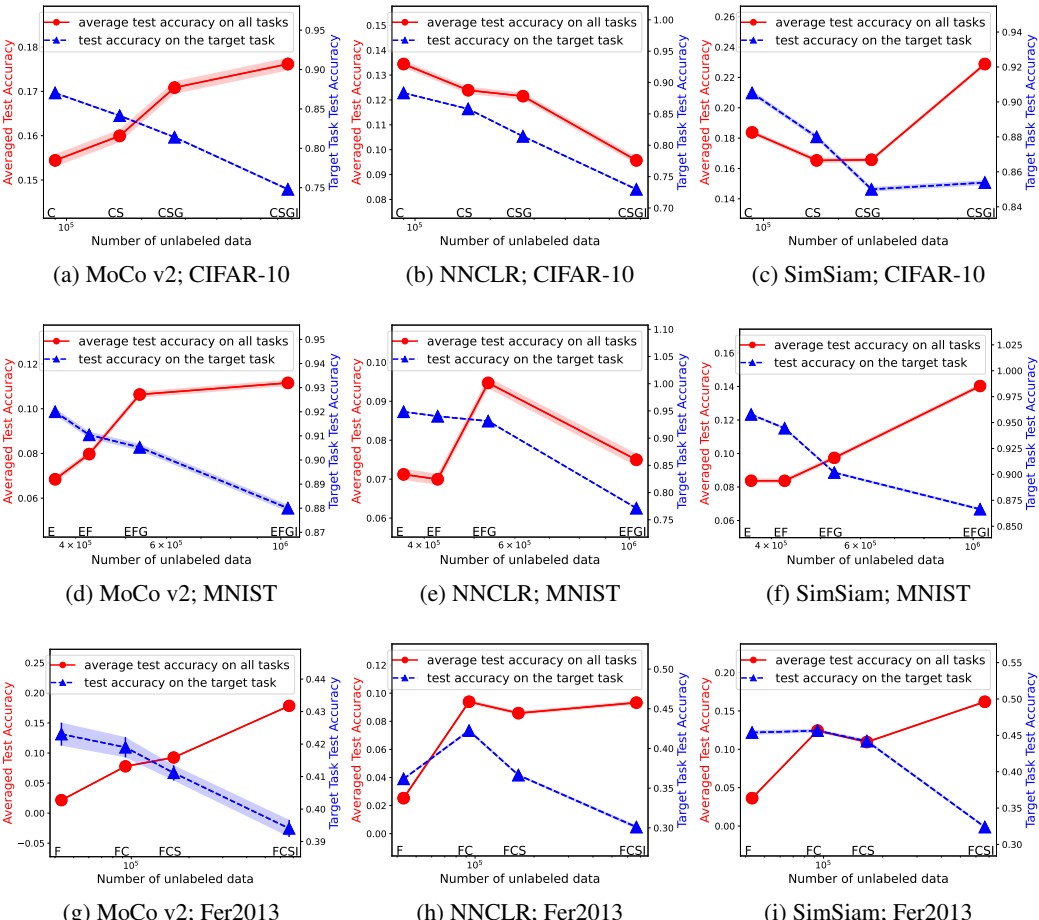

Figure 17: Trade-off of universality and label efficiency for MoCo v2, NNCLR, SimSiam on downstream tasks CIFAR-10, MNIST, Fer2013. $x$-axis: incrementally add datasets for pre-training. Pre-training data: (a)(b)(c) CINIC-10 (C), SVHN (S), GTSRB (G), and ImageNet32 (I). For example, "CS" on the $x$-axis means CINIC-10+SVHN. Target task: CIFAR-10. Red line: average test accuracy of Linear Probing on all 4 datasets. Blue line: test accuracy on the target task. (d)(e)(f) EMNIST-Digits&Letters (E), Fashon-MNIST (F), GTSRB (G), ImageNet32 (I). Target: MNIST. (g)(h)(i) FaceScrub (F), CIFAR-10 (C), SVHN (S), ImageNet32 (I). Target: Fer2013. All evaluations are trained with 20% labeled data.

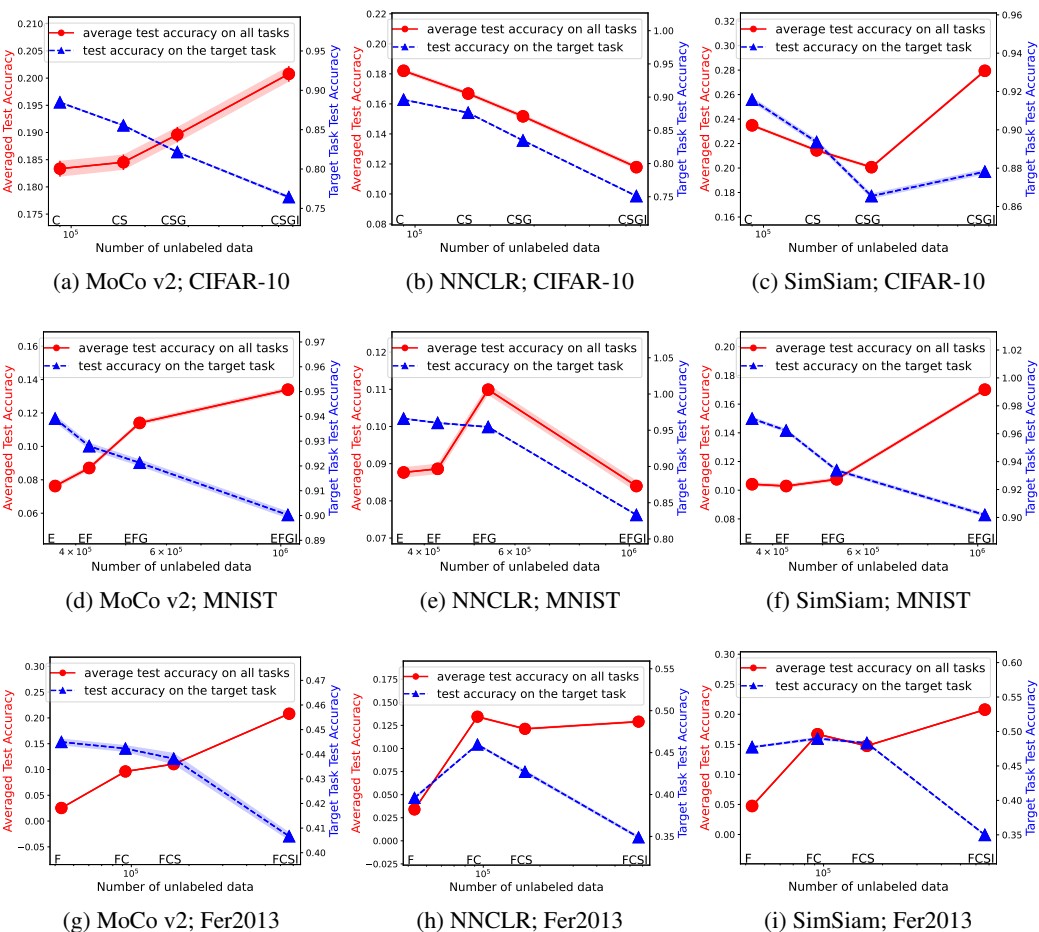

Figure 18: Trade-off of universality and label efficiency for MoCo v2, NNCLR, SimSiam on downstream tasks CIFAR-10, MNIST, Fer2013. $x$-axis: incrementally add datasets for pre-training. Pre-training data: (a)(b)(c) CINIC-10 (C), SVHN (S), GTSRB (G), and ImageNet32 (I). For example, "CS" on the $x$-axis means CINIC-10+SVHN. Target task: CIFAR-10. Red line: average test accuracy of Linear Probing on all 4 datasets. Blue line: test accuracy on the target task. (d)(e)(f) EMNIST-Digits&Letters (E), Fashon-MNIST (F), GTSRB (G), ImageNet32 (I). Target: MNIST. (g)(h)(i) FaceScrub (F), CIFAR-10 (C), SVHN (S), ImageNet32 (I). Target: Fer2013. All evaluations are trained with 100% labeled data.

