# OpenReview forum: "The Trade-off between Universality and Label Efficiency of Representations from Contrastive Learning"
_ICLR.cc/2023/Conference — ICLR 2023 notable top 25%_

### Official Review · Reviewer_P3pG · 2022-10-22

**Confidence:** 2
**Correctness:** 3
**Technical Novelty And Significance:** 4
**Empirical Novelty And Significance:** 3
**Recommendation:** 6

**Clarity, Quality, Novelty And Reproducibility:**

Although the paper is overall well written, a few unclear parts exist. I may need more information to evaluate the quality and novelty of this work.

- Figure 1:
  - "Specifically, we first pre-train on a specific dataset similar to that of the target task, and then incrementally add more datasets into pre-training": Does this mean the training is done in an online learning fashion? For example, when training on the CSG datasets, was the trained model for the CS datasets used? If so, does the order of datasets matter much?
  - Shouldn't one dimension (e.g., x-axis) be the number of *labeled* target task data if the *label efficiency* is defined with pre-trained representation and downstream prediction task?
  - The two y-axes may be confusing. The difference in average accuracy is around 2%, while the difference in target task accuracy is more than 10%. The absolute value seems suspicious too. Why is the average test accuracy less than 20% while the target task accuracy is more than 75%?
- Notation: $\phi: \mathcal{X} \mapsto \bar{\mathcal{Z}}$ does not follow the convention. A function $f$ with domain $X$ and codomain $Y$ is denoted $f: X \to Y$, and a function $f$ mapping $x$ to $y$ is denoted $f: x \mapsto y$, e.g., $\mathrm{square}: \mathbb{R} \to [0, \infty) := x \mapsto x^2$.
- Maybe assumption A1 can be restated more succinctly as "$g: z \mapsto x$ is injective".
- The term "regular" is used in many fields of mathematics, but I'm unaware of its use in this sense. Is there a more informative name for it? By the way, the implication is not obvious to me. Is it proven somewhere?
- The loss $-t$ was referred to as "unhinged loss" in some papers. It is not negatively bounded, so it does not satisfy A3. Then, what is the purpose of Theorem 2.1? It seems not so related to the trade-off to me.
- The trade-off sounds really like disentanglement. It would be better if the author could discuss their relationship and differences.

**Strength And Weaknesses:**

(I'm not very familiar with theoretical analysis of contrastive learning and several previous works. I did not check the proof details in the appendix. My review might be biased.)

### Strengths

- The author aimed to analyze an important aspect of contrastive learning.
- This paper is well-written and contextualized. The introduction section provides a nice overview of this work.
- The author conducted abundant experiments to support their theory.

### Weaknesses

- How Theorem 2.2 is related to label efficiency is unclear to me. The explanation sounds more like "pre-training on multiple datasets vs. training for a specific dataset" instead of "representation that achieves the optimal performance for many tasks but requires a lot of labeled data (extreme case: identity function) vs. representation that is easy to train for a specific task but can not be used for other tasks (extreme case: the optimal predictor for a task)".
- The experimental settings seem detached from the theory. Figures 1 and 3 show the number of unlabeled data instead of labeled data.

**Summary Of The Paper:**

This paper theoretically analyzed the trade-off between **universality** (measured by the average performance of multiple tasks) and **label efficiency** (measured by the amount of labeled data needed for a downstream task) in **constrastive learning**. Both empirical evidence and theoretical guarantee were given.

**Summary Of The Review:**

This paper presents solid work on the analysis of contrastive learning with theoretical guarantees and empirical evidence. However, I'm not certain how significant and novel the result is. I may need more time to read it again to give my recommendation. For now, I lean toward acceptance with low confidence.

---

> ### Author Response · Authors · 2022-11-10
> **Clarifications Part 3/3**
>
> ## The trade-off sounds like a disentanglement
> Roughly speaking, our analysis of the trade-off shows the learned representation separates out the invariant and the spurious features but does not show it disentangles the invariant features. (We are not completely sure about the exact meaning of disentanglement in the review, since it’s typically used as a high-level and intuitive term and there is no generally agreed-upon definition. The discussion here is based on our understanding of disentanglement following [1,2,3].)
>
> [1] mentioned “the effectiveness of deep learning is often ascribed to the ability of deep networks to learn representations that are insensitive
> (invariant) to nuisances such as translations, rotations, occlusions, and also “disentangled,” that is, separating factors in the high-dimensional space of data (Bengio, 2009).” Our analysis confirms that contrastive learning encodes the invariant features and removes the spurious features, so insensitive to the nuisances corresponding to the transformations used. However, our analysis only shows that the representation encodes the invariant features but does not show that the invariant features are disentangled. It can be a rotation of the weighted invariant features (see Proposition 2.3 and 2.4). Due to the rotation, one dimension of the representation vector can depend on multiple invariant features, so it’s not disentangled. Still, this is sufficient for us to show the trade-off. We further note that even if there is no rotation so that the representation is disentangled, we can still have the trade-off. The trade-off happens due to that pre-training on diverse data can emphasize the common features across all tasks and thus relatively down-weight the task-specific features. We thus do not see a direct connection between the trade-off and the disentanglement.
>
> Finally, although our analysis does not show the disentanglement of the invariant features, this doesn’t mean there is no disentanglement. It is just that our analysis does not need disentanglement.
>
> [1] Achille, Alessandro, and Stefano Soatto. "Emergence of invariance and disentanglement in deep representations." The Journal of Machine Learning Research 19.1 (2018): 1947-1980.
>
> [2] Higgins, Irina, et al. "Towards a definition of disentangled representations." arXiv preprint arXiv:1812.02230 (2018).
>
> [3] Bengio, Yoshua. "Learning deep architectures for AI." Foundations and trends in Machine Learning 2.1 (2009): 1-127.

---

> ### Author Response · Authors · 2022-11-10
> **Clarifications Part 2/3**
>
> ## Question about Figure 1
> 1. The training does not follow the online learning fashion. The model will pre-train from scratch (random initialization) on the CSG datasets rather than using the model pre-trained on the CS datasets. Also, all data points are shuffled randomly. Thus, the order of datasets does not matter in each pre-training.
> 2.  As we discussed before, we use target task performance (right $y$ axes) to evaluate the label efficiency. The x-axis corresponds to the pre-training unlabeled datasets.
> 3. For the average performance in Figure 1, it is the weighted average test accuracy on CIFAR10, SVHN, GTSRB, and ImageNet32. The weight is proportional to the class number in each dataset (consistent with random guessing as a baseline). Note that the ImageNet32 has 1000 classes and the top 1 test accuracy is roughly 20% (using ResNet18 on 32*32 resolution images). The difference in average accuracy across different pre-training settings is around 2%, which is significant when the class number is larger than 1000. For the target task, CIFAR10 only has 10 classes. Thus, the target task test accuracy may have a larger variation (more than 10%) and higher value (more than 75%).
>
> ## Notation and assumption
> 1. We fixed the notation in the revision version. Thank you for pointing this out.
> 2.  In assumption A1, $g$ is injective only when the dimension of $z_U$ is 0. To avoid misunderstanding, we do not use “injective”.
> 3.  Here we need some mild conditions on the functions to rule out some rare pathological cases. For simplicity, we call the condition “regular” though it’s not the best naming. We used the condition in the proof of Theorem 2.2, to extend our statement from a finite $z$ case to an infinite $z$ case. See the last paragraph in the proof of Theorem 2.2 in Appendix A.
>
> ## Theorem 2.1 and Loss -t
> 1. Theorem 2.1 is to analyze the pre-training stage and not directly about the trade-off.
> Recall that the pre-training paradigm consists of two stages: pre-training the representation, and learning the predictor on the representation for the target task. So we first study the pre-training stage in Section 2.1, and then study the prediction performance and the trade-off in Section 2.2. To study the pre-training, we begin with understanding the contrastive learning objective (Theorem 2.1) and then what features are encoded (Theorem 2.2).
>
> 2. Loss -t is used in Theorem 2.1, while A3 is used in Theorem 2.2 on some other losses.
> For the pre-training, note that the contrastive learning objective can use various loss functions $\ell(t)$, including those not satisfying A3 (e.g., $\ell(t) = -t$) and those satisfying A3 (e.g., the typical logistic loss).  Theorem 2.1 considers the former and shows that $\ell(t)=-t$ corresponds to nonlinear PCA, so contrastive learning can be viewed as its generalization. Some intuition of PCA (encoding more “spread-out” features) may apply to other losses. Following this, Theorem 2.2 turns to other losses (those satisfying A3), and analyzes what features are encoded by contrastive learning with these losses. These results lay the foundation for the analysis of the trade-off in Section 2.2.

---

> ### Author Response · Authors · 2022-11-10
> **Clarifications Part 1/3**
>
> We thank the reviewer for the detailed comments. We will provide clarifications below.
>
> ## Theorem 2.2 and label efficiency
> We will first clarify label efficiency and then clarify Theorem 2.2. In summary, Theorem 2.2 is analyzing the pre-training stage and is only indirectly related to label efficiency.
>
> ### Label efficiency
> Our focus is to study the two key properties, label efficiency and universality, of the pre-trained representations, and how they are affected by the pre-training. Here, label efficiency means the ability to learn an accurate classifier on top of the representation with a small amount of labeled data, and universality means usefulness across a wide range of downstream tasks. We would like to compare the two settings: pre-training on diverse data vs pre-training on specific data.
>
> For the study, we need to make the above more formal. To this end, we model pre-training on diverse data by pre-training on a mixture of various tasks (multiple datasets), and model pre-training on specific data by pre-training on a specific task (one dataset). The trade-off then means: compared to pre-training on a specific dataset, pre-training on diverse datasets can improve the average prediction accuracy on all the tasks, but harm the performance on the specific task.  Note that we use the prediction performance on a specific target task to evaluate label efficiency. This then leads to our analysis and experimental setups.
>
> ### Theorem 2.2
> The theorem is indirectly related to label efficiency (i.e., the prediction performance on the target task). Recall that the pre-training paradigm consists of two stages: pre-training the representation using unlabeled data, then learning a predictor on top of the representation using labeled data from the target task. To analyze the prediction performance on the target task in the second stage, we first need to analyze the pre-trained representation from the first stage. Theorem 2.2 is analyzing the representation, but not yet on the prediction performance (that is done in Proposition 2.3 and 2.4).
>
> While Theorem 2.2 does not directly analyze the prediction performance, its insights on the representation give hints for the analysis of the prediction performance. It says that the representation will encode invariant features and emphasize the more “spread-out” ones. The explanation after the theorem applies it to the two settings in the study of the trade-off: pre-training on multiple datasets v.s. pre-training on a specific dataset. It suggests that (1) in the first setting, the representation encodes all the invariant features from all the tasks, leading to usefulness over all the tasks; (2) in the second setting, features unique to this specific dataset are more “spread-out” than those unique to other datasets, and gets emphasized, leading to better prediction on this specific dataset. In other words, pre-training over diverse data will relatively down-weight the features unique to the specific dataset while encoding useful features for all the datasets. This leads to the trade-off. Such intuition is later formalized in Proposition 2.3 and 2.4.
>
> ## The experimental settings and x-axis of Figure 1 and 3
> 1. The x-axes of Figure 1 and 3 indeed correspond to the unlabeled data.
> As we discussed above, we would like to compare pre-training on a specific task v.s. pre-training on a mixture of multiple tasks. Therefore, we begin with pre-training on a specific task, and incrementally add other tasks into pre-training. For example, in Figure 1, we have four settings: pre-training on CINIC-10 (denoted as C on the x-axis), on CINIC-10+SVHN (denoted as CS), on CINIC-10+SVHN+GTSRB (denoted as CSG), on CINIC-10+SVHN+GTSRB+ImageNet32 (denoted as CSGI). These are the unlabeled data and thus the x-axis corresponds to unlabeled data.
>
> 2. The y-axes in Figure 1 and 3 are related to labeled data.
> As discussed above, we use the prediction performance on a specific target task to evaluate label efficiency (right $y$ axes in Figures 1 and 3) and use the average performance of multiple tasks to evaluate universality (left $y$ axes in Figures 1 and 3).  To get the prediction performance on the target task, we use the labeled data from the task (e.g., CIFAR-10 in Figure 1). Similarly, to get the average performance of multiple tasks, we use the labeled data from each of them to get the individual performance and then compute the weighted average.

---

> ### Author Response · Authors · 2022-11-18
> **Follow up with Reviewer P3pG**
>
> Dear Reviewer,
>
> Since the deadline for the author-reviewer discussion phase is fast approaching, we would like to follow up with you to see if you have any further questions. In our rebuttal, we have tried to address all your questions. In particular, we made a clarification for Theorem 2.2 and Figure1. We are willing to provide more clarification if you have any additional concerns. Thank you!
>
> Best,
>
> Authors

---

### Official Review · Reviewer_gG3v · 2022-10-24

**Confidence:** 3
**Correctness:** 4
**Technical Novelty And Significance:** 3
**Empirical Novelty And Significance:** 4
**Recommendation:** 8

**Clarity, Quality, Novelty And Reproducibility:**

The purpose of the research, the issues and the principles of the contribution are clearly explained and defined. The subject is clearly introduced and explained as well as the previous works.

The theoretical analysis could be better explained and simplified for readers who are not familiar with the notions presented.

For the experiments, the methodology is presented correctly, and the goal linked to the hypotheses introduced by the authors.

However, the results could be better presented and explained. For example, the results shown in Figure 4 could be better explained. At first glance, it is difficult to understand what was evaluated, and the relationship between the observed similarities and the data used for pre-training.

**Strength And Weaknesses:**

In this paper the authors argue for the existence of a trade-off between the universality and the label efficiency of pre-trained representations. While the formal and empirical proofs for the existence of this trade-off are convincing and strongly supported, it seems to me that it can be reduced to the traditional no-free lunch principle well known in machine learning, and the universality/task specific features to the bias/variance trade-off also well known. It is therefore not surprising that representations that are good for a variety of tasks are not well adapted for a specific one.

However, I think the authors are correct when they state that this trade-off required a deep and complete study and the formal and experimental analysis proposed by the authors bring valuable information, results and insights. Although the formal analysis could be clearer and explained more simply, it shows what features are learned by contrastive learning when trained on diverse or specific data. It also shows how the data chosen for pretraining impact the prediction performances of predictor and gives an interesting framework for further investigation. Furthermore, I think that the evidence is numerous enough and that the experiments carried out are rigorous enough to support the assertions of the authors.

The more interesting contribution of this paper is the contrastive regularisation loss. Here also the evidence is clearly stated and strong enough to demonstrate a significant effect on the prediction performances. Results shows clearly that adding contrastive regularisation improves performances over traditional training protocol, which is a valuable information and contribution for the community.

**Summary Of The Paper:**

In this paper, the authors propose to explore the pre-training of representations with contrastive learning and the impact of training data diversity on the prediction performances of linear predictors. They propose that a trade-off exists between universality of pre-trained representations (diverse data) and efficiency on downstream tasks. They confirm their hypothesis first by a formal demonstration on a simplified model and then by empirical evaluations. Based on resulting insights, they propose a contrastive regularisation loss and proceed to empirical comparisons with other training protocols.

For their formal demonstration, the authors propose to formalize the concept of semantic features useful for specific tasks. They then introduce a hidden representation data model which allows to sample over the distribution of the hidden representations and to separate the learned features into two subsets: spurious features that are affected by transformations and invariants features that stay constant over different tasks. The authors then use this formal description to show that contrastive learning encodes features from unlabelled data which appear to be shared by a diversity of tasks, while down-weighting features specific to a task. Their analysis also shows that contrastive learning favours the invariant features with high variances, the authors conclude that contrastive learning is a generalized nonlinear PCA.

Based on the theoretical analysis, the authors proposed to test their claim empirically. For this, they evaluated the similarity of features using Centered Kernel Alignment in two ways. First with pre-trained features on different dataset for a target task, and secondly with features pre-trained on gradually increasing datasets. Results showed low similarities between features pre-trained on the different datasets, which confirms that contrastive learning encodes mainly features that are private to these tasks. Results also showed decreasing similarities between features pre-trained on all datasets and those specific to a task, confirming that representations obtained on diverse data encodes more information irrelevant to downstream tasks.

The authors also evaluated their proposed contrastive regularization method empirically. For that, they compared test results obtained with a predictor trained with contrastive regularization and other training protocols. Results showed that contrastive regularisation outperformed other fine-tuning methods. Finally, authors compared their proposed method on large representation models (foundation models) to the same baseline fine-tuning protocol and showed that their method can improve prediction performance

**Summary Of The Review:**

Even if the compromise between invariant and spurious representations presented in the paper seems to me to be a reformulation of well-known principles in machine learning, the formal and empirical analysis proposed by the authors brings interesting and important contributions to the community. The experimental results confirm the theoretical analysis and the interest of contrastive regularization for models based on pre-training and contrastive learning.

---

> ### Author Response · Authors · 2022-11-10
> **Clarification**
>
> We thank the reviewer for the detailed positive reviews. We will improve the presentation of the results.
>
> We would like to clarify that our analysis of the trade-off is not a reformulation of or reduction to the No Free Lunch principle (NFL), though appearing to have that flavor.
>
> In our setting, there is a representation function that works for all the tasks, i.e., the function that recovers the ground-truth invariant features. It’s just that the contrastive learning algorithm doesn’t find it. In contrast, NFL says no algorithm can work well for all the tasks. This means no single solution works well for all tasks; otherwise, the algorithm can just always output the universally good representation function. So our result does not come from NFL.
>
> The technical reason for the difference is that NFL considers all possible tasks, and compares the number of tasks for which algorithm A outperforms algorithm B to the number of tasks for which B outperforms A (usually using the tool of uniform randomness or other prior distributions over all possible tasks). Our analysis is not over all possible tasks, but for a set of tasks with specific structures: e.g., common features and task-specific features. The purpose of our setup is to capture the structure of the practical tasks, since practical tasks are clearly not arbitrary.
>
> On the other hand, it’s an interesting question that over what family of tasks we can get NFL. Intuitively, NFL should hold over all possible tasks (properly defined, e.g., over discrete input/output spaces). However, that’s different from our setting.

---

### Official Review · Reviewer_heJa · 2022-10-25

**Confidence:** 4
**Correctness:** 4
**Technical Novelty And Significance:** 4
**Empirical Novelty And Significance:** 2
**Recommendation:** 8

**Clarity, Quality, Novelty And Reproducibility:**

well-written, high quality, the novelty is clear, and reproducibility should be good (easy to implement)

**Strength And Weaknesses:**

This is a well-written paper, the idea is very motivated, and the analysis seems solid and interesting.

One thing I'd like to see in the experiments is some results that can verify Thm 2.2. Fig. 4 shows some, but not sufficient. Is it possible to show some results related to the increase of $B_r$? For instance, will it be good to show some similarity matrices with different dimensions of features, e.g., 512, 1024, 2048, similar to Fig. 4? Then by comparing the off-diagonal similarity, if the average similarity decreases with the increase of dimensions, then it will verify the theorem (correct me if I am wrong), to a certain degree.

**Summary Of The Paper:**

The work claims there is a trade-off between two properties of representation learning 1. When a model is pre-trained, the learned representation will be useful for downstream tasks with a small number of labels. 2. The representation is useful for different downstream tasks. i.e. the average accuracy on multiple downstream tasks increases while single-task test accuracy decreases with more unlabeled data. The work provides some theoretical analysis. Starting from the connection between infoNCE loss and nonlinear PCA, the work conjecture that contrastive learning encodes important invariant features but not spurious ones.

**Summary Of The Review:**

Overall this is a good theoretical paper for understanding contrastive learning.

---

> ### Author Response · Authors · 2022-11-10
> **New experimental results**
>
> We thank the reviewer for the positive comments.
>
> Our new experimental results below follow the suggestion of the reviewer and provide support for Theorem 2.2.
>
> We would like to clarify that in Theorem 2.2, the key factor is the bound of the norm $B_r$, instead of the dimensions of the representation. The theorem says that when we increase the norm bound, the representation can encode more and more features. To verify this, our experiments will vary the weight decay regularization coefficient (larger weight decay corresponds to a smaller norm bound and fewer features learned in the representation).   We can see that the linear CKA similarity decreases with the increase of the weight decay, then it verifies our theorem to a certain degree.
>
>
> | weight decay coefficient   | 5e-5    | 5e-4 | 5e-3 |
> |-----------------------------------|-------|----------------|----------------|
> | 5e-5                   | 1.0 |  0.651         | 0.641         |
> |5e-4                    |  | 1.0          |  0.675          |
> |5e-3                    |  |          | 1.0          |
>
> Details: The table shows linear CKA similarity among CIFAR10 features. Columns and rows correspond to the weight decay parameter used in the model MoCo v2 with ResNet18 backbone. The pre-training data is CINIC10.

---

### Official Review · Reviewer_mgh6 · 2022-10-25

**Confidence:** 4
**Correctness:** 3
**Technical Novelty And Significance:** 3
**Empirical Novelty And Significance:** 3
**Recommendation:** 6

**Clarity, Quality, Novelty And Reproducibility:**

The paper is clearly written. The novelty might be an issue here, as I think the new regularization trick is not first proposed in this work. But the tradeoff analysis might be, though it relies on too strong assumptions. The results should be reproducible.

**Strength And Weaknesses:**

Strength:

The theoretical analysis and empirical results about the tradeoff between target performance and the pretraining dataset are interesting, at least on the small-scale and small-resolution datasets. The improvement by the contrastive regularization trick is also exciting and promising.

Weaknesses:

The biggest issue, in my opinion, is that it’s unclear how significant this tradeoff is in large-scale and large-resolution datasets. The tested case is mini compared to the actual training datasets people would use to get foundation models in both the resolution and the scale. Isn’t the fact that the CLIP, MoCo v3, and MAE can work already suggesting that this tradeoff is not critical? If the authors can show this tradeoff in high-resolution datasets in the 1M to 10M scales (which is still much smaller than what people may use but is much closer), I would be more convinced.

Many things in this work are not novel, which is another big issue, IMO (see the next section for more details).

Although the regularization trick leads to consistent improvement, fine-tuning is much more important in almost all cases. Is this trick useful in the Linear Probing case?

(some of these weaknesses have been addressed by the authors, see their response and my follow-up comment)

**Summary Of The Paper:**

This work shows a trade-off between training on a large dataset and the performance on a target dataset. The authors first show some theoretical analysis of why this tradeoff appears. The authors then show empirical results that this tradeoff happens when training the networks on small-scale and small-resolution datasets. Furthermore, they show that a finetuning trick by including the contrastive loss during the finetuning helps the final performance, even in large-scale and large-resolution datasets with deep architectures.

**Summary Of The Review:**

This paper shows interesting results about the tradeoff between the specific target task and the diversity in the training dataset. However, it’s unclear how important this tradeoff is in real-world applications. It is also not clear how novel the things in this paper are.

---

> ### Author Response · Authors · 2022-11-10
> **More experiments and Clarification Part 3/3**
>
> ## Linear probing and contrastive regularization
>
> Our experimental results show that contrastive regularization can improve over linear probing. We would also like to clarify that contrastive regularization is of comparable importance as fine-tuning in training the predictor.
>
> ### Contrastive regularization then Linear probing
> Contrastive regularization can also improve over linear probing. The results in the table below show that contrastive regularization leads to better prediction accuracy. Furthermore, the improvement is more significant on diverse pre-training data, consistent with our analysis.
>
> | Pre-training task | CINIC10    |  +SVHN  | +GTSRB  | +ImageNet32  |
> |-------------------|-------|-------|--------|--------------|
> | Linear probing                       | 88.41 | 85.18          | 82.07 |75.64 |
> | Contrastive regularization then Linear probing    | 88.38 | 86.91      | 85.95  | 82.43 |
>
> Details: Columns correspond to pre-training datasets, and rows correspond to the two methods for training the predictor. The model is MoCo v2 with ResNet18 backbone. The Contrastive regularization task is CIFAR10 and the target task is CIFAR10.
> (Note that the contrastive regularization loss term is only relevant with the backbone; see the definition in Equation (9). While linear probing fixes the backbone. Thus, we cannot do contrastive regularization and linear probing at the same time. To show its effect, we first apply contrastive regularization to update the backbone, and after that use linear probing. )
>
> ### Effect of Contrastive regularization
> We would like to clarify that in training the predictor, contrastive regularization is of comparable importance as fine-tuning; in some cases, the former leads to more improvement and in other cases, the latter leads to more.
>
> We emphasize that in Table 1 and 2 in Section 3.3, the results in the row “ FT” are for fine-tuning + augmented data, not just for the typical fine-tuning without data augmentation. (This is described in the Evaluation & Methods paragraph and Table 2 caption.) In particular, there are three techniques in training the predictor: fine-tuning, data augmentation using transformations, and contrastive loss. Our regularization method uses all three, while the typical training uses only the first one. The results in the row “FT” in Table 1 and 2 are for fine-tuning with the same amount of data augmentation as our method, so as to evaluate the effect of the third technique alone (contrastive loss).
>
> Therefore,  to evaluate the importance of different techniques, we should compare the following: linear probing, typical fine-tuning without data augmentation, fine-tuning with data augmentation, and our method with the same amount of data augmentation. (The results have been included in Appendix C.4, in particular, Table 4 and 5. Due to space limitation, only part of the results are presented in the main body to evaluate the effect of the contrastive loss alone.) Here we present the results on CLIP:
>
> | Target task | ImageNet    | SVHN | GTSRB |
> |-------------------|-------|----------------|----------------|
> | Linear probing                          | 77.84 |  63.44         | 86.56         |
> | Typical fine-tuning without data augmentation        | 82.97 | 65.76          |  89.83          |
> |  Fine-tuning with 10 data augmentations                             |  83.65 | 78.22          | 90.74          |
> |  Ours with 10 data augmentations                            | 84.94 | 78.72          | 92.01          |
>
> We can see that while the typical fine-tuning (without data augmentation) can lead to significant improvement over linear probing, our method gives significant improvement over fine-tuning which can sometimes be larger. And for that improvement, sometimes contrastive loss is more important (e.g., on GTSRB) while other times data augmentation is more important (e.g., on SVHN).

---

> ### Author Response · Authors · 2022-11-10
> **More experiments and Clarification Part 2/3**
>
> ## Novelty
> We respectfully disagree with the comment that some of our key contributions are not novel. Our connection to PCA is different from existing work, and our other theoretical results are also novel. We are also not aware of publications proposing contrastive regularization. We would appreciate it if the reviewer can provide specific pointers to related references.
>
> ### The connection to nonlinear PCA
> The connection we established is different from some connections in existing work. Our result is that, with a special case of the loss function, the contrastive learning is equivalent to nonlinear PCA on the smoothed data in our hidden representation data model.
>
> (a) Our conclusion is different. Existing work [3] proved that contrastive learning with linear representation functions is a reparameterization of linear PCA on the contrastive covariance of input. The connection is established only for linear representations, while ours is not limited to linear ones. Furthermore, their result is for PCA on a sophisticated statistic of the input called the contrastive covariance, while ours is on the smoothed data (averaged over the transformations used in contrastive learning). Therefore, the results are very different. There are also other works (e.g., [4,5])  connecting contrastive learning to factor analysis on neighboring structures. However, their connections are to spectral methods on the neighboring graph/matrix on the inputs that are different from PCA, and thus those connections are very different from our PCA connection.
>
> (b) Our analysis setup is novel. To study the trade-off, we assume a hidden representation model for generating the data: first sample a hidden representation, and then use it to generate the input and the label; the hidden representation consists of invariant features (not affected by the transformations in the contrastive learning) and spurious features (affected by the transformations). None of the existing work analyzing contrastive learning uses this hidden representation model.
>
> (c) Our analysis techniques are different. The analysis (in Appendix A) makes use of the partitioning of the hidden features into invariant and spurious features, and the connection between the PCA objective and the contrastive learning objective for $\ell(t)=-t$. These have not been used in existing work as far as we know.
>
>
> ### The tradeoff analysis
> Our analysis setup, tools, and results are all novel. As discussed above, our hidden representation data model with invariance/spurious features has not been used for analyzing contrastive learning. The technical tools include the convexity combined with the invariance/symmetry in the data, and properties of the chi-distributions (see Appendix B). The resulting bounds the generalization error of downstream prediction in terms of the pre-training data properties, which have not been demonstrated before. If the reviewer can give references with similar analyses, we are happy to cite and compare.
>
> ### Our regularization method
> We are not aware of existing work explicitly proposing this method for better adaptation of pre-trained models, in particular, among peer-viewed publications. If the reviewer can provide references to existing publications proposing the method, we are happy to discuss further and adjust the writing when needed.
>
> [3] Tian, Yuandong. "Understanding Deep Contrastive Learning via Coordinate-wise Optimization." arXiv preprint arXiv:2201.12680 (2022).
>
> [4] Balestriero, Randall, and Yann LeCun. "Contrastive and non-contrastive self-supervised learning recover global and local spectral embedding methods." arXiv preprint arXiv:2205.11508 (2022).
>
> [5] Ko, Ching-Yun, et al. "Revisiting contrastive learning through the lens of neighborhood component analysis: an integrated framework." International Conference on Machine Learning. PMLR, 2022.

---

> ### Author Response · Authors · 2022-11-10
> **More experiments and Clarification Part 1/3**
>
> We thank the reviewer for the comments and address the questions below.
>
> ## Larger scale experiments
> Our new experiments on large datasets also confirm the trade-off.
>
> ### Our results on ImageNet with MoCo v3
> We have conducted experiments on ImageNet (~1M images of resolution 224*224) with MoCo v3. Comparing pre-training on the diverse data ImageNet to pre-training on the specific data of bird images (or vehicle images), we observe that the accuracy of the specific task decreases while the accuracy over all classes in ImageNet increases. This confirms the trade-off.
>
> |           | Pre-train on bird images | Pre-train on ImageNet |
> |-------------------|-------------------------------|-------------------------------|
> | Classification Accuracy on birds| 81.42                      | 69.39   |
> | Classification Accuracy on ImageNet | 41.38                      | 73.20   |
>
> Experimental setup: ImageNet (1.2M+ images of resolution 224*224) is used as the diverse dataset, all bird images from ImageNet (72k+ images, 59 classes) are used as the specific dataset, and the specific target task is classifying the birds using 1k labeled training data. MoCo v3 with the backbone ViT-S is used.
>
> |           | Pre-train on vehicle images | Pre-train on ImageNet |
> |-------------------|-------------------------------|-------------------------------|
> |  Classification Accuracy on vehicles  | 70.93                      | 67.90   |
> |  Classification Accuracy on ImageNet | 39.84                      | 73.20   |
>
> Experimental setup: Same as above, except that the specific dataset/task is the vehicle images  (55k+ images, 43 classes).
>
> ### Experimental support from related work
> There are also some scattered related experimental results reported in existing studies, though they didn’t study the universality-efficiency trade-off. In Table 2 from [1], the iNat21, ImageNet, Places365, and GLC20 datasets are used for SimCLR, and the pre-training dataset size is around 1M with the resolution 224*224. Here are the most related results from there (please refer to the paper [1] for the full results).
>
> | | In-Domain (250k) |  In-Domain (1M) | (iNat21+ImageNet+Places365+GLC20)*250k |
> |-------------------|-------------------------------|-------------------------------|-------------------------------|
> | iNat21                            | 45.1             | 49.3        | 41.0                                   |
> | ImageNet                          | 60.8          | 64.4       | 57.4                                   |
> | Places365                         | 48.5           | 50.1       | 48.2                                   |
>
> Details: the rows correspond to target tasks, and the columns correspond to pre-training datasets. The column In-Domain(250k) means using 250k data points from the target task for pre-training. The column (iNat21+ImageNet+Places365+GLC20)*250k means using 1M points with 250k from each of the 4 datasets. For example, the entry 45.1 for row=iNat21 and column=In-Domain means the test accuracy on iNat21 using the representation pre-trained on 250k images from iNat21.
>
> The results showed that pooling datasets (i.e., adding diverse data for pre-training) decrease prediction performance relative to in-domain pre-training (i.e., pre-training only on the specific task data). These also provide support for our claim about the trade-off. Note that [1] didn’t consider the universality aspect and also didn’t perform further investigations. Instead, they hypothesized that contrastive learning on diverse data is easier and results in lower-quality representation and called for more investigation, which is different from our focus.
>
> More generally, the survey [2] also pointed out that for foundation models “training on a more diverse dataset is not always better for downstream performance than a more specialized foundation model” (on page 112, Specialization vs. diversity in foundation model training data).
>
> ### Summary
> In summary, the trade-off is critical. The pre-training data have a fundamental impact on the trained representation and thus the downstream prediction performance. The trade-off between universality and efficiency is a critical aspect, as it is directly on the two of the most important properties of pre-trained representations. The investigation can provide hints on designing better pre-training/adaptation methods or better construction of pre-training data, and thus allows further improvement over the current working performance.
>
> [1] Cole, Elijah, et al. "When does contrastive visual representation learning work?." Proceedings of the IEEE/CVF Conference on Computer Vision and Pattern Recognition. 2022.
>
> [2] Bommasani, Hudson, et al. "On the Opportunities and Risks of Foundation Models." arXiv preprint arXiv: 2108.07258 (2021).

---

> ### Comment · Reviewer_mgh6 · 2022-11-14
> **score improved and response below**
>
> I thank the authors for their detailed responses and new results. I will improve my score to 6 and am open to further improving it if the authors can add other results I am suggesting below.
> About the large-scale large-resolution comment, the new results confirm that this trade-off also happens in this case, at least for models trained on 1M images. That being said, it would be even better if the authors could show that this trade-off still exists for models trained on even more images (like 5M), as that’s closer to (though still much less than) the number used in real applications. My intuition is that this trade-off will disappear with more images and larger architectures, but I am happy to be wrong about this intuition.
> For the novelty comment, I agree with the authors that the theoretical analysis differs from the PCA analysis. Although I think this contrastive regularization method is not first proposed in this paper (as the authors mentioned in their work, similar methods were used in other situations), it is exciting and promising that consistent improvement can be gained from applying this method to the fine-tuning case.
> Finally, to link the new results to the proposed regularization method, it would be great to see that the method can exactly bridge the gap between the narrow and the general pretraining shown in the first comment. That will make the logic of the whole paper much more coherent.

---

> > ### Author Response · Authors · 2022-11-18
> > **Thanks for raising score and More experiments**
> >
> > We thank the reviewer for raising the score! We will address the additional comments below.
> >
> > ## Larger scale experiments
> > Our new experiments pretrained on more than 16M images also confirm the trade-off.
> >
> > The datasets involved are ImageNet1k (1.2M data points, 1k classes), ImageNet22k (14M, 22k classes), and GCC-15M (15M). We compared two UniCL [1] representations: the more specific representation pre-trained on ImageNet1k, and the one pre-trained on the more diverse dataset ImageNet1k+GCC-15M. We compare their performance in two tasks: classification on ImageNet1k and classification on ImageNet22k.
> >
> > From the specific representation to the diverse one, we observe that the test accuracy on ImageNet1k decreases (i.e., efficiency drops), while the test accuracy over ImageNet22k increases (i.e., universality improves). This confirms the trade-off.
> >
> > |           | Pre-train on ImageNet1k | Pre-train on ImageNet1k+GCC-15M |
> > |-------------------|-------------------------------|-------------------------------|
> > | LP Classification Accuracy on ImageNet1k| 79.05                      | 77.66   |
> > | LP Classification Accuracy on ImageNet22k | 9.02                      | 9.86   |
> >
> > Experimental Details: ImageNet1k (1.2M images of resolution 224*224) is used as the specific dataset, and ImageNet1k+GCC-15M (a total of 16M images)  is used as the diverse dataset. The target task is classifying the ImageNet1k or ImageNet22k (a superset of ImageNet1k, 14M images) using 2 * #label labeled training data. The UniCL with the backbone Swin-T is used. Note that we use these models since the UniCL pretrained on ImageNet1k+GCC-15M is the largest scale contrastive learning experiment for classification tasks we can find after checking existing work to our best. Also, note that GCC-15M does not have well-formed classification tasks, so we use classification on ImageNet22k as a set of diverse tasks to evaluate the universality.
> >
> > ## Contrastive Regularization
> > We follow the suggestion from the reviewer and show that Contrastive Regularization can reduce the gap between the narrow and the general pretraining shown in the first comment in our previous response.
> >
> > Based on our previous response, we have conducted Contrastive Regularization on the model pretrained on ImageNet (1.2M) with MoCo v3 and then we do Linear Probing on bird images  (or vehicle images). After Contrastive Regularization, we observe that the test accuracy increases. This confirms the effect of Contrastive Regularization.
> >
> > |           | Pre-train on bird images | Pre-train on ImageNet |Pre-train on ImageNet then Contrastive Reguarlization on birds|
> > |-------------------|-------------------------------|-------------------------------|-----------------------|
> > | LP Classification Accuracy on birds| 81.42                      | 69.39   |  73.25     |
> >
> > Experimental setup: ImageNet (1.2M) is used as the diverse dataset, all bird images from ImageNet (72k+ images, 59 classes) are used as the specific dataset or regularization dataset, and the specific target task is classifying the birds using 1k labeled training data. We run Contrastive Regularization for 90 epochs. MoCo v3 with the backbone ViT-S is used.
> >
> > |           | Pre-train on vehicle images | Pre-train on ImageNet |Pre-train on ImageNet then Contrastive Reguarlization on vehicles|
> > |-------------------|-------------------------------|-------------------------------|-----------------------|
> > |  LP Classification Accuracy on vehicles  | 70.93                      | 67.90   |    71.34    |
> >
> > Experimental setup: Same as above, except that the specific dataset/task is the vehicle images  (55k+ images, 43 classes).
> >
> > [1] Yang, Jianwei, et al. "Unified contrastive learning in image-text-label space." Proceedings of the IEEE/CVF Conference on Computer Vision and Pattern Recognition. 2022.

---

### Decision · Program_Chairs · 2023-01-20

**Decision:**

Accept: notable-top-25%

**Justification For Why Not Higher Score:**

Many additional experimental results have been provided in the rebuttal and the authors did not modify the paper to integrate these elements. In addition, some remarks about the presentation have been raised. So I'm not leaning toward an oral.



**Justification For Why Not Lower Score:**

The contribution is interesting and authors have provided additional results to complete the paper. The result is of importance for the community. I think that it can be an interesting contribution for a spotlight.

**Metareview: Summary, Strengths And Weaknesses:**

This paper focuses on models pre-trained by contrastive learning and studies two important properties of such models: (i) the capacity to learn efficient predictors on top of these models when using (small) labelled data, (ii) their universality in the sense of their ability to relevant for diverse tasks.

The contribution of the paper highlights tradeoff between universality and label efficiency.
All reviewers agreed that the contribution is interesting and the is novel.
Some weaknesses about the experiments and other explanations have been raised.

In the rebuttal, authors have provided additional experimental studies that answer the issues raised in rebuttal.
The integration of these elements in the final version is important. The paper can then be improved in terms of presentation and clarity.

Overall, there is a clear consensus for accepting this paper.
I propose then acceptance.

**Note From Pc:**

if the above contains the word "oral" or "spotlight" please see: "oral" presentation means -> notable-top-5% and "spotlight" means -> notable-top-25%. As stated in our emails, we are disassociating presentation type from AC recommendations